# TABULAR DATA: IS DEEP LEARNING ALL YOU NEED?

## ABSTRACT

Tabular data represent one of the most prevalent data formats in applied machine learning, largely because they accommodate a broad spectrum of real-world problems. Existing literature has studied many of the shortcomings of neural architectures on tabular data and has repeatedly confirmed the scalability and robustness of gradient-boosted decision trees across varied datasets. However, recent deep learning models have not been subjected to a comprehensive evaluation under conditions that allow for a fair comparison with existing classical approaches. This situation motivates an investigation into whether recent deep-learning paradigms outperform classical ML methods on tabular data. Our survey fills this gap by benchmarking twenty state-of-the-art methods, spanning neural networks, classical ML and AutoML techniques. Our empirical results over 68 diverse classification datasets from a well-established benchmark indicate a paradigm shift, where Deep Learning methods outperform classical approaches.

## 1 INTRODUCTION

Tabular data has long been one of the most common and widely used data formats, with applications spanning various fields such as healthcare (Johnson et al., 2016; Ulmer et al., 2020), finance (Nureni & Adekola, 2022), and manufacturing (Chen et al., 2023), among others. Despite being a ubiquitous data modality, tabular data has only been marginally impacted by the deep learning revolution (Van Breugel & Van Der Schaar, 2024). A significant portion of the research community in tabular data continues to advocate for traditional machine learning methods, such as gradient-boosting decision trees (GBDTs) (Friedman, 2001; Chen & Guestrin, 2016; Prokhorenkova et al., 2018; Ke et al., 2017). Recent empirical studies suggest that GBDTs are still competitive for tabular data (Shwartz-Ziv & Armon, 2022; Grinsztajn et al., 2022; McElfresh et al., 2023). Nevertheless, an increasing segment of the community highlights the benefits of deep learning methods (Kadra et al., 2021; Gorishniy et al., 2021; Arik & Pfister, 2021; Somepalli et al., 2021; Kadra et al., 2024; Holzmüller et al., 2024).

The community remains divided on whether Deep Learning approaches are the undisputed state-of-the-art methods for tabular data (Shwartz-Ziv & Armon, 2022). To resolve this debate and determine the most effective methods for tabular data, multiple recent studies have focused on empirically comparing GBDTs with Deep Learning methods (Grinsztajn et al., 2022; Borisov et al., 2022; McElfresh et al., 2023). These studies suggest that tree-based models outperform deep learning models on tabular data even after tuning neural networks. However, these recent empirical surveys only include non-meta-learned neural networks (Grinsztajn et al., 2022; Borisov et al., 2022) and do not incorporate the recent stream of methods that leverage foundation models and LLMs for tabular data (Zhu et al., 2023; Hollmann et al., 2023; Yan et al., 2024; Kim et al., 2024). Furthermore, the empirical setup of the recent empirical benchmarks is sub-optimal because no thorough hyperparameter optimization (HPO) techniques were applied to carefully tune the hyperparameters of neural networks.

In this empirical survey paper, we address a simple question: *"Is Deep Learning now state-of-the-art on tabular data, compared to GBDTs?"*. Providing an unbiased and empirically justified answer to this question has a significant impact on the large community of practitioners. Therefore, we designed a large-scale experimental protocol using 68 diverse classification OpenML datasets and 20 recent baselines, including foundation models for tabular data. We classify models according to their underlying paradigm and provide a taxonomy tree in Figure 1. In our protocol, we use 10-fold cross-validation experiments for all the datasets and fairly tune the hyperparameters of all the baselines with an equally large HPO budget. In our study, we focus on the predictive quality of models rather

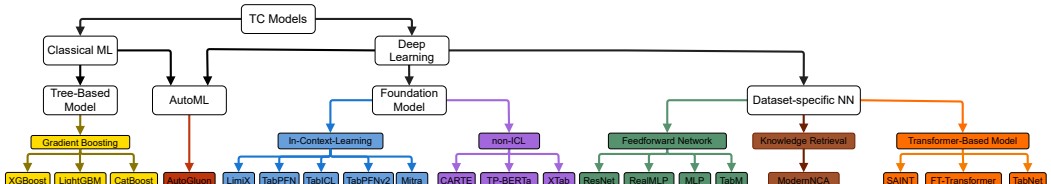

Figure 1: Taxonomy tree of algorithms applied to tabular classification (TC) models

than interpretability. However, recent works exist that propose interpretable counterparts for the top-performing deep learning methods in our study (Kadra et al., 2024; Mueller et al., 2024).

Moreover, to fully unlock a model's potential, contrary to prior work, after HPO we refit all models on the joined training and validation set. Hence, our study provides a fair investigation of post-hyperparameter optimization. We argue that this is a crucial oversight because training on the combined dataset can further improve a model's predictive performance and change the ranking of the models, as indicated by our empirical results. Our findings highlight a paradigm shift, where Deep Learning methods achieve state-of-the-art results and manage to outperform classical approaches.

In summary, we provide the following main insights:

- Meta-learned foundation models and simple feed-forward neural networks outperform GB-DTs in all dataset regimes.
- Feed-forward neural networks outperform related dataset-specific architectures, and ICL foundation models outperform non-ICL ones.
- Refitting on the validation dataset after performing hyperparameter optimization significantly improves predictive quality and affects the overall model rankings.
- As an overarching contribution, to facilitate future research, we open-source our code and we release a large benchmark that includes 20 baselines run on 68 classification datasets, repeated 10 times with different test outer folds, for up to 100 HPO trials, yielding a total number of 8 million evaluations or a total of 11.81 GPU years.

## 2 RELATED WORK

Given the prevalence of tabular data in numerous areas, including healthcare, finance, psychology, and anomaly detection, as highlighted in various studies (Chandola et al., 2009; Johnson et al., 2016; Guo et al., 2017; Ulmer et al., 2020; Urban & Gates, 2021; Nureni & Adekola, 2022; Van Breugel & Van Der Schaar, 2024), there has been significant research dedicated to developing algorithms that effectively address the challenges inherent in these domains.

**Classical Machine Learning.** Gradient Boosted Decision Trees (GBDTs) (Friedman, 2001), including popular implementations like XGBoost (Chen & Guestrin, 2016), LightGBM (Ke et al., 2017), and CatBoost (Prokhorenkova et al., 2018), are widely favored by practitioners for their robust performance on tabular datasets and their short training times.

**Deep Learning.** In terms of neural networks, prior work shows that meticulously searching for the optimal combination of regularization techniques in simple multilayer perceptrons (MLPs) called *Regularization Cocktails* (Kadra et al., 2021) can yield impressive results. Additionally, the models in (Kadra et al., 2021; Gorishniy et al., 2021) propose adaptations of the ResNet (He et al., 2016) architecture for tabular data, demonstrating the potential of deep learning approaches in handling tabular data.

Furthermore, recent research underscores that numerical embeddings (Gorishniy et al., 2022) for tabular data are underexplored. Incorporating these embeddings into neural network architectures, including MLPs and transformer-based models, can substantially enhance performance. Moreover, novel approaches such as RealMLP (Holzmüller et al., 2024) introduce various enhancements to the standard MLP architecture. These include using robust scaling at the pre-processing stage and experimenting with alternative numerical embedding strategies. Lastly, recent research (Gorishniy et al., 2025) achieves state-of-the-art performance by combining simple

| | Protocol | | | Model families | | | | | | | |
|---|---|---|---|---|---|---|---|---|---|---|---|
| **Study** | **Refitting** | **Model-based HPO** | **# Datasets** | **GBDT** | **AutoML** | **ICL** | **nICL** | **FNN** | **KR** | **TF** | **# Baselines** |
| Erickson et al. (2025) | ✗ | ✗ | 51 | ✓ | ✓ | ✓ | ✗ | ✓ | ✓ | ✓ | 18 |
| Ye et al. (2025) | ✗ | ✓ | 300 | ✓ | ✗ | ✓ | ✗ | ✓ | ✓ | ✓ | 31 |
| Rubachev et al. (2024) | ✗ | ✓ | 8 | ✓ | ✗ | ✗ | ✗ | ✓ | ✓ | ✓ | 14 |
| McElfresh et al. (2023) | ✗ | ✗ | 176 | ✓ | ✗ | ✓ | ✗ | ✓ | ✗ | ✓ | 19 |
| Borisov et al. (2022) | ✗ | ✓ | 5 | ✓ | ✗ | ✗ | ✗ | ✓ | ✗ | ✓ | 20 |
| Grinsztajn et al. (2022) | ✗ | ✗ | 45 | ✓ | ✗ | ✗ | ✗ | ✓ | ✗ | ✓ | 7 |
| Shwartz-Ziv & Armon (2022) | ✗ | ✓ | 11 | ✓ | ✗ | ✗ | ✗ | ✗ | ✗ | ✓ | 5 |
| Gorishniy et al. (2021) | ✗ | ✓ | 11 | ✓ | ✗ | ✗ | ✗ | ✓ | ✗ | ✓ | 11 |
| **Ours** | ✓ | ✓ | 68 | ✓ | ✓ | ✓ | ✓ | ✓ | ✓ | ✓ | 20 |

Table 1: Comparison with prior survey works. In our study, we include 6 model families: Gradient Boosted Decision Trees (GBDT), AutoML, In-Context Learning (ICL), non-ICL (nICL), Feed forward neural networks (FNN), Knowledge Retrieval (KR), and Transformer-based Models (TF).

feed-forward neural networks with efficient ensembling techniques effectively mimicking gradient boosted decision trees.

Reflecting their success in various domains, transformers have also garnered attention in the tabular data domain. TabNet (Arik & Pfister, 2021), an innovative model in this area, employs attention mechanisms sequentially to prioritize the most significant features. SAINT (Somepalli et al., 2021) draws inspiration from the seminal transformer architecture (Vaswani et al., 2017). It addresses data challenges by applying attention both to rows and columns. They also offer a self-supervised pre-training phase, particularly beneficial when labels are scarce. The FT-Transformer (Gorishniy et al., 2021) stands out with its two-component structure: the Feature Tokenizer and the Transformer. The Feature Tokenizer is responsible for converting numerical and categorical features into embeddings. These embeddings are then fed into the Transformer, forming the basis for subsequent processing.

Alongside transformer-based approaches, retrieval-augmented models have recently emerged as another promising direction, e.g., TabR (Gorishniy et al., 2023) and ModernNCA (Ye et al., 2024) incorporate nearest-neighbor information at inference time to improve predictive performance.

Recently, a new avenue of research has emerged, focusing on the use of foundation models for tabular data. XTab (Zhu et al., 2023) utilizes shared Transformer blocks, similar to those in FT-Transformer (Gorishniy et al., 2021), followed by fine-tuning dataset-specific encoders. Another notable work, TabPFN (Hollmann et al., 2023), employs in-context learning (ICL), by leveraging sequences of labeled examples provided in the input for predictions, thereby eliminating the need for additional parameter updates after training. The most recent version, TabPFNv2 (Hollmann et al., 2025), addresses the limitations of the first version, handling tables with up to 10K samples, and incorporating row- and column-wise attention, improving predictive performance. TabICL (Qu et al., 2025), similar to the TabPFN models, is pretrained on millions of synthetic datasets and can scale to tables with up to 500K samples. Furthermore, Mitra (Zhang et al., 2025b) demonstrates that pre-training on a mixture of synthetic priors yields stronger performance than relying on a single prior. LimiX (Zhang et al., 2025a) further expands the scope of tabular foundation models, supporting classification, regression, missing-value imputation, data generation, and even sample selection for interpretability. TP-BERTa (Yan et al., 2024), a pre-trained language model for tabular data prediction, uses relative magnitude tokenization to convert scalar numerical features into discrete tokens. The last layer of the model is then fine-tuned on a per-dataset basis. In contrast, CARTE (Kim et al., 2024) utilizes a graph representation of tabular data and a neural network capable of capturing the context within a table. The model is then fine-tuned on a per-dataset basis.

**Empirical Studies.** Significant research has delved into understanding the contexts where neural networks (NNs) excel, and where they fall short (Shwartz-Ziv & Armon, 2022; Borisov et al., 2022; Grinsztajn et al., 2022; Rubachev et al., 2024; Ye et al., 2025). The recent study by McElfresh et al. (2023) is highly related to ours in terms of research focus. However, the authors use only random search for tuning the hyperparameters of neural networks, whereas we employ Tree-structured Parzen Estimator (TPE) (Bergstra et al., 2011) as employed by Gorishniy et al. (2021), which provides a more guided and efficient search strategy. Additionally, recent studies (McElfresh et al., 2023) are limited to evaluating a maximum of 30 hyperparameter configurations, in contrast to our more extensive exploration of up to 100 configurations. Furthermore, despite using the validation set for hyperparameter optimization (HPO), they do not retrain the model on the combined training and validation data using the best-found configuration before evaluating the model on the test set.

Our paper differs from prior studies by applying a methodologically correct experimental protocol involving thorough HPO for neural networks. Recently, TabArena (Erickson et al., 2025) has been proposed as a living benchmark with the goal of continuous maintenance. However, in their study they exclude non-ICL foundation models, rely on random search for HPO, and do not apply refitting opposed to our evaluation protocol. Lastly, Table 1 summarizes the model families evaluated in related empirical studies and highlights the differences in the evaluation protocol. To the best of our knowledge, we are the first to provide a thorough assessment of foundation models and AutoML to other learning paradigms.

## 3 EXPERIMENTAL PROTOCOL

In our study, we focus on binary and multi-class classification problems on tabular data. The general learning task is described in Section 3.1. A detailed description of our evaluation protocol is provided in Section 3.2.

### 3.1 LEARNING WITH TABULAR DATA

A tabular dataset contains $N$ samples with $d$ features defining an $N \times d$ table. A sample $x_i \in \mathbb{R}^d$ is defined by its $d$ feature values. The features can be continuous numerical values or categorical, where for the latter, a common heuristic is to transform the values into numerical space. Given labels $y_i \in \mathcal{Y}$ being associated with the instances (rows) in the table, the task in our study is to solve a binary or multi-class classification problem. Hence, given a tabular dataset $\mathcal{D} = \{(x_i, y_i)\}_{i=1}^{N}$, the aim is to learn a prediction model $f(\cdot)$ to minimize a classification loss function $\ell(\cdot, \cdot)$:

$$\arg\min_{\theta} \sum_{(x_i, y_i) \in \mathcal{D}} \ell(y_i, f(x_i; \theta, \lambda)), \tag{1}$$

where we use $f(x_i; \theta, \lambda)$ for denoting the predicted label by a trained model parameterized by the model weights $\theta$ and hyperparameter configuration $\lambda$.

### 3.2 EXPERIMENTAL SETUP

**Datasets.** In our study, we assess all the methods using OpenMLCC18 (Bischl et al., 2021), a well-established tabular benchmark in the community, which comprises 72 diverse datasets[1]. The datasets contain 5 to 3073 features and 500 to 100,000 instances, covering various binary and multi-class problems. The benchmark excludes artificial datasets, subsets or binarizations of larger datasets, and any dataset solvable by a single feature or a simple decision tree. For the full list of datasets used, we kindly refer the reader to Appendix E.

**Preprocessing**. We use a consistent preprocessing pipeline across all methods whenever possible. By default, we apply a quantile transformation using the scikit-learn library (Pedregosa et al., 2011), and categorical features are encoded with an ordinal encoder similar to prior work (Gorishniy et al., 2021). Methods for which we do not apply this preprocessing are those that inherently require a different approach, such as TP-BERTa and CARTE, or those implemented within libraries where modifying the preprocessing pipeline is not trivial. In these cases, we use the preprocessing strategies from the original works. Regarding batch size, we do not tune it in our experiments due to memory constraints. Instead, we determine batch size heuristically, similar to the setup proposed by Chen et al. (2024), based on the number of features in the dataset. While batch sizes may vary across datasets, they remain consistent across methods.

**Evaluation Protocol.** Our evaluation employs a nested cross-validation approach. Initially, we partition the data into 10 folds. Nine of these folds are then used for hyperparameter tuning. Each hyperparameter configuration is evaluated using 9-fold cross-validation. The results from the cross-validation are used to estimate the performance of the model under a specific hyperparameter configuration. For hyperparameter optimization, we utilize Optuna (Akiba et al., 2019), a well-known HPO library with the Tree-structured Parzen Estimator (TPE) (Bergstra et al., 2011) algorithm, the default Optuna HPO method. The optimization is constrained by a budget of either 100 trials or a

---

[1]Due to memory issues encountered with several methods, we exclude four datasets from our analysis.

maximum duration of 23 hours, similar to prior work (Kadra et al., 2021). Upon determining the optimal hyperparameters using Optuna, we train the model on the combined training and validation splits. All experiments are run on NVIDIA RTX2080Ti GPUs with 11 GB of memory. Our evaluation protocol dictates that for every algorithm, up to 68K different models will be evaluated, leading to a total of approximately 900K individual evaluations. As our study encompasses twenty distinct methods, this methodology culminates in a substantial total of 8M evaluations, involving 900K unique models.

A detailed description of our evaluation protocol is provided in Appendix A.1. In our study, we adhere to the official hyperparameter search spaces from the respective papers for tuning every method. **As a sole exception, the early stopping procedure is performed implicitly from the HPO procedure, where the number of training iterations is a hyperparameter similar to prior work (Kadra et al., 2021). We observed that this alternative form of early stopping yields better generalization.** For a detailed description of the hyperparameter search spaces of all methods included in our analysis, we refer the reader to Appendix A.

**Metrics.** Lastly, we report the model's performance as the average Area Under the Receiver Operating Characteristic (ROC-AUC) across 10 outer test folds. Given the prevalence of imbalanced datasets in the OpenMLCC18 benchmark, we employ ROC-AUC as our primary metric, since it offers a more reliable assessment of model performance.

**Code:** For reproducibility, our code is available at: https://anonymous.4open.science/r/TabularStudy-0EE2.

## 4 BASELINES

In our experiments, we compare a range of methods categorized into three distinct groups: Classical Machine Learning Classifiers, Deep Learning Methods, and AutoML frameworks, as shown in Figure 1.

**Classical Machine Learning Classifiers.** First, we consider *XGBoost* (Chen & Guestrin, 2016), a well-established GBDT library that uses asymmetric trees. Moreover, we consider *CatBoost* (Prokhorenkova et al., 2018), a well-known library for GBDT that employs oblivious trees as weak learners and natively handles categorical features with various strategies. Finally, we also include *LightGBM* (Ke et al., 2017), a widely used GBDT framework that grows trees leaf-wise and supports efficient handling of large datasets.

**Deep Learning Methods.** In terms of classical deep learning methods, we include the *ResNet* implementation provided in the work by Gorishniy et al. (2021). Furthermore, we include three recent and competitive variants of the MLP architecture: *i)* an MLP architecture enhanced with numerical embeddings (Gorishniy et al., 2022) to which we refer as MLP, *ii)* RealMLP (Holzmüller et al., 2024), an MLP enhanced with several additions like robust scaling, numerical embeddings, etc, *iii)* TabM (Gorishniy et al., 2025), an efficient ensemble of MLP models. A more recent direction explores retrieval-augmented models for tabular data, which incorporate nearest-neighbour information at prediction time (Gorishniy et al., 2023; Ye et al., 2024). These approaches have shown competitive performance across diverse benchmarks. In our study, we focus on ModernNCA as the representative retrieval-based method for further analysis.

In terms of transformer-based architectures, we consider *TabNet* (Arik & Pfister, 2021), an architecture that employs sequential attention to selectively utilize the most pertinent features at every decision step. Next, we consider *SAINT* (Somepalli et al., 2021), a hybrid deep learning approach tailored for tabular data challenges. SAINT applies attention mechanisms across both rows and columns and integrates an advanced embedding technique. Lastly, we consider *FT-Transformer* (Gorishniy et al., 2021), an adaptation of the Transformer architecture for tabular data. It transforms categorical and numerical features into embeddings, which are then processed through a series of Transformer layers.

**Foundation Models for Tabular Classification.** For **in-context learning**, we consider *TabPFN* (Hollmann et al., 2023), a meta-learned transformer architecture. Next, we consider TabPFNv2 (Hollmann et al., 2025), which alternates attention first across features, then across samples. We also consider TabICL (Qu et al., 2025), pretrained on synthetic datasets similar to the

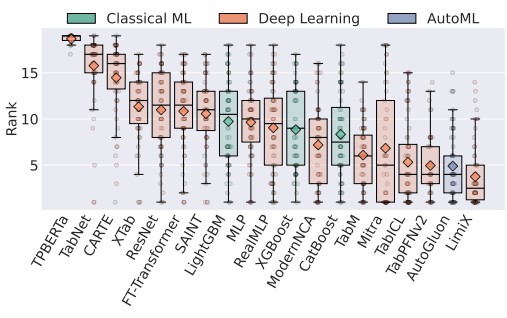
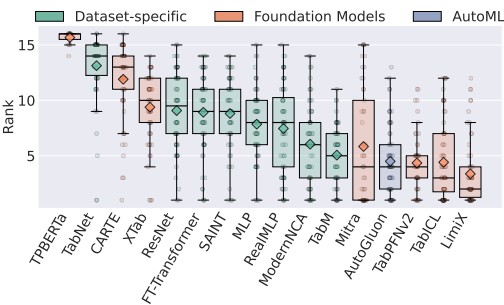

Figure 3: **Left:** Distribution of ranks for the Deep Learning (15 methods), Classical ML (3 methods) and AutoML (1 method) classifier families. **Right:** Distribution of ranks for the Foundation Models (7 methods), Dataset-Specific (8 methods) and AutoML (1 method) classifier families. The boxplots illustrate the rank spread, with medians represented by black lines, diamonds representing the means, and whiskers showing the range.

TabPFN models, which can handle up to 500K samples. Finally, we include the recent Mitra (Zhang et al., 2025b) and LimiX (Zhang et al., 2025a) models as part of our experimental evaluation. Among **non-ICL meta-learned models**, we include *XTab* (Zhu et al., 2023), a method that proposes a cross-table pretraining approach that can work across multiple tables with different column types and structures. Next, we consider *TP-BERTa* (Yan et al., 2024), a variant of the BERT language model that is adapted for tabular prediction. It introduces a relative magnitude tokenization to transform continuous numerical values into discrete high-dimensional tokens. Lastly, we include *CARTE* (Kim et al., 2024) in our experimental study. *CARTE* utilizes a graph representation of tabular data to process tables with differing structures. Since all the non-ICL models were pretrained on real-world datasets, we ensure that there are no datasets that overlap with the OpenMLCC18 benchmark.

**AutoML Frameworks.** Due to the large number of AutoML frameworks available in the community (Feurer et al., 2015; Erickson et al., 2020; LeDell & Poirier, 2020; Feurer et al., 2022), it was infeasible to include all of them in our experimental study. Therefore, we selected AutoGluon (Erickson et al., 2020), a framework that achieves the highest predictive performance in the recent AutoML Benchmark study (Gijsbers et al., 2024).

For all methods, we use their official implementations. We refer the readers to Appendix A for more details.

## 5 EXPERIMENTS AND RESULTS

**Research Question 1: Do DL models outperform gradient boosting methods in tabular data classification?** To address our research question, we initially compare the performance of Deep Learning methods and Classical ML methods jointly, by ranking the methods per-dataset and analyzing the rank distribution (the lower the rank, the better). The results provided in Figure 3 (Left) indicate that DL methods outperform the previous state-of-the-art GBDTs approaches. The best performing methods are LimiX with a median rank of 2.5, followed by AutoGluon and TabPFNv2, both with a median rank of 4. In terms of non-meta learned methods, TabM achieves a median rank of 6, followed by CatBoost, ModernNCA, and XGBoost, with median ranks of 7.5, 8, and 9, respectively.

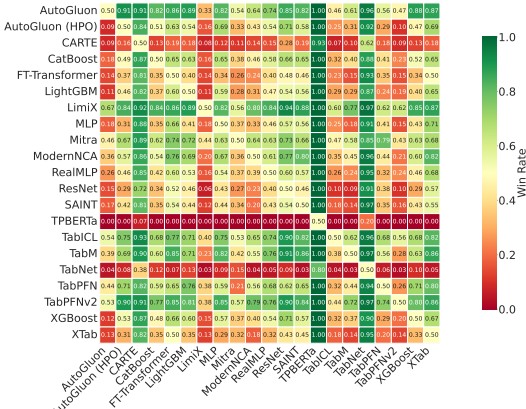

Figure 2: Win-rate dueling matrix comparing learning methods across shared datasets. Each cell (row $i$, column $j$) shows the fraction of common datasets on which method $i$ outperforms $j$.

Next, we investigate how the models compare in a one-versus-one setting to eliminate the effect of related baselines. We present the results in Figure 2, where we observe that the one-versus-

one results are consistent with the results where all the methods are considered jointly. In terms of meta-learned architectures, all ICL models outperform tree-based architectures in the majority of the datasets. From the non-meta learned models, only TabM and ModernNCA manage to outperform all variants of tree-based architectures.

Lastly, we investigate whether there exists a certain region where deep learning methods are superior compared to the tree-based baselines, or where the opposite holds. For that purpose, we highlight the winning method family in Figure 4 for every dataset, over the number of examples and number of features of a dataset. The results show that Deep Learning methods dominate tree-based methods in datasets that have less than 5000 examples, by winning 31-3. In cases where a dataset has more than 5000 examples, tree-based methods become more competitive. However, they are still outperformed by deep learning methods 17-7.

For additional analyses, such as evaluations of predictive performance across different data regimes, examinations of the cost–efficiency trade-off, and investigations into the influence of meta-features on predictive performance, we refer readers to Appendices F, H, and I.

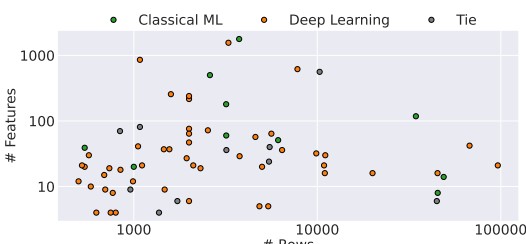

Figure 4: Dataset landscape showing winning method families across different dataset sizes. Each point represents a dataset, positioned by number of rows (x-axis) and features (y-axis) on log scales. Colors indicate which method family achieved the highest accuracy: Deep Learning methods (orange), Classical ML tree-based models (green), and ties (gray).

**Research Question 2: Do meta-learned NNs outperform data-specific NNs in tabular data classification?** To answer the second research question, we analyze the distribution of ranks between the two subfamilies within the Deep Learning category: foundation models and dataset-specific neural networks. Figure 3 (Right) plot illustrates that in-context learning models are very competitive, with LimiX having the best overall rank, followed by TabICL and TabPFNv2. Within the dataset-specific family, TabM demonstrates the best performance, attaining a median rank of 5 across all 68 datasets, followed closely by ModernNCA with a median rank of 6. The non-ICL foundation models XTab, CARTE, and TP-BERTa obtain the worst performance compared to all remaining dataset-specific neural networks, except for TabNet. Interestingly to note is that, except for the in-context learning models, which are meta-learned architectures, the feed-forward neural networks TabM, ModernNCA, RealMLP, and MLP outperform the attention-based architectures.

Next, we compare foundation models with dataset-specific NNs by generating critical difference diagrams. To generate the CD diagrams, we utilize the `autorank` package (Herbold, 2020), which performs a Friedman test followed by a Nemenyi post-hoc test at a significance level of 0.05.

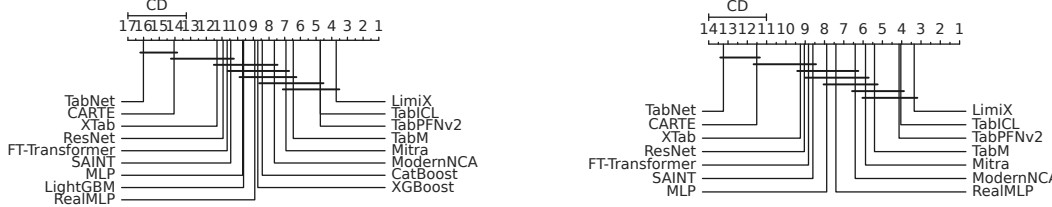

Figure 5: Critical difference (CD) diagram of the methods, where a horizontal bar indicates the absence of statistical significance. **Left:** CD diagram of Deep Learning vs. GBDTs, **Right:** CD diagram of dataset-specific vs. foundation models.

We present the results in Figure 5, where the black bars connecting methods indicate that there is no statistically significant difference in performance. Due to the limited number of datasets shared among the methods, TP-BERTa was excluded from this comparison. The left diagram of Figure 5 illustrates that LimiX, TabICL and TabPFNv2 outperform all the other methods, demonstrating superior performance. TabM trails the top methods by a narrow margin, ranking fourth overall. Mitra

and ModernNCA follow, with Mitra's performance not significantly different from that of the top three methods. With the exception of the other top 4 methods, LimiX, outperforms every other method significantly, including CatBoost and XGBoost. We also compare the performance of models within the deep learning family. The right plot of Figure 5 shows a critical difference diagram indicating that LimiX, TabICL and TabPFNv2 again attain the top average ranks. Except for TabM and Mitra, LimiX, TabICL and TabPFNv2 manage to significantly outperform all the remaining methods.

A comprehensive presentation of the raw results for all methods, both after hyperparameter optimization (HPO) and with default hyperparameter configurations, is provided in Appendix D.

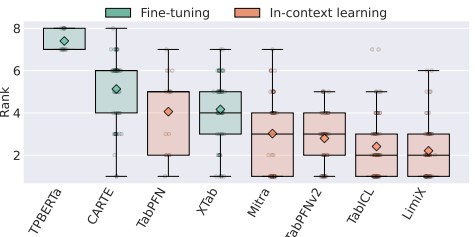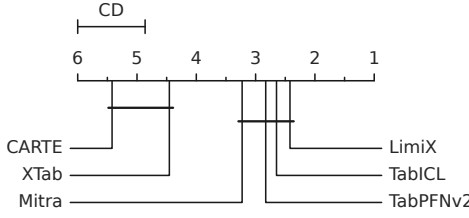

Figure 6: **Left:** Distribution of ranks for the non-ICL (3 models) and In-context learning (5 models) classifier families. The boxplots illustrate the rank spread, with medians represented by black lines and whiskers showing the range. **Right:** CD diagram of non In-Context Learning models against In-context learning models. The horizontal bar indicates the absence of statistically significant differences.

**Research Question 3: Which paradigm in transfer learning performs better: Do in-context models or non in-context models perform better?**   To further investigate the family of foundation models, whether non-ICL or in-context learning models yield better performance, we conducted an analysis similar to our previous research questions. Figure 6 (Left) illustrates that both LimiX and TabICL, achieve a median rank of 2, followed by TabPFNv2 and Mitra both achieving a median rank of 3 with TabPFNv2s mean being slightly better. Among the non-ICL methods, XTab showed the best performance with a median rank of 4, followed by TabPFN, CARTE and TP-BERTa. We hypothesize that this discrepancy in performance primarily stems from the fact that the non-ICL methods all require dataset-specific fine-tuning and do not generalize to new tables without gradient updates, which is the key way in which they differ from ICL models. Additionally, CARTE was designed to exploit informative column names, whereas such semantically meaningful headers are largely absent in standard benchmarks, XTab instantiates a relatively small FT-Transformer backbone that operates in a purely, supervised fashion, which may restrict its ability to exploit dataset-level structure, and for TP-BERTa, the particular scheme used to encode numerical values may further constrain how well it can represent quantitative information.

To investigate whether the differences in performance are statistically significant, we present a CD diagram in Figure 6 (Right), from which TP-BERTa and TabPFN are excluded due to the limited number of common datasets among the methods. The CD diagram reveals that the in-context learning methods, LimiX, TabICL, TabPFNv2 and Mitra, significantly outperform the non-ICL methods XTab and CARTE.

**Research Question 4: Does refitting after performing hyperparameter optimization have a significant impact on the predictive quality of the models, and does it impact the overall model ranking?**   To investigate the impact of refitting, we select two distinct methods from the Deep Learning family, namely FT-Transformer as a transformer-based architecture, and TabM as an MLP-based architecture, while also selecting CatBoost and XGBoost from the tree-based family as the top-performing models.

Initially, we compare the models in isolation to investigate how refitting affects the distribution of predictive performances across tasks. We present the results in Figure 7 (Left), where, as observed, all the methods that incorporate refitting feature a lower rank and outperform their non-refitting

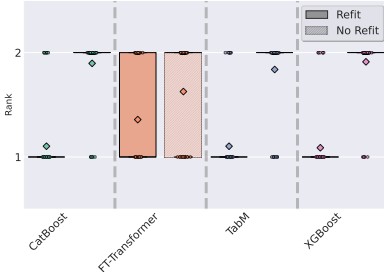 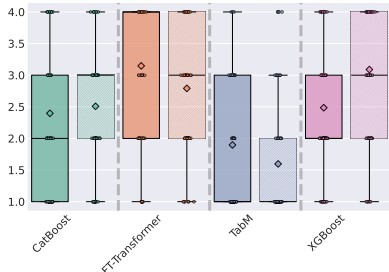

Figure 7: Refitting impact on the predictive performance. **Left:** Investigating the rank distribution of the methods in isolation, with and without refitting. **Right:** Investigating the distribution of the ranks for the methods jointly, with and without refitting.

counterparts. Additionally, as we show in Appendix C, the difference in results is statistically significant in the majority of cases.

Moreover, we investigate whether refitting affects the ranking of the methods when considered jointly. The results in Figure 7 (Right) indicate that refitting does change the ranking of the methods, where, e.g., after refitting, XGBoost manages to outperform FT-Transformer and achieves a better median rank compared to the non-refitting counterpart. We continue by evaluating the impact of refitting via head-to-head, dataset-level comparisons, and applying statistical tests to assess the significance of the observed differences. We kindly refer the reader to Appendix C for details.

**Research Question 5: What is the influence of hyperparameter optimization on a method's predictive performance?** To investigate the impact of hyperparameter optimization, we calculate the per-dataset rank of every method with and without performing hyperparameter optimization and compare the median rank improvement for every method compared to the other baselines. In general, the majority of methods improve in predictive performance when considered in isolation, as validated in Appendix B.15. However, the results in Figure 8 indicate that only TabM, RealMLP, SAINT, XGBoost, XTab and Modern-NCA improve in terms of median rank compared to the other contenders, when hyperparameter optimization is performed.

In Appendix B, we provide a detailed analysis of hyperparameter importance, showing both the overall contribution of hyperparameters to model performance and the individual effect of each hyperparameter for every method.

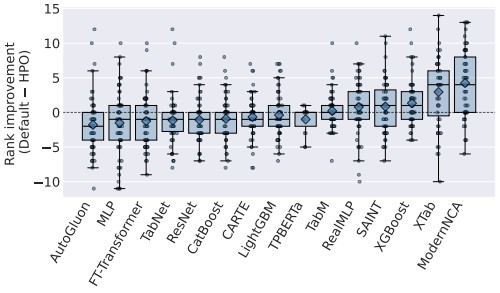

Figure 8: Rank improvement of methods with hyperparameter optimization compared to using the default configuration. Rankings are computed relative to other methods on each dataset (default rank – HPO rank; positive values indicate improvement). Box plots show the distribution across datasets, with points for individual datasets, horizontal bar for the median and diamonds for the mean.

## 6 CONCLUSION

Our comprehensive empirical study evaluates the quality of seventeen state-of-the-art tabular classification approaches across 68 diverse classification datasets from the OpenMLCC18 benchmark with a rigorous setup, employing cross-validation, model-based hyperparameter optimization, and refitting. Our results indicate a paradigm shift, where deep learning methods outperform traditional baselines in all dataset regimes of the considered benchmark. Next to a fair comparison of model families, we provide an in-depth analysis of the importance of refitting, the influence of hyperparameter optimization on the models' performance, the most important hyperparameters per method, and the cost-performance efficiency of various methods. Our study contributes valuable insights into the current landscape of tabular data modeling and encourages further potential research directions with promising model families.

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

# A  EVALUATION PROTOCOL AND CONFIGURATION SPACES

## A.1  EVALUATION PROTOCOL

---

**Algorithm 1:** Nested Cross-Validation for Hyperparameter Optimization

**Input** : Dataset $D$, Number of outer folds $K = 10$, Number of inner folds $J = 9$, Number
of hyperparameter optimization trials $T = 100$, Search space $\Lambda$

**Output:** Overall performance $\bar{P}_{\text{outer}}$

1  **for** $k \leftarrow 1$ **to** $K$ **do**
2     Split $D$ into training set $D_{\text{train}}^k$ and test set $D_{\text{test}}^k$;
3     **for** $t \leftarrow 1$ **to** $T$ **do**
4        Sample hyperparameter configuration $\theta_t$ from the search space $\Lambda$;
5        **for** $j \leftarrow 1$ **to** $J$ **do**
6           Split $D_{\text{train}}^k$ into inner training set $D_{\text{train}}^{k,j}$ and validation set $D_{\text{val}}^{k,j}$;
7           Train model $M(\lambda_t)$ on $D_{\text{train}}^{k,j}$;
8           Evaluate $M(\lambda_t)$ on $D_{\text{val}}^{k,j}$ to get performance $P^{k,j}(\lambda_t)$;
9        **end**
10       Compute mean performance $\bar{P}^k(\lambda_t) = \frac{1}{J}\sum_{j=1}^{J} P^{k,j}(\lambda_t)$;
11       Use $\bar{P}^k(\lambda_t)$ as the objective value for $\lambda_t$;
12    **end**
13    Select the best hyperparameter configuration $\lambda_k^*$ ;
14    Train final model $M(\lambda_k^*)$ on $D_{\text{train}}^k$;
15    Evaluate $M(\lambda_k^*)$ on $D_{\text{test}}^k$ to get outer performance $P_{\text{outer}}^k$;
16 **end**
17 Compute overall performance $\bar{P}_{\text{outer}} = \frac{1}{K}\sum_{k=1}^{K} P_{\text{outer}}^k$;
18 **return** $\bar{P}_{\text{outer}}$;

---

Algorithm 1, shows the nested-cross validation with the outer folds (lines 1-16) and inner folds (lines 5-9). In each trial (lines 3-12), the mean performance across inner folds are calculated in line 10 which is used as the objective value for Optuna in line 11. After the maximal number of trials $T$ is reached or the time budget is exceeded, we select the best hyperparameter setting in line 13.

## A.2  CATBOOST

Table 2: Search space for CatBoost.

| Parameter | Type | Range | Log Scale |
|---|---|---|---|
| max_depth | Integer | [3, 10] | |
| learning_rate | Float | $[10^{-5}, 1]$ | ✓ |
| bagging_temperature | Float | [0, 1] | |
| l2_leaf_reg | Float | [1, 10] | ✓ |
| leaf_estimation_iterations | Integer | [1, 10] | |
| iterations | Integer | [100, 2000] | |

The specific search space employed for CatBoost is detailed in Table 2. Our implementation heavily relies on the framework provided by the official implementation of the FT-Transformer, as found in the following repository[2]. We do this to ensure a consistent pipeline across all methods, that we compare. The CatBoost algorithm implementation, however, is the official one[3].

---

[2]https://github.com/yandex-research/rtdl-revisiting-models
[3]https://catboost.ai/

For the default configuration of CatBoost, we do not modify any hyperparameter values. This approach allows the library to automatically apply its default settings, ensuring that our implementation is aligned with the most typical usage scenarios of the library.

## A.3 XGBOOST

Table 3: Search space for XGBoost.

| Parameter | Type | Range | Log Scale |
|---|---|---|---|
| max_depth | Integer | [3, 10] | |
| min_child_weight | Float | $[10^{-8}, 10^5]$ | ✓ |
| subsample | Float | [0.5, 1] | |
| learning_rate | Float | $[10^{-5}, 1]$ | ✓ |
| colsample_bylevel | Float | [0.5, 1] | |
| colsample_bytree | Float | [0.5, 1] | |
| gamma | Float | $[10^{-8}, 10^2]$ | ✓ |
| reg_lambda | Float | $[10^{-8}, 10^2]$ | ✓ |
| reg_alpha | Float | $[10^{-8}, 10^2]$ | ✓ |
| n_estimators | Integer | [100, 2000] | |

We utilized the official XGBoost implementation[4]. While the data preprocessing steps were consistent across all methods, a notable exception was made for XGBoost. For this method, we implemented one-hot encoding on categorical features, as XGBoost does not inherently process categorical values, in line with the implementation from the FT-Transformer repository.

The comprehensive search space for the XGBoost hyperparameters is detailed in Table 3. In the case of default hyperparameters, our approach mirrored the CatBoost implementation where we opted not to set any hyperparameters explicitly but instead, use the library defaults.

Furthermore, it is important to note that XGBoost lacks native support for the ROC-AUC metric in multiclass problems. To address this, we incorporated a custom ROC-AUC evaluation function. This function first applies a softmax to the predictions and then employs the ROC-AUC scoring functionality provided by scikit-learn, which can be found at the following link[5].

## A.4 LIGHTGBM

Table 4: Search space for LightGBM.

| Parameter | Type | Range | Log Scale |
|---|---|---|---|
| feature_fraction | Float | [0.5, 1.0] | |
| lambda_l2 | Float | {0.0, [0.1, 11.0]} | ✓ |
| learning_rate | Float | [0.001, 1.0] | ✓ |
| num_leaves | Integer | [4, 768] | |
| min_sum_hessian_in_leaf | float | [0.0001, 100] | ✓ |
| bagging_fractions | Float | [0.5, 1.0] | |
| bagging_fractions | Float | [0.5, 1.0] | |
| n_estimators | Integer | [100, 2000] | |

---

[4]https://xgboost.readthedocs.io/en/stable/

[5]https://scikit-learn.org/stable/modules/generated/sklearn.metrics.roc_auc_score.html

The hyperparameter search space for LightGBM is shown in Table 4. As with other methods, we adopt the preprocessing pipeline from the FT-Transformer repository.

For the default configuration, we retain all library-defined hyperparameters without modification.

## A.5 FT-TRANSFORMER

Table 5: Search space for FT-Transformer.

| Parameter | Type | Range | Log Scale |
|---|---|---|---|
| n_layers | Integer | [1, 6] | |
| d_token | Integer | [64, 512] | |
| residual_dropout | Float | [0, 0.2] | |
| attn_dropout | Float | [0, 0.5] | |
| ffn_dropout | Float | [0, 0.5] | |
| d_ffn_factor | Float | $[\frac{2}{3}, \frac{8}{3}]$ | |
| lr | Float | $[10^{-5}, 10^{-3}]$ | ✓ |
| weight_decay | Float | $[10^{-6}, 10^{-3}]$ | ✓ |
| epochs | Integer | [10, 500] | |

In our investigation, we adopted the official implementation of the FT-Transformer (Gorishniy et al., 2021). Diverging from the approach from the original study, we implemented a uniform search space applicable to all datasets, rather than customizing the search space for each specific dataset. This approach ensures a consistent and comparable application across various datasets. The uniform search space we employed aligns with the structure proposed in Gorishniy et al. (2021). Specifically, we consolidated the search space by integrating the upper bounds defined in the original paper with the minimum bounds identified across different datasets.

Regarding the default hyperparameters, we adhered strictly to the specifications provided in Gorishniy et al. (2021).

## A.6 SAINT

We utilize the official implementation of the method as detailed by the respective authors (Somepalli et al., 2021). The comprehensive search space employed for hyperparameter tuning is illustrated in Table 6.

Regarding the default hyperparameters, we adhere to the specifications provided by the authors in their original implementation.

Table 6: Search space for SAINT.

| Parameter | Type | Range | Log Scale |
|---|---|---|---|
| embedding_size | Categorical | {4, 8, 16, 32} | |
| transformer_depth | Integer | [1, 4] | |
| attention_dropout | Float | [0, 1.0] | |
| ff_dropout | Float | [0, 1.0] | |
| lr | Float | $[10^{-5}, 10^{-3}]$ | ✓ |
| weight_decay | Float | $[10^{-6}, 10^{-3}]$ | ✓ |
| epochs | Integer | [10, 500] | |

## A.7 TABNET

Table 7: Search space for TabNet.

| Parameter | Type | Range | Log Scale |
|---|---|---|---|
| n_a | Integer | [8, 64] | |
| n_d | Integer | [8, 64] | |
| gamma | Float | [1.0, 2.0] | |
| n_steps | Integer | [3, 10] | |
| cat_emb_dim | Integer | [1, 3] | |
| n_independent | Integer | [1, 5] | |
| n_shared | Integer | [1, 5] | |
| momentum | Float | [0.001, 0.4] | ✓ |
| mask_type | Categorical | {entmax, sparsemax} | |
| epochs | Integer | [10, 500] | |

For TabNet's implementation, we utilized a well-maintained and publicly available version, accessible at the following link[6]. The hyperparameter tuning search space for TabNet, detailed in Table 7, was derived from McElfresh et al. (2023).

Regarding the default hyperparameters, we followed the recommendations provided by the original authors.

## A.8 RESNET

Table 8: Search space for ResNet.

| Parameter | Type | Range | Log Scale |
|---|---|---|---|
| layer_size | Integer | [64, 1024] | |
| lr | Float | $[10^{-5}, 10^{-2}]$ | ✓ |
| weight_decay | Float | $[10^{-6}, 10^{-3}]$ | ✓ |
| residual_dropout | Float | [0, 0.5] | |
| hidden_dropout | Float | [0, 0.5] | |
| n_layers | Integer | [1, 8] | |
| d_embedding | Integer | [64, 512] | |
| d_hidden_factor | Float | [1.0, 4.0] | |
| epochs | Integer | [10, 500] | |

We employed the ResNet implementation as described in prior work (Gorishniy et al., 2021). The entire range of hyperparameters explored for ResNet tuning is detailed in Table 8. Since the original study did not specify default hyperparameter values, we relied on the search space provided in a prior work (Kadra et al., 2021).

## A.9 MLP-PLR

We employ the MLP implementation proposed by (Gorishniy et al., 2022). The search space used for hyperparameter optimization is detailed in Table 9. Default hyperparameters are adapted from (McElfresh et al., 2023), while the search space is based on the original work of (Gorishniy et al., 2022).

---

[6]https://github.com/dreamquark-ai/tabnet

Table 9: Search space for MLP-PLR.

| Parameter | Type | Range | Log Scale |
|---|---|---|---|
| lr | Float | $[10^{-5}, 10^{-3}]$ | ✓ |
| weight_decay | Float | $[10^{-6}, 10^{-3}]$ | ✓ |
| dropout | Float | $[0, 0.5]$ | |
| n_layers | Integer | $[1, 16]$ | |
| d_embedding | Integer | $[64, 512]$ | |
| d_num_embedding | Integer | $[1, 128]$ | |
| d_first | Integer | $[1, 1024]$ | |
| d_middle | Integer | $[1, 1024]$ | |
| d_last | Integer | $[1, 1024]$ | |
| n | Integer | $[1, 128]$ | |
| sigma | Float | $[0.01, 100]$ | ✓ |
| epochs | Integer | $[10, 500]$ | |

## A.10 TABM

To run TabM in our experiments, we use the `pytabkit` implementation[7]. The hyperparameter search space for TabM, presented in Table 10, is adapted from the original work (Gorishniy et al., 2025).

Table 10: Search space for TabM.

| Parameter | Type | Range | Log Scale |
|---|---|---|---|
| n_blocks | Integer | $[1, 5]$ | |
| d_block | Integer | $[64, 1024]$ | |
| dropout | Float | $[0, 0.5]$ | |
| hidden_dropout | Float | $[0, 0.5]$ | |
| lr | Float | $[10^{-4}, 5 \times 10^{-3}]$ | ✓ |
| weight_decay | Float | $[10^{-4}, 10^{-1}]$ | ✓ |
| epochs | Integer | $[10, 500]$ | |

## A.11 REALMLP

For our RealMLP experiments, we use the official implementation in `pytabkit`[7]. Following the authors' recommendations, we impute missing values using the mean of the training split before applying their preprocessing pipeline. We adopt the recommended default hyperparameters and search space, detailed in Table 11. Additionally, we extend the search space for initializing the standard deviation of the first embedding layer and tune the embedding dimensions, as suggested by the authors.

---

[7]`https://github.com/dholzmueller/pytabkit`

Table 11: Search space for RealMLP.

| Parameter | Type | Range | Log Scale |
|---|---|---|---|
| num_emb_type | Categorical | {None, PBLD, PL, PLR} | |
| add_front_scale | Categorical | {True, False} | |
| lr | Float | [2e-2, 3e-1] | ✓ |
| p_drop | Categorical | {0.0, 0.15, 0.3} | |
| act | Categorical | {relu, selu, mish} | |
| hidden_sizes | Categorical | {[256, 256, 256], [64, 64, 64, 64, 64], [512]} | |
| wd | Categorical | {0.0, 0.02} | |
| plr_sigma | Float | [0.05, 1e1] | ✓ |
| ls_eps | Categorical | {0.0, 0.1} | |
| embedding_size | Integer | [1, 64] | |
| n_epochs | Integer | [10, 500] | |

## A.12 MODERNNCA

To run ModernNCA in our experiments we use the `TALENT` implementation[8]. As suggested by the authors, we adopt the same search space, as well as their default hyperparameters for our default experiments detailed in Table 12.

Table 12: Search space for ModernNCA.

| Parameter | Type | Range | Log Scale |
|---|---|---|---|
| dropout | Float | [0.0, 0.5] | |
| d_block | Integer | [64, 1024] | |
| dim | Integer | [64, 1024] | |
| n_blocks | Integer | [0, 2] | |
| n_frequencies | Integer | [16, 96] | |
| frequency_scale | Float | [0.005, 10.0] | ✓ |
| lr | Float | [1e-5, 0.1] | ✓ |
| wd | Categorical | [0.0, [1e-6, 1e-3]] | ✓ |
| sample_rate | Float | [0.05, 0.6] | |
| d_embedding | Integer | [16, 64] | |
| n_epochs | Integer | [10, 500] | |

## A.13 XTAB

For XTab, we utilize the official implementation[9]. To ensure comparability with other methods, we decouple XTab from AutoGluon and apply the same preprocessing and training pipeline as used for the other models. The original work reports results for both light finetuning and heavy finetuning, so we introduce this as a categorical hyperparameter. If `light_finetuning` is set to `True`, the model is finetuned for only 3 epochs. Otherwise, we follow the same epoch range as for the other methods, i.e., $[10, 500]$. Furthermore, we use the checkpoint after 2000 iterations (`iter_2k.ckpt`), provided by the authors. Table 13 outlines the complete search space used for XTab during hyperparameter optimization.

---

[8]`https://github.com/LAMDA-Tabular/TALENT`
[9]`https://github.com/BingzhaoZhu/XTab`

Table 13: Search space for XTab.

| Parameter | Type | Range | Log Scale |
|---|---|---|---|
| lr | Float | $[10^{-5}, 10^{-3}]$ | ✓ |
| weight_decay | Float | $[10^{-6}, 10^{-3}]$ | ✓ |
| light_finetuning | Categorical | {True, False} | |
| epochs | Integer | 3 (if light_finetuning=True) or $[10, 500]$ (otherwise) | |

## A.14  CARTE

For CARTE, we use the official implementation[10]. Similar to XTab, since it is a pretrained model, we do not tune the architectural hyperparameters but keep them fixed and load the checkpoint provided by the authors. The search space used for CARTE during our hyperparameter optimization (HPO) process is shown in Table 14.

Table 14: Search space for CARTE.

| Parameter | Type | Range | Log Scale |
|---|---|---|---|
| lr | Float | $[10^{-5}, 10^{-3}]$ | ✓ |
| weight_decay | Float | $[10^{-6}, 10^{-3}]$ | ✓ |
| epochs | Integer | $[10, 500]$ | |

## A.15  TP-BERTA

We use the official implementation for TP-BERTa[11]. Similar to the other pretrained models, we only tune the `learning rate`, `weight decay`, and the number of finetuning `epochs`. The search space is shown in Table 15.

Table 15: Search space for TP-BERTa.

| Parameter | Type | Range | Log Scale |
|---|---|---|---|
| lr | Float | $[10^{-5}, 10^{-3}]$ | ✓ |
| weight_decay | Float | $[10^{-6}, 10^{-3}]$ | ✓ |
| epochs | Integer | $[10, 500]$ | |

## A.16  TABPFN

For TabPFN and TabPFNv2, we utilized the official implementations from the authors[12]. We followed the settings suggested by the authors and we did not preprocess the numerical features as TabPFN does that natively, we ordinally encoded the categorical features and we used an ensemble size of 32 for TabPFN to achieve peak performance as suggested by the authors. For TabPFNv2 we use the default settings and do not change anything.

---

[10]https://github.com/soda-inria/carte
[11]https://github.com/jyansir/tp-berta
[12]https://github.com/automl/TabPFN

### A.17 TABICL

We use the official implementation of TabICL[13] and follow the authors' instructions without modification. In particular, we employ the default checkpoint `tabicl-classifier-v1.1-0506.ckpt`.

### A.18 MITRA

For Mitra, we rely on the official implementation available in the AutoGluon library, using the default configuration and without applying any additional fine-tuning.

### A.19 LIMIX

For LimiX, we use the official implementation[14]. The authors provide two variants—LimiX-2M and LimiX-16M. Following their recommendations, we conduct our experiments using only the LimiX-16M model.

### A.20 AUTOGLUON

For our experiments, we utilize the official implementation of AutoGluon[15](version 1.1.1). Specifically, we evaluate two configurations of AutoGluon: the HPO version and the recommended version.

- For the HPO version, we use the default search spaces for the models included in Auto-Gluon's ensemble.
- For the recommended version, we set `presets="best_quality"` as per the official documentation and do not perform hyperparameter optimization.

---

[13]https://github.com/soda-inria/tabicl
[14]https://github.com/limix-ldm/LimiX
[15]https://auto.gluon.ai/stable/index.html

# B  HYPERPARAMETER ANALYSIS

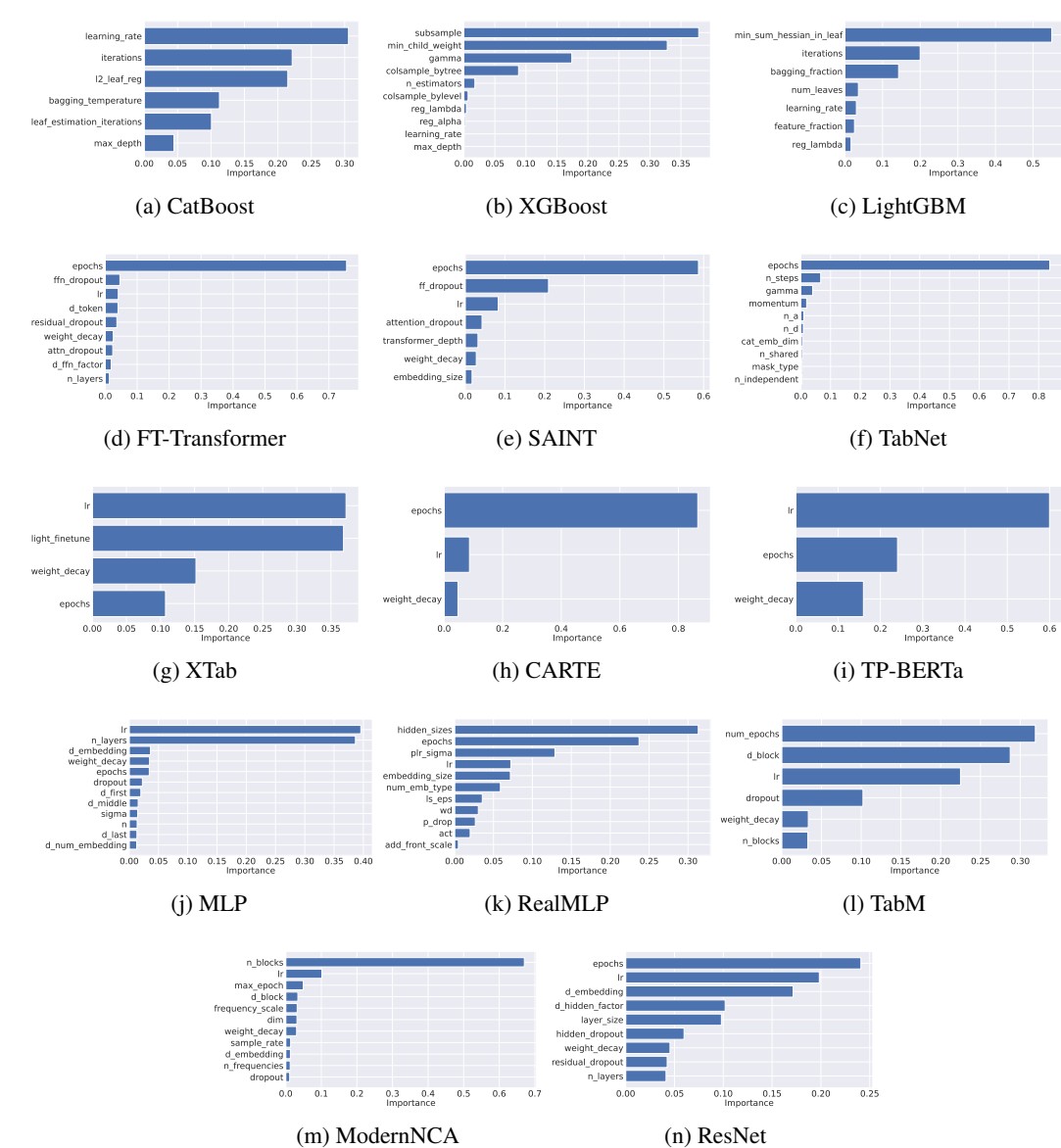

Figure 9: Hyperparameter importance for various methods

In this section, we first examine the overall importance of hyperparameters for each method, as shown in Figure 9, which quantifies the contribution of each hyperparameter to model performance. The subsequent figures in this section illustrate the effect of individual hyperparameters on the performance metric. The x-axis represents the hyperparameters, while the y-axis denotes the ROC-AUC performance. We calculate hyperparameter importance using the fANOVA (Hutter et al., 2014) implementation in Optuna (Akiba et al., 2019). According to our analysis, the most important hyperparameter for CatBoost is the learning rate, while for XGBoost, it is the subsample ratio of the training instances. For XTab, the learning rate is also the most important hyperparameter, closely followed by the light_finetune hyperparameter, which is a categorical parameter taking values True or False. When light_finetune is True, we fine-tune XTab for only 3 epochs; when it is False, we use the same range of epochs as for the other methods (10 to 500). Similarly, for the MLP with PLR embeddings, the learning rate proves to be the most influential hyperparameter, whereas for RealMLP, the number of units in the hidden layers. For ModernNCA the number

of blocks turns out to be the most important hyperparameter. For the remaining dataset-specific neural networks in the deep learning family, as well as for CARTE, the number of training epochs is the most important hyperparameter, indicating that training duration plays a critical role in their performance.

## B.1 CATBOOST

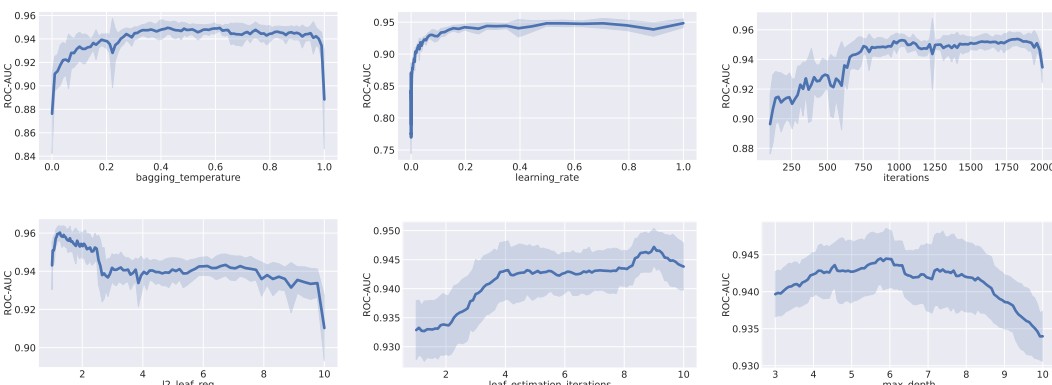

Figure 10: Effect of all the hyperparameters on model performance for CatBoost. The x-axis represents the hyperparameter values, while the y-axis shows the corresponding performance.

## B.2 RESNET

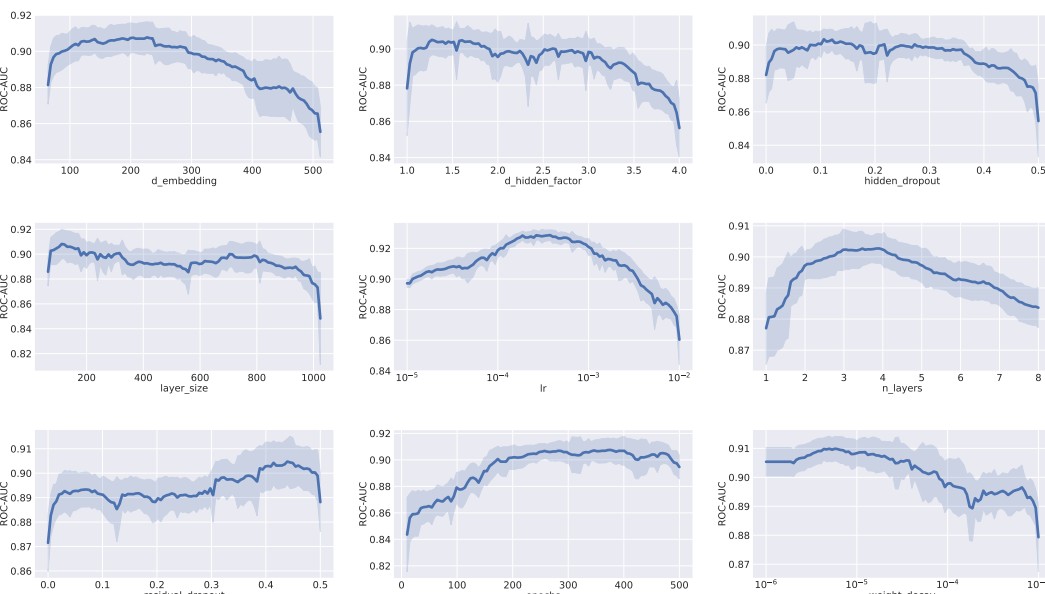

Figure 11: Effect of all the hyperparameters on model performance for ResNet. The x-axis represents the hyperparameter values, while the y-axis shows the corresponding performance.

## B.3 MLP-PLR

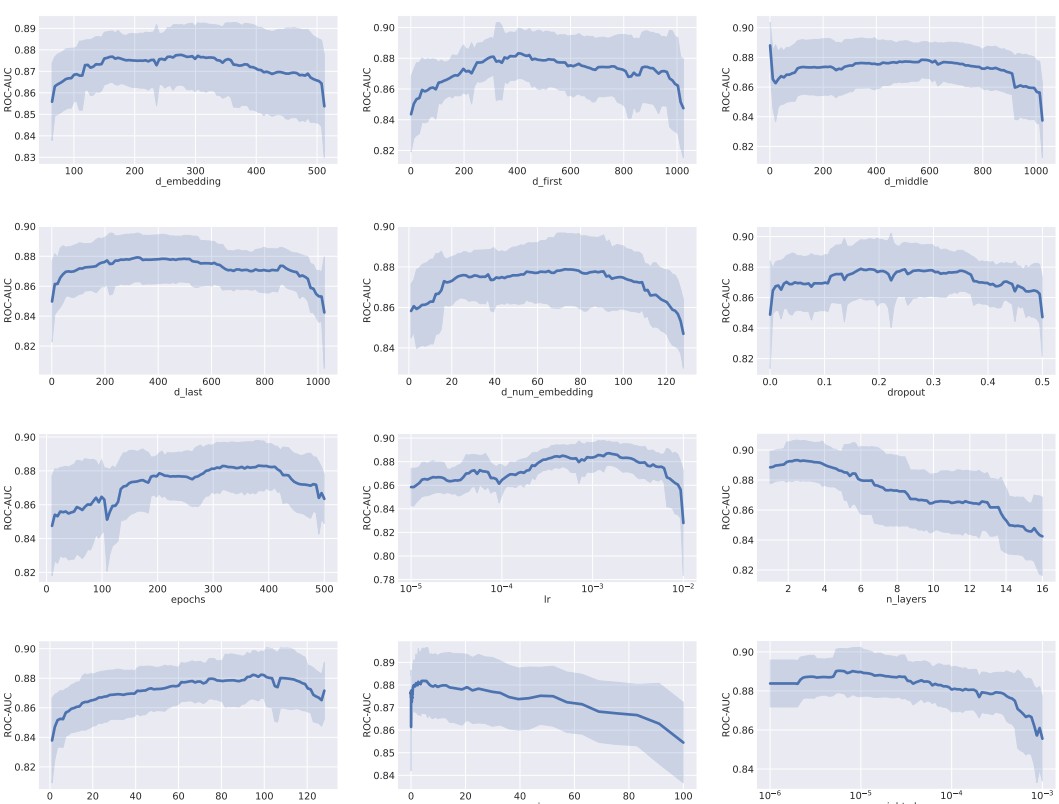

Figure 12: Effect of all the hyperparameters on model performance for MLP-PLR. The x-axis represents the hyperparameter values, while the y-axis shows the corresponding performance.

## B.4 REALMLP

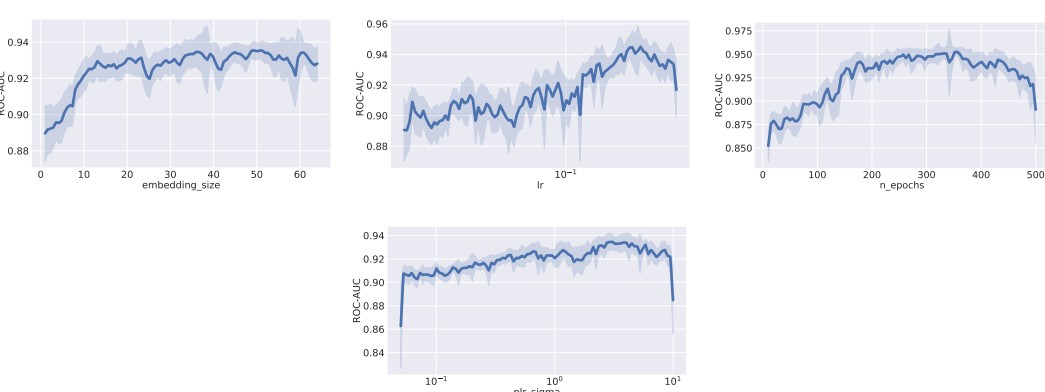

Figure 13: Effect of all the hyperparameters on model performance for RealMLP. The x-axis represents the hyperparameter values, while the y-axis shows the corresponding performance.

Since fANOVA does not support categorical hyperparameters, we exclude them from this analysis.

## B.5  TABM

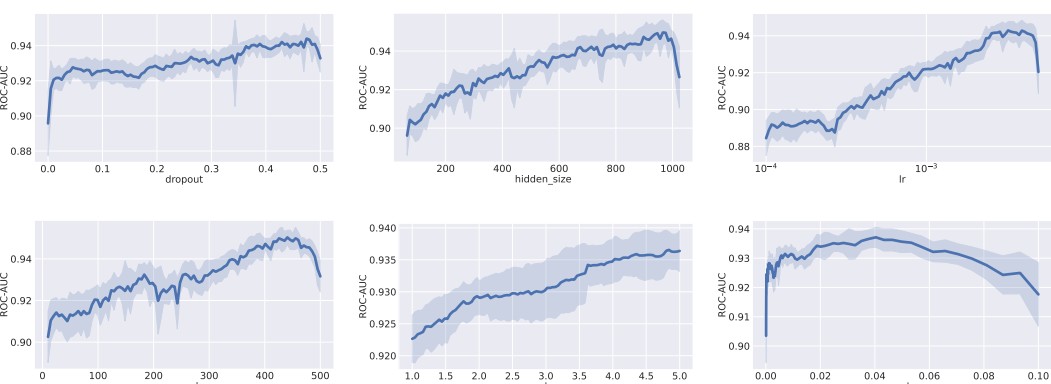

Figure 14: Effect of all the hyperparameters on model performance for TabM. The x-axis represents the hyperparameter values, while the y-axis shows the corresponding performance.

## B.6  XGBOOST

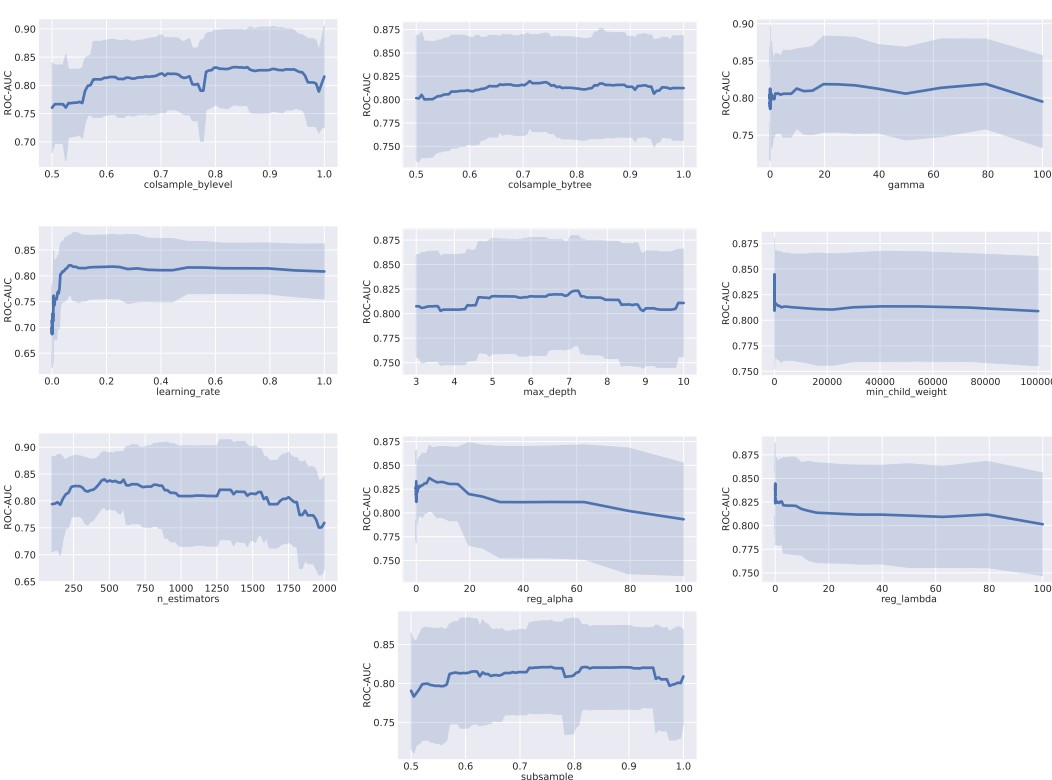

Figure 15: Effect of all the hyperparameters on model performance for XGBoost. The x-axis represents the hyperparameter values, while the y-axis shows the corresponding performance.

## B.7 LightGBM

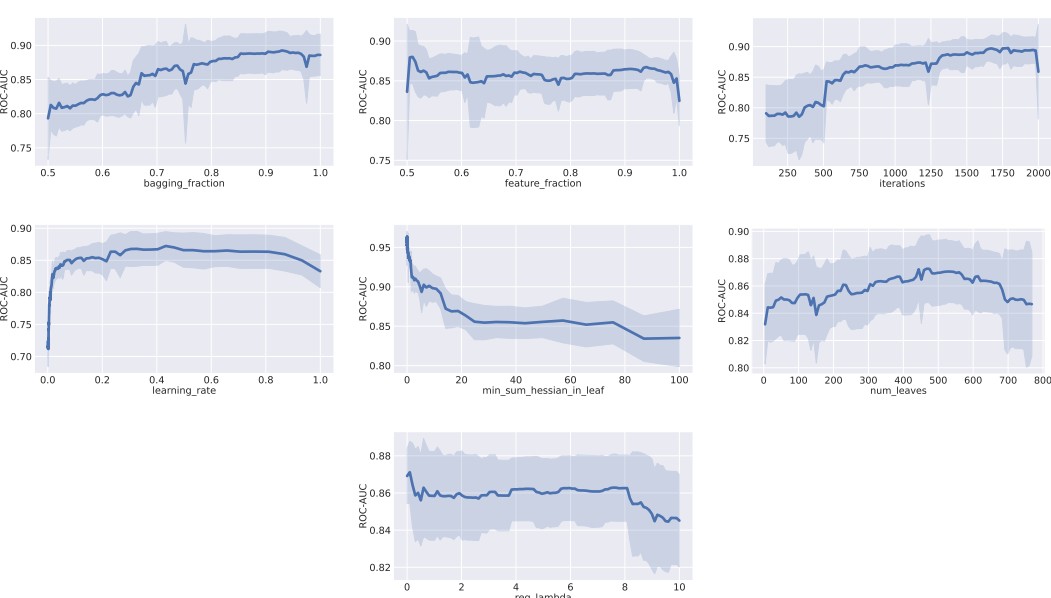

Figure 16: Effect of all the hyperparameters on model performance for LightGBM. The x-axis represents the hyperparameter values, while the y-axis shows the corresponding performance.

## B.8 FT-Transformer

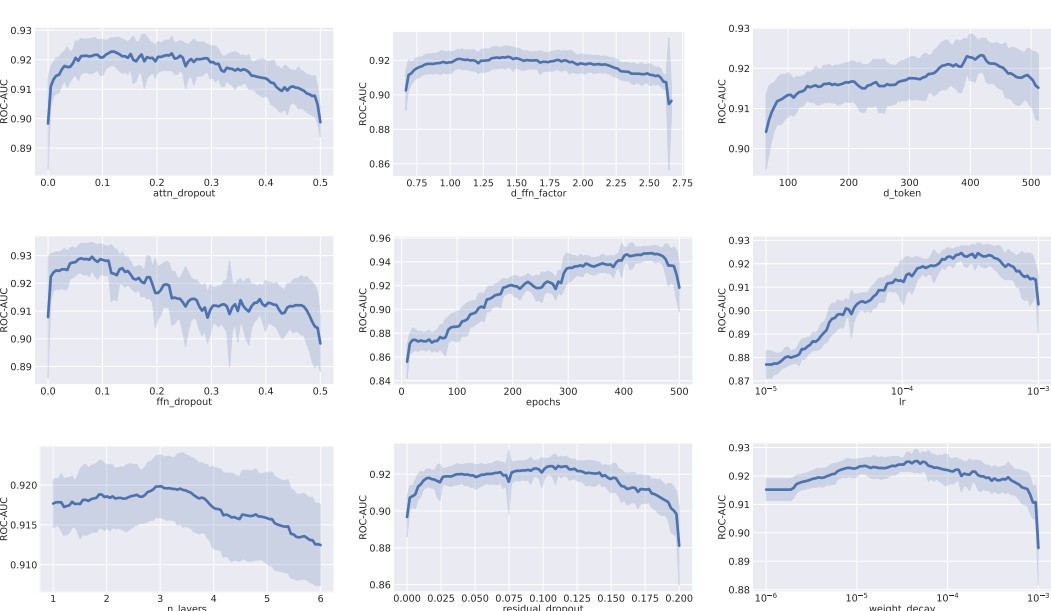

Figure 17: Effect of all the hyperparameters on model performance for FT-Transformer. The x-axis represents the hyperparameter values, while the y-axis shows the corresponding performance.

## B.9   SAINT

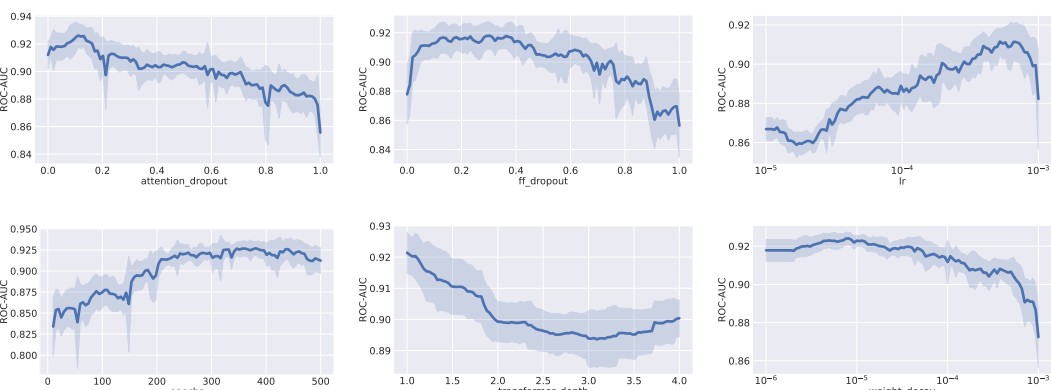

Figure 18: Effect of all the hyperparameters on model performance for SAINT. The x-axis represents the hyperparameter values, while the y-axis shows the corresponding performance.

## B.10   ModernNCA

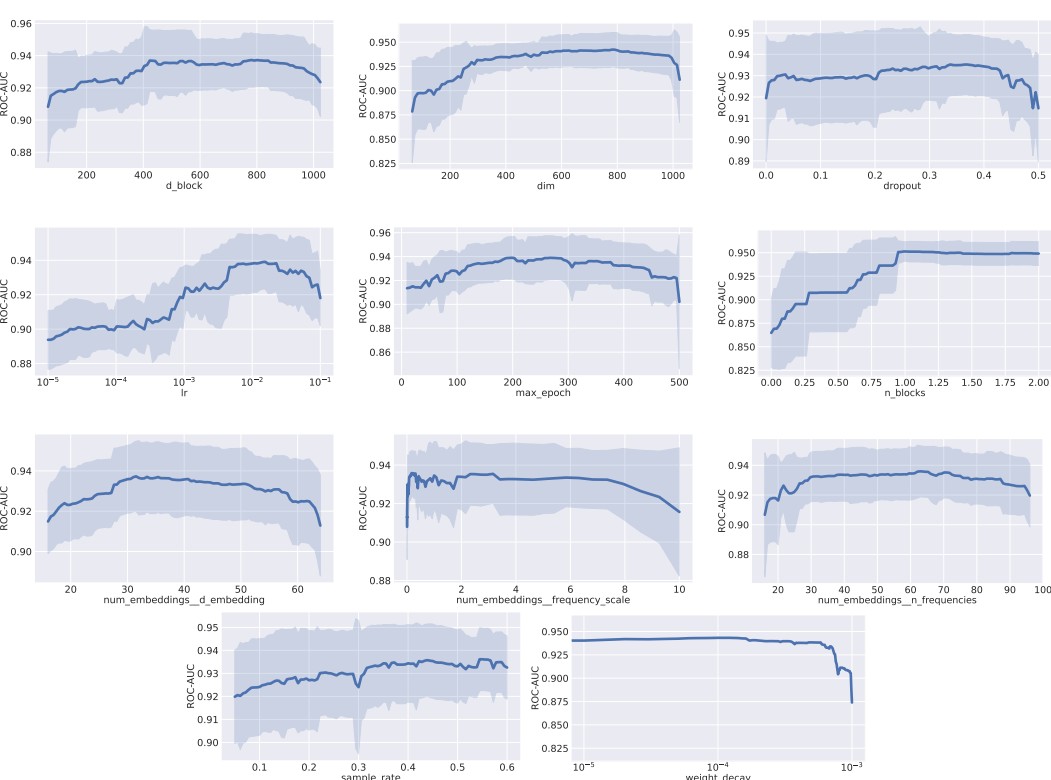

Figure 19: Effect of all the hyperparameters on model performance for ModernNCA. The x-axis represents the hyperparameter values, while the y-axis shows the corresponding performance.

## B.11 TABNET

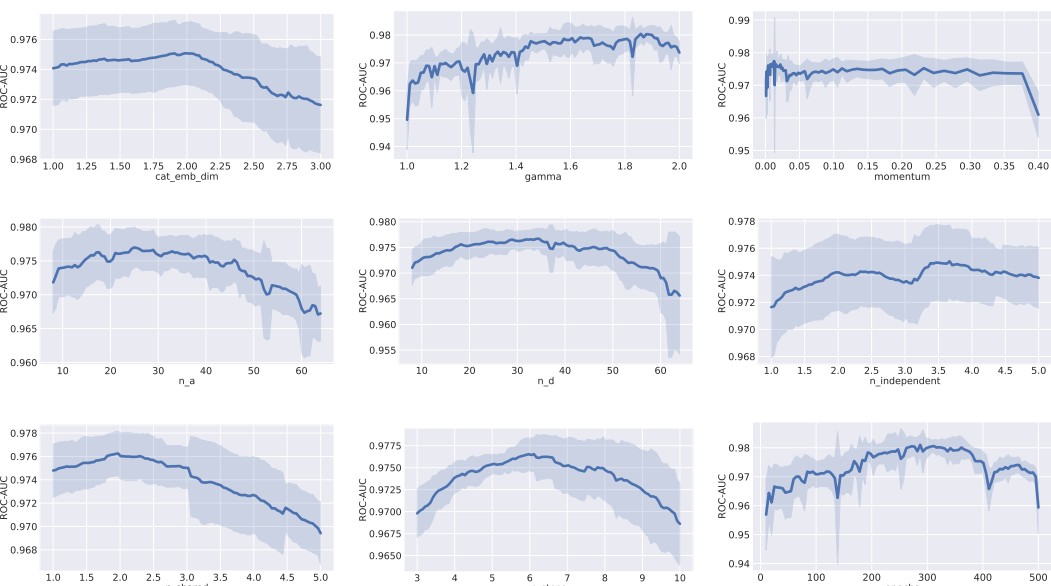

Figure 20: Effect of all the hyperparameters on model performance for TabNet. The x-axis represents the hyperparameter values, while the y-axis shows the corresponding performance.

## B.12 XTAB

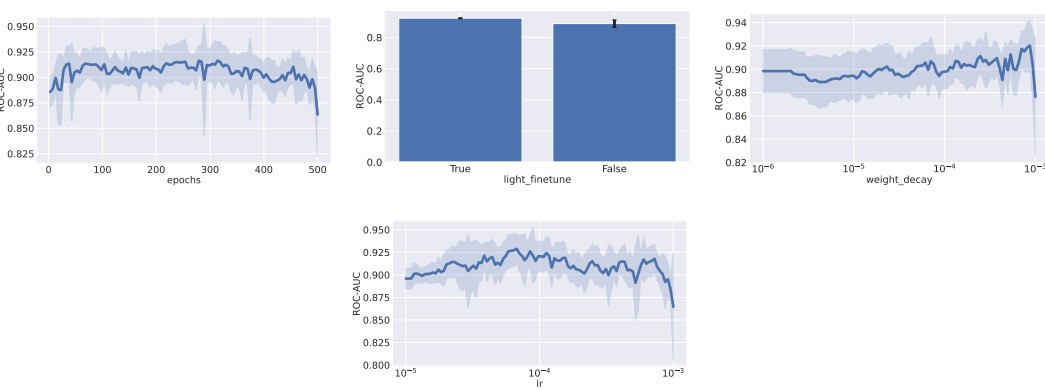

Figure 21: Effect of all the hyperparameters on model performance for XTab. The x-axis represents the hyperparameter values, while the y-axis shows the corresponding performance.

## B.13 CARTE

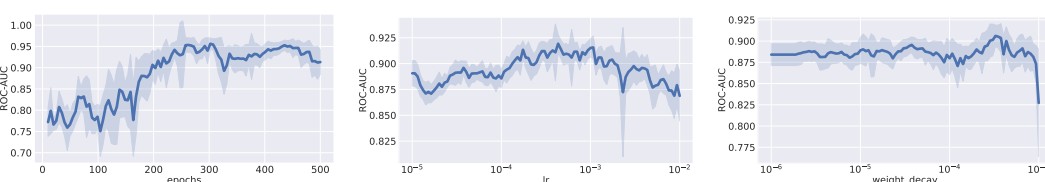

Figure 22: Effect of all the hyperparameters on model performance for CARTE. The x-axis represents the hyperparameter values, while the y-axis shows the corresponding performance.

## B.14 TP-BERTA

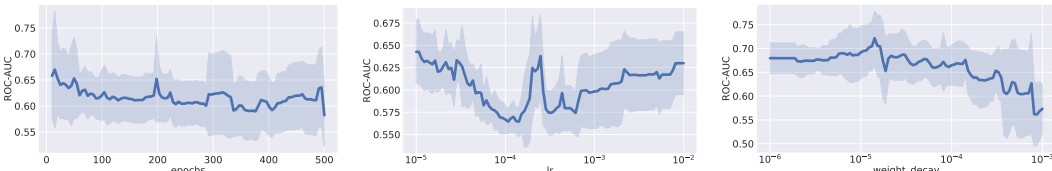

Figure 23: Effect of all the hyperparameters on model performance for TP-BERTa. The x-axis represents the hyperparameter values, while the y-axis shows the corresponding performance.

## B.15 HPO INFLUENCE ON A PER-MODEL LEVEL

In our analysis of hyperparameter optimization (HPO) versus default configurations across various machine learning methods, we observed that HPO generally led to improved performance. The analysis is depicted in Figure 24. This improvement is reflected in the average rank reductions for most methods when HPO was applied. For example, XGBoost's and ModernNCA's average rank improved significantly from 1.94 in their default configurations to 1.06 with HPO, and XTab showed a similar enhancement, moving from a rank of 1.96 down to 1.04.

These findings are visually represented in the accompanying plot, which illustrates the performance gains achieved through HPO. An exception to the general trend was observed with TP-BERTa, where the default configuration slightly outperformed the HPO version (average ranks of 1.47 and 1.53, respectively). This anomaly can be attributed to the computational demands of TP-BERTa. Due to its large model size, TP-BERTa was unable to complete the full 100 hyperparameter tuning trials within the allotted 23-hour time frame, often finishing only a few trials. Consequently, the HPO process may have converged to a suboptimal configuration that did not surpass the performance of the default settings.

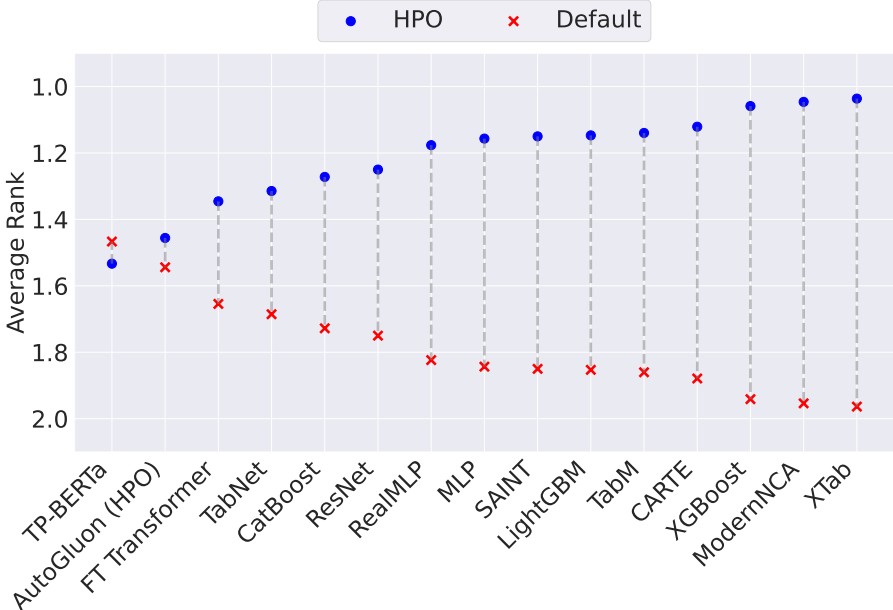

Figure 24: Comparison of average rank performance between hyperparameter-optimized (HPO) models and default models. The blue dots represent the performance of the HPO models, while the red crosses denote the default models. Lower ranks indicate better performance.

## C   ABLATING THE CHOICE OF REFITTING

In this ablation, we explore whether refitting the model on the combined training and validation sets (after hyperparameter optimization) provides any measurable benefit. The standard procedure, as described in Section 3.2, uses a 10-fold nested cross-validation: we split the data into 10 folds, use 9 folds for inner cross-validation and HPO, then identify the best hyperparameter configuration and refit the model on all 9 folds before testing on the remaining fold.

We compare this approach to a no-refitting variant. Here, we still employ 10-fold cross-validation, but replace the inner cross-validation with a single 70/30 split of the 9 folds for training and validation. We train the model on the 70% partition, perform HPO on the 30% partition, and then save both the optimal hyperparameter configuration and the resulting trained model. Hence, when moving to the test fold, we simply load this trained model (with its fixed hyperparameters) instead of retraining on the entire 9-fold set. We repeat this for each of the 10 folds, ensuring the test set remains identical across both approaches.

Due to the computational resources required, we restrict this analysis to four methods: CatBoost, XGBoost, FT-Transformer, and TabM. The results of this ablation study are presented below, comparing performance with and without refitting.

Section 5, Research Question 4 (Figure 7) summarizes the main ablation results. Here, we extend that analysis by comparing the refitting and no-refitting variants head-to-head at the dataset level and conducting statistical tests to assess significance.

Figure 25 illustrates the performance difference between CatBoost with and without refitting across all datasets. The results clearly indicate that, with only a few exceptions, the refitted version consistently outperforms its non-refitted counterpart.

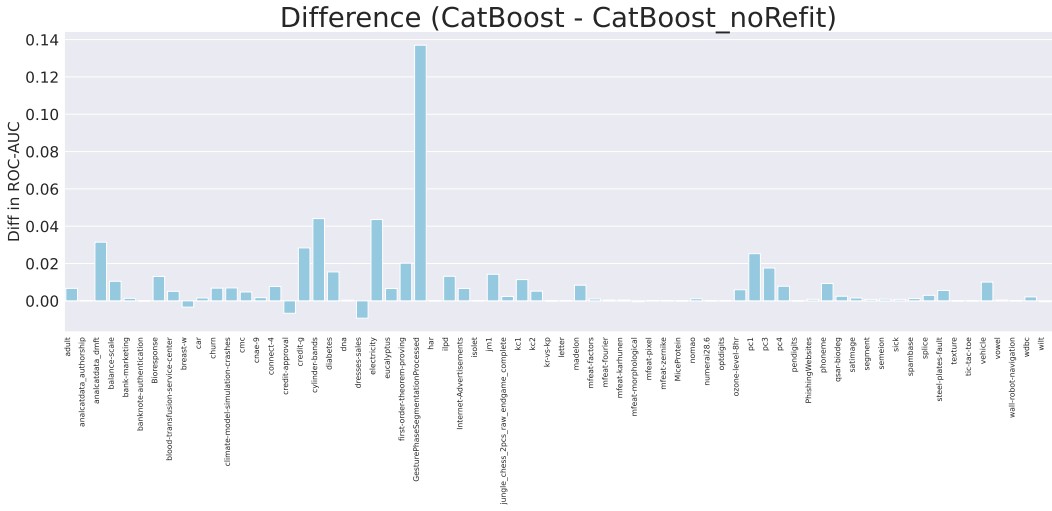

Figure 25: Performance difference between CatBoost with refitting and CatBoost without refitting across Datasets. Positive values indicate an improvement in ROC-AUC when refitting is applied, while negative values indicate a performance drop.

A similar pattern is observed in Figure 26 for XGBoost. Likewise, Figure 27 shows that refitting yields superior performance on most datasets for TabM too. In contrast, Figure 28 reveals that a greater number of datasets favor the non-refitted FT-Transformer. Nonetheless, the majority still benefit from refitting overall.

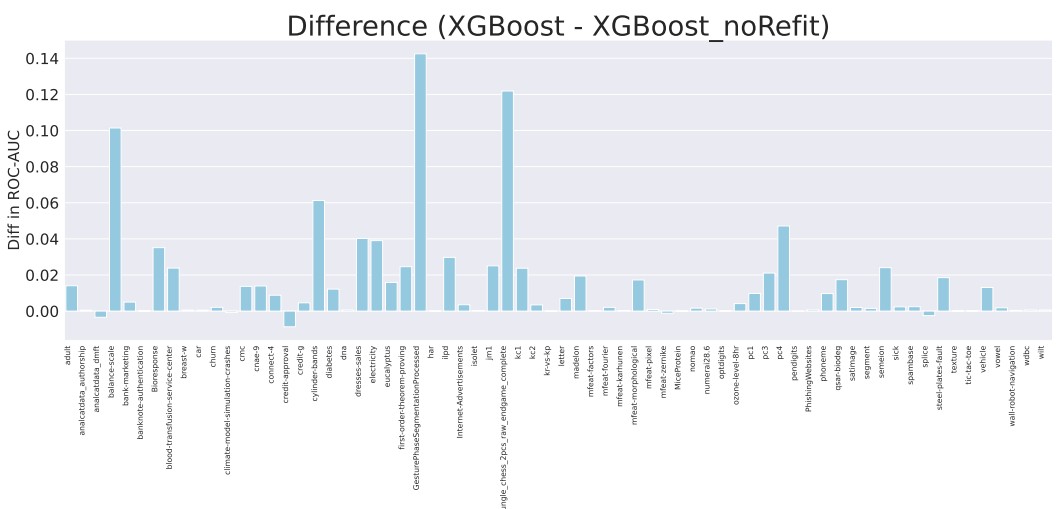

Figure 26: Performance difference between XGBoost with refitting and XGBoost without refitting across Datasets. Positive values indicate an improvement in ROC-AUC when refitting is applied, while negative values indicate a performance drop.

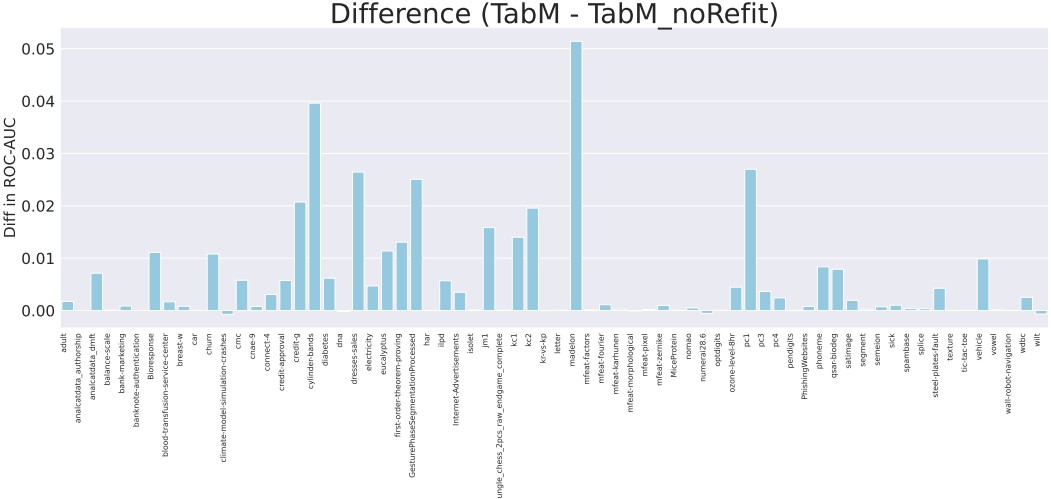

Figure 27: Performance difference between TabM with refitting and TabM without refitting across Datasets. Positive values indicate an improvement in ROC-AUC when refitting is applied, while negative values indicate a performance drop.

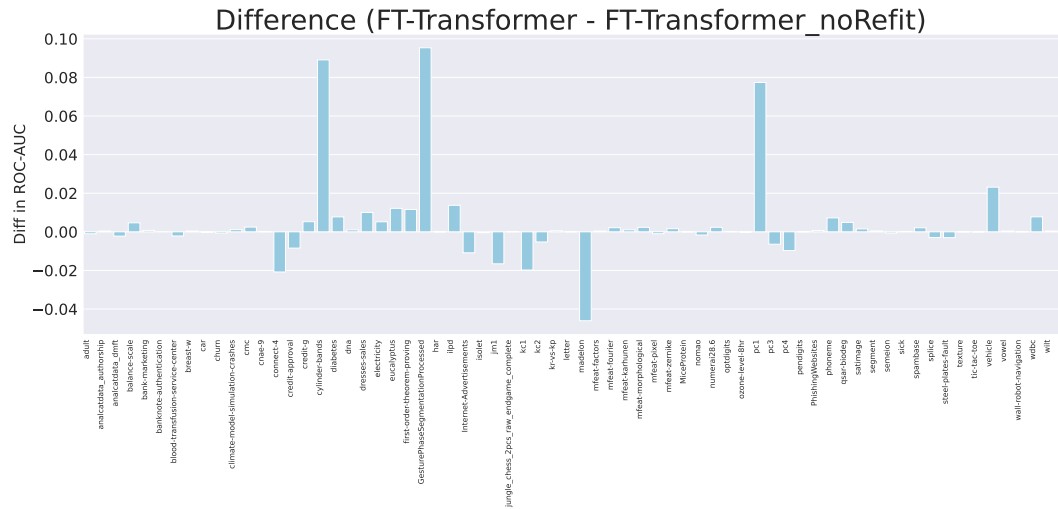

Figure 28: Performance difference between FT-Transformer with refitting and FT-Transformer without refitting Across Datasets. Positive values indicate an improvement in ROC-AUC when refitting is applied, while negative values indicate a performance drop.

Furthermore, we conducted a Wilcoxon signed-rank test to compare the performance of refitting versus no-refitting across multiple datasets for each method. The statistical results are summarized in Table 16.

Table 16: Statistical Comparison of Refit vs. No-Refit Methods

| Method Pair | #Datasets | Avg. Diff | Median Diff | Wilcoxon Stat | p-value |
|---|---|---|---|---|---|
| CatBoost vs. CatBoost_noRefit | 68 | 0.0079 | 0.0016 | 180.0000 | 1.298511e-09 |
| XGBoost vs. XGBoost_noRefit | 68 | 0.0144 | 0.0029 | 152.0000 | 4.413043e-10 |
| TabM vs. TabM_noRefit | 68 | 0.0056 | 0.0010 | 101.0000 | 3.395174e-10 |
| FT vs. FT_noRefit | 68 | 0.0035 | 0.0004 | 843.0000 | 9.356765e-02 |

For CatBoost, we observed an average performance improvement of $0.0079$ when refitting, with a median difference of $0.0016$. The Wilcoxon test yielded a test statistic of $180.0000$ and a highly significant p-value of $1.2985 \cdot 10^{-9}$. This strongly suggests that refitting leads to a statistically significant and consistent improvement in CatBoost's performance across datasets. Given the very low p-value ($p < 0.001$), we can confidently reject the null hypothesis that refitting has no effect.

For XGBoost, the average improvement with refitting was $0.0144$, with a median difference of $0.0029$. The Wilcoxon test statistic was $152.0000$, with a highly significant p-value of $4.4130 \cdot 10^{-10}$. These results indicate that, similar to CatBoost, refitting yields a consistent and statistically significant improvement in performance for XGBoost.

In contrast, for FT-Transformer, the average improvement due to refitting was $0.0035$, with a median difference of $0.0004$. However, the Wilcoxon test yielded a test statistic of $843.0000$ and a p-value of $0.0936$, which is not statistically significant ($p > 0.05$). This suggests that while refitting improves FT-Transformer's performance on average, the improvement is not consistent or significant across datasets.

For TabM, refitting led to an average performance improvement of $0.0056$ with a median difference of $0.0010$. The Wilcoxon test yielded a test statistic of $101.0000$ and a highly significant p-value of $3.3952 \cdot 10^{-10}$. This provides strong evidence that refitting substantially and consistently improves TabM's performance across datasets. Given the extremely low p-value ($p < 0.001$), we can confidently reject the null hypothesis, concluding that refitting has a significant positive impact on TabM.

Additionally, Table 17 presents the raw results of FT-Transformer, CatBoost, XGBoost and TabM, in comparison to their non-refitted counterparts.

Table 17: Average test ROC-AUC per dataset for FT, CatBoost, TabM and XGBoost using refitting vs. no refitting across CV folds.

| Dataset | CatBoost | CatBoost_norefit | XGBoost | XGBoost_norefit | FT | FT_norefit | TabM | TabM_norefit |
|---|---|---|---|---|---|---|---|---|
| adult | **0.930747** | 0.924052 | **0.930482** | 0.916441 | 0.914869 | **0.915875** | **0.919662** | 0.917949 |
| analcatdata_authorship | **0.999662** | 0.999470 | **0.999816** | 0.999304 | **0.999985** | 0.999566 | **1.000000** | **1.000000** |
| analcatdata_dmft | **0.579136** | 0.547691 | 0.572150 | **0.575579** | 0.576947 | **0.579169** | **0.576017** | 0.568896 |
| balance-scale | **0.972625** | 0.962132 | **0.991268** | 0.889868 | **0.999735** | 0.995086 | **0.998912** | 0.998805 |
| bank-marketing | **0.938831** | 0.937464 | **0.938384** | 0.933394 | **0.938198** | 0.937470 | **0.941872** | 0.941008 |
| banknote-authentication | 0.999935 | **0.999979** | **0.999935** | 0.999849 | **1.000000** | **1.000000** | **1.000000** | **1.000000** |
| Bioresponse | **0.885502** | 0.872449 | **0.888615** | 0.853483 | **0.820159** | N/A | **0.876671** | 0.865593 |
| blood-transfusion-service-center | **0.754965** | 0.749848 | **0.750671** | 0.726849 | 0.745975 | **0.748119** | **0.748538** | 0.746875 |
| breast-w | 0.989162 | **0.992507** | **0.992112** | 0.991595 | **0.989503** | 0.989074 | **0.995845** | 0.995032 |
| car | **1.000000** | 0.998453 | **0.999902** | 0.999412 | 0.999751 | **0.999969** | **1.000000** | **1.000000** |
| churn | **0.922968** | 0.916146 | **0.914432** | 0.912343 | 0.914596 | **0.915300** | **0.929636** | 0.918852 |
| climate-model-simulation-crashes | **0.951480** | 0.944551 | 0.947000 | **0.947724** | **0.934671** | 0.933561 | 0.939969 | **0.940694** |
| cmc | **0.740149** | 0.735398 | **0.735649** | 0.721967 | **0.739402** | 0.736959 | **0.743797** | 0.738024 |
| cnae-9 | **0.996316** | 0.994599 | **0.997454** | 0.983546 | **0.994497** | 0.994377 | **0.998100** | 0.997319 |
| connect-4 | **0.921050** | 0.913372 | **0.931952** | 0.923175 | 0.901170 | **0.921978** | **0.941654** | 0.938588 |
| credit-approval | 0.934006 | **0.940661** | 0.934692 | **0.943379** | 0.935798 | **0.944236** | **0.934458** | 0.928719 |
| credit-g | **0.801762** | 0.773381 | **0.798571** | 0.794000 | **0.783048** | 0.777810 | **0.790905** | 0.770190 |
| cylinder-bands | **0.912070** | 0.867995 | **0.928116** | 0.866898 | **0.915494** | 0.826412 | **0.926477** | 0.886880 |
| diabetes | **0.837869** | 0.822365 | **0.835638** | 0.823473 | **0.831108** | 0.823379 | **0.829801** | 0.823632 |
| dna | **0.995028** | 0.994658 | **0.995278** | 0.994620 | 0.990937 | 0.989937 | 0.994505 | **0.994692** |
| dresses-sales | 0.595731 | **0.605008** | **0.622414** | 0.582184 | **0.620033** | 0.610016 | **0.642200** | 0.615764 |
| electricity | **0.980993** | 0.937421 | **0.987790** | 0.948766 | **0.963076** | 0.957884 | **0.968731** | 0.964049 |
| eucalyptus | **0.923334** | 0.916719 | **0.918055** | 0.902200 | **0.923933** | 0.911772 | **0.931897** | 0.920520 |
| first-order-theorem-proving | **0.831775** | 0.811589 | **0.834883** | 0.810288 | **0.796707** | 0.785106 | **0.818255** | 0.805200 |
| GesturePhaseSegmentationProcessed | **0.916674** | 0.779683 | **0.916761** | 0.774266 | **0.895166** | 0.799810 | **0.933828** | 0.908776 |
| har | **0.999941** | 0.999887 | **0.999960** | 0.999919 | 0.999685 | **0.999706** | **0.999966** | 0.999927 |
| ilpd | **0.744702** | 0.731536 | **0.748019** | 0.718251 | **0.751488** | 0.737753 | **0.744875** | 0.739189 |
| Internet-Advertisements | **0.979120** | 0.972513 | **0.982276** | 0.978762 | 0.974513 | **0.985391** | **0.985640** | 0.982167 |
| isolet | **0.999389** | 0.999282 | **0.999488** | 0.999225 | 0.998817 | **0.999282** | **0.999750** | 0.999628 |
| jm1 | **0.756611** | 0.742362 | **0.759652** | 0.734592 | 0.709321 | **0.725904** | **0.751557** | 0.735722 |
| jungle_chess_2pcs_raw_endgame_complete | **0.976349** | 0.973983 | **0.974087** | 0.852209 | **0.999975** | 0.999861 | **0.999985** | 0.999945 |
| kc1 | **0.825443** | 0.814042 | **0.832007** | 0.808270 | 0.783519 | **0.803310** | **0.813763** | 0.799767 |
| kc2 | **0.846802** | 0.841593 | **0.843295** | 0.839860 | 0.832014 | **0.837281** | **0.833491** | 0.813933 |
| kr-vs-kp | 0.999392 | **0.999419** | **0.999796** | 0.999785 | **0.999777** | 0.999173 | 0.999652 | **0.999659** |
| letter | **0.999854** | 0.999802 | **0.999819** | 0.992828 | **0.999919** | 0.999886 | **0.999943** | 0.999906 |
| madelon | **0.937562** | 0.929178 | **0.932249** | 0.912814 | 0.747391 | **0.793476** | **0.809941** | 0.758538 |
| mfeat-factors | **0.998910** | 0.997917 | **0.999004** | 0.998767 | **0.999015** | 0.998560 | **0.999700** | 0.999550 |
| mfeat-fourier | **0.984714** | 0.984229 | **0.983375** | 0.981292 | **0.984511** | 0.982372 | **0.988497** | 0.987389 |
| mfeat-karhunen | **0.999264** | 0.998802 | **0.999211** | 0.998908 | 0.998682 | 0.997649 | **0.999521** | 0.999267 |
| mfeat-morphological | 0.965406 | **0.965867** | **0.963075** | 0.945833 | **0.970198** | 0.967869 | 0.969433 | **0.969514** |
| mfeat-pixel | **0.999422** | 0.999183 | **0.999378** | 0.998464 | 0.997451 | **0.998448** | **0.999478** | 0.999292 |
| mfeat-zernike | **0.977986** | 0.977831 | 0.974231 | **0.975436** | **0.983479** | 0.981858 | **0.984997** | 0.984036 |
| MiceProtein | **1.000000** | 0.999991 | **0.999923** | 0.999871 | 0.999973 | **1.000000** | **1.000000** | 0.999963 |
| nomao | **0.996439** | 0.995329 | **0.996676** | 0.995051 | 0.990908 | **0.992552** | **0.994828** | 0.994329 |
| numerai28.6 | **0.529404** | 0.529350 | **0.529457** | 0.528295 | **0.530315** | 0.527963 | 0.529336 | **0.529882** |
| optdigits | **0.999844** | 0.999780 | **0.999855** | 0.999738 | **0.999616** | 0.999487 | **0.999939** | 0.999891 |
| ozone-level-8hr | **0.929094** | 0.923125 | **0.922663** | 0.918516 | 0.919484 | **0.919689** | **0.930601** | 0.926141 |
| pc1 | **0.875471** | 0.850199 | **0.863061** | 0.853272 | **0.917591** | 0.840223 | **0.889312** | 0.862341 |
| pc3 | **0.851122** | 0.833421 | **0.854543** | 0.833489 | 0.828743 | **0.835171** | **0.843468** | 0.839846 |
| pc4 | **0.953309** | 0.945471 | **0.951037** | 0.903891 | 0.934944 | **0.944674** | **0.952956** | 0.950541 |
| pendigits | **0.999752** | 0.999728 | 0.999703 | **0.999777** | **0.999703** | 0.999668 | 0.999739 | **0.999756** |
| PhishingWebsites | **0.996482** | 0.995649 | **0.997425** | 0.996704 | **0.996760** | 0.996105 | **0.997636** | 0.996866 |
| phoneme | **0.968024** | 0.958699 | **0.967421** | 0.957712 | **0.965071** | 0.957862 | **0.971200** | 0.962861 |
| qsar-biodeg | **0.930649** | 0.928167 | **0.934875** | 0.917479 | **0.919584** | 0.914716 | **0.937730** | 0.929856 |
| satimage | **0.991978** | 0.990444 | **0.992114** | 0.990114 | **0.993516** | 0.992003 | **0.994291** | 0.992353 |
| segment | **0.996231** | 0.995441 | **0.996126** | 0.994624 | **0.994124** | 0.993598 | **0.994943** | 0.994696 |
| semeion | **0.998687** | 0.997784 | **0.998272** | 0.974216 | 0.995548 | **0.996208** | **0.998425** | 0.997714 |
| sick | **0.998331** | 0.997520 | **0.997950** | 0.995587 | **0.997937** | 0.997762 | **0.997317** | 0.996299 |
| spambase | **0.989935** | 0.988718 | **0.990726** | 0.988292 | **0.985969** | 0.983881 | **0.989244** | 0.988850 |
| splice | **0.995472** | 0.992511 | 0.995049 | **0.997548** | 0.992276 | **0.995195** | **0.995054** | 0.994709 |
| steel-plates-fault | **0.974350** | 0.968766 | **0.972743** | 0.954217 | 0.959182 | **0.962215** | **0.971043** | 0.966795 |
| texture | **0.999948** | 0.999946 | **0.999940** | 0.999834 | **0.999983** | 0.999973 | **0.999997** | 0.999997 |
| tic-tac-toe | **1.000000** | 0.999952 | **0.999710** | 0.999567 | 0.996152 | **0.996209** | **1.000000** | **1.000000** |
| vehicle | **0.943460** | 0.933394 | **0.942080** | 0.929008 | **0.963362** | 0.940233 | **0.965156** | 0.955308 |
| vowel | **0.999259** | 0.998833 | **0.999428** | 0.997587 | **0.999713** | 0.999198 | **0.999966** | 0.999854 |
| wall-robot-navigation | **0.999990** | 0.999910 | **0.999981** | 0.999586 | **0.999900** | 0.999870 | **0.999912** | 0.999873 |
| wdbc | **0.993813** | 0.991693 | **0.994467** | 0.993817 | **0.993967** | 0.986203 | **0.996573** | 0.994058 |
| wilt | 0.990950 | **0.991393** | **0.992192** | 0.991602 | **0.993047** | 0.992642 | 0.994857 | **0.995517** |

# D RAW RESULTS TABLES

## D.1 RESULTS AFTER HYPERPARAMETER OPTIMIZATION

Table 18 shows the raw results after HPO for CatBoost, LightGBM and XGBoost.

Table 18: Average test ROC-AUC per dataset for CatBoost, LightGBM and XGBoost after hyperparameter optimization across CV folds.

| Dataset | CatBoost | LightGBM | XGBoost |
| --- | --- | --- | --- |
| adult | 0.930747 | **0.931261** | 0.930482 |
| analcatdata_authorship | 0.999662 | **0.999986** | 0.999816 |
| analcatdata_dmft | **0.579136** | 0.573445 | 0.572150 |
| balance-scale | 0.972625 | 0.976799 | **0.991268** |
| bank-marketing | **0.938831** | 0.938470 | 0.938384 |
| banknote-authentication | 0.999935 | **0.999979** | 0.999935 |
| Bioresponse | 0.885502 | 0.886734 | **0.888615** |
| blood-transfusion-service-center | **0.754965** | 0.723635 | 0.750671 |
| breast-w | 0.989162 | 0.989162 | **0.992112** |
| car | **1.000000** | 0.999703 | 0.999902 |
| churn | **0.922968** | 0.913858 | 0.914432 |
| climate-model-simulation-crashes | **0.951480** | 0.950490 | 0.947000 |
| cmc | **0.740149** | 0.730807 | 0.735649 |
| cnae-9 | 0.996316 | 0.984404 | **0.997454** |
| connect-4 | 0.921050 | **0.932440** | 0.931952 |
| credit-approval | 0.934006 | 0.930848 | **0.934692** |
| credit-g | **0.801762** | 0.795774 | 0.798571 |
| cylinder-bands | 0.912070 | **0.929117** | 0.928116 |
| diabetes | **0.837869** | 0.827963 | 0.835638 |
| dna | 0.995028 | 0.994942 | **0.995278** |
| dresses-sales | 0.595731 | 0.617323 | **0.622414** |
| electricity | 0.980993 | **0.989807** | 0.987790 |
| eucalyptus | **0.923334** | 0.912027 | 0.918055 |
| first-order-theorem-proving | 0.831775 | 0.833949 | **0.834883** |
| GesturePhaseSegmentationProcessed | 0.916674 | **0.920034** | 0.916761 |
| har | 0.999941 | 0.999959 | **0.999960** |
| ilpd | 0.744702 | 0.711289 | **0.748019** |
| Internet-Advertisements | 0.979120 | 0.980094 | **0.982276** |
| isolet | 0.999389 | 0.999401 | **0.999488** |
| jm1 | 0.756611 | 0.753206 | **0.759652** |
| jungle_chess_2pcs_raw_endgame_complete | 0.976349 | **0.977605** | 0.974087 |
| kc1 | 0.825443 | 0.803456 | **0.832007** |
| kc2 | **0.846802** | 0.843645 | 0.843295 |
| kr-vs-kp | 0.999392 | 0.999755 | **0.999796** |
| letter | **0.999854** | 0.999825 | 0.999819 |
| madelon | **0.937562** | 0.924095 | 0.932249 |
| mfeat-factors | 0.998910 | **0.999125** | 0.999004 |
| mfeat-fourier | **0.984714** | 0.983817 | 0.983375 |
| mfeat-karhunen | **0.999264** | 0.998958 | 0.999211 |
| mfeat-morphological | **0.965406** | 0.961250 | 0.963075 |
| mfeat-pixel | **0.999422** | 0.999131 | 0.999378 |
| mfeat-zernike | **0.977986** | 0.973627 | 0.974231 |
| MiceProtein | **1.000000** | **1.000000** | 0.999923 |
| nomao | 0.996439 | **0.996835** | 0.996676 |
| numerai28.6 | 0.529404 | 0.529077 | **0.529457** |
| optdigits | 0.999844 | 0.999818 | **0.999855** |
| ozone-level-8hr | **0.929094** | 0.923040 | 0.922663 |
| pc1 | **0.875471** | 0.874508 | 0.863061 |
| pc3 | 0.851122 | 0.843094 | **0.854543** |
| pc4 | **0.953309** | 0.946295 | 0.951037 |
| pendigits | **0.999752** | 0.999735 | 0.999703 |
| PhishingWebsites | 0.996482 | **0.997542** | 0.997425 |
| phoneme | **0.968024** | 0.965147 | 0.967421 |
| qsar-biodeg | 0.930649 | 0.931863 | **0.934875** |
| satimage | 0.991978 | 0.991309 | **0.992114** |
| segment | 0.996231 | **0.996233** | 0.996126 |
| semeion | **0.998687** | 0.997646 | 0.998272 |
| sick | **0.998331** | 0.998073 | 0.997950 |
| spambase | 0.989935 | 0.989995 | **0.990726** |
| splice | **0.995472** | 0.995103 | 0.995049 |
| steel-plates-fault | **0.974350** | 0.973788 | 0.972743 |
| texture | **0.999948** | 0.999890 | 0.999940 |
| tic-tac-toe | **1.000000** | **1.000000** | 0.999710 |
| vehicle | **0.943460** | 0.935611 | 0.942080 |
| vowel | 0.999259 | 0.998466 | **0.999428** |
| wall-robot-navigation | **0.999990** | 0.999968 | 0.999981 |
| wdbc | 0.993813 | **0.994729** | 0.994467 |
| wilt | 0.990950 | 0.985504 | **0.992192** |

Table 19 shows the raw results after HPO for dataset-specific neural networks.

Table 19: Average test ROC-AUC per dataset for dataset-specific neural networks after hyperparameter optimization across CV folds. Missing datasets are represented by "-".

| Dataset | FT-Transformer | MLP | ModernNCA | RealMLP | ResNet | SAINT | TabM | TabNet |
|---|---|---|---|---|---|---|---|---|
| adult | 0.914869 | 0.928689 | **0.930138** | 0.923327 | 0.913790 | 0.920246 | 0.919662 | 0.882450 |
| analcatdata_authorship | 0.999985 | 0.999770 | **1.000000** | 1.000000 | 1.000000 | 0.999974 | 1.000000 | 0.999249 |
| analcatdata_dmft | 0.576947 | 0.574532 | 0.569168 | 0.574396 | **0.584338** | 0.544695 | 0.576017 | 0.515962 |
| balance-scale | 0.999735 | 0.998659 | 0.996276 | **1.000000** | 0.989061 | 0.999266 | 0.998912 | 0.979668 |
| bank-marketing | 0.938198 | 0.937054 | 0.936903 | 0.937031 | 0.935740 | 0.936560 | **0.941872** | 0.887319 |
| banknote-authentication | **1.000000** | 1.000000 | 1.000000 | 1.000000 | 1.000000 | 1.000000 | 1.000000 | 1.000000 |
| Bioresponse | 0.820159 | 0.825631 | - | 0.859065 | 0.850801 | - | **0.876671** | - |
| blood-transfusion-service-center | 0.745975 | **0.770627** | 0.749882 | 0.746350 | 0.738502 | 0.746726 | 0.748538 | 0.660675 |
| breast-w | 0.989503 | 0.992380 | 0.991173 | 0.992882 | 0.995477 | 0.988470 | **0.995845** | 0.986694 |
| car | 0.999751 | 0.999992 | 0.999950 | 1.000000 | 0.994154 | 1.000000 | 1.000000 | 1.000000 |
| churn | 0.914596 | 0.922938 | 0.927641 | 0.913533 | 0.918713 | 0.915603 | **0.929636** | 0.891443 |
| climate-model-simulation-crashes | 0.934671 | 0.948857 | 0.939969 | **0.962163** | 0.918990 | 0.925643 | 0.939969 | 0.868204 |
| cmc | 0.739402 | 0.744580 | **0.744760** | 0.735472 | 0.737829 | 0.738490 | 0.743797 | 0.647121 |
| cnae-9 | 0.994497 | 0.996716 | **0.998881** | 0.997569 | 0.997106 | - | 0.998100 | - |
| connect-4 | 0.901170 | 0.927373 | 0.936124 | 0.928258 | 0.933333 | - | **0.941654** | - |
| credit-approval | 0.935798 | **0.938866** | 0.934957 | 0.917352 | 0.933113 | 0.933493 | 0.934458 | 0.878500 |
| credit-g | 0.783048 | 0.788476 | 0.782810 | 0.779381 | 0.783524 | 0.786402 | **0.790905** | 0.696905 |
| cylinder-bands | 0.915494 | 0.886405 | **0.932243** | 0.910680 | 0.909989 | 0.923391 | 0.926477 | 0.837792 |
| diabetes | 0.831108 | 0.837342 | **0.838672** | 0.837507 | 0.821798 | 0.827285 | 0.829801 | 0.756416 |
| dna | 0.990937 | 0.992220 | 0.993932 | 0.994111 | 0.992543 | 0.992473 | **0.994505** | 0.991448 |
| dresses-sales | 0.620033 | 0.635468 | **0.657143** | 0.537849 | 0.575205 | 0.624704 | 0.642200 | 0.555993 |
| electricity | 0.963076 | 0.969201 | **0.995199** | 0.961467 | 0.960658 | 0.967012 | 0.968731 | 0.938656 |
| eucalyptus | 0.923933 | 0.921873 | 0.928964 | 0.915693 | 0.916785 | 0.925970 | **0.931897** | 0.872365 |
| first-order-theorem-proving | 0.796707 | 0.798812 | **0.825830** | 0.795637 | 0.784636 | 0.802392 | 0.818255 | 0.774094 |
| GesturePhaseSegmentationProcessed | 0.895166 | 0.911434 | **0.956087** | 0.901441 | 0.914196 | 0.919006 | 0.933828 | 0.850596 |
| har | 0.999685 | 0.999783 | - | 0.999959 | 0.999921 | - | **0.999966** | 0.999515 |
| ilpd | **0.751488** | 0.671938 | 0.736509 | 0.729412 | 0.747491 | 0.698718 | 0.744875 | 0.704840 |
| Internet-Advertisements | 0.974513 | - | 0.983818 | 0.973810 | 0.974187 | - | **0.985640** | - |
| isolet | 0.998817 | 0.998295 | 0.998432 | 0.999635 | 0.999401 | - | **0.999750** | 0.998813 |
| jm1 | 0.709321 | 0.715620 | **0.759996** | 0.713988 | 0.720444 | 0.719464 | 0.751557 | 0.674043 |
| jungle_chess_2pcs_raw_endgame_complete | 0.999975 | 0.999965 | **0.999996** | 0.999774 | 0.999956 | 0.999926 | 0.999985 | 0.991981 |
| kc1 | 0.783519 | 0.805465 | **0.819052** | 0.796117 | 0.806819 | 0.796918 | 0.813763 | 0.762807 |
| kc2 | 0.832014 | 0.829426 | 0.835713 | **0.845768** | 0.833248 | 0.834436 | 0.833491 | 0.713458 |
| kr-vs-kp | 0.999777 | 0.999686 | 0.999217 | 0.999704 | 0.999369 | **0.999789** | 0.999652 | 0.998872 |
| letter | 0.999919 | 0.999894 | **0.999964** | 0.999914 | 0.999926 | 0.999853 | 0.999943 | 0.999606 |
| madelon | 0.747391 | 0.883991 | 0.851408 | **0.930302** | 0.605018 | - | 0.809941 | 0.630669 |
| mfeat-factors | 0.999015 | 0.998875 | 0.999478 | 0.999625 | 0.999472 | 0.999385 | **0.999700** | 0.998125 |
| mfeat-fourier | 0.984511 | 0.984929 | 0.984532 | 0.985483 | 0.981725 | 0.980508 | **0.988497** | 0.970539 |
| mfeat-karhunen | 0.998682 | 0.998849 | 0.999208 | 0.999019 | 0.998448 | 0.999078 | **0.999521** | 0.996960 |
| mfeat-morphological | **0.970198** | 0.967719 | 0.964008 | 0.969994 | 0.968651 | 0.967681 | 0.969433 | 0.955818 |
| mfeat-pixel | 0.997451 | 0.998674 | 0.999256 | **0.999492** | 0.998690 | 0.999217 | 0.999478 | 0.998200 |
| mfeat-zernike | 0.983479 | 0.984610 | 0.977900 | 0.982993 | 0.984488 | 0.981874 | **0.984997** | 0.968629 |
| MiceProtein | 0.999973 | 0.999973 | **1.000000** | 0.999971 | 0.999973 | 1.000000 | 1.000000 | 0.999344 |
| nomao | 0.990908 | 0.986577 | - | 0.989803 | 0.993048 | - | **0.994828** | - |
| numerai28.6 | **0.530315** | 0.525920 | 0.508863 | 0.529534 | 0.528012 | 0.525822 | 0.529336 | - |
| optdigits | 0.999616 | 0.999794 | 0.999886 | **0.999968** | 0.999927 | 0.999841 | 0.999939 | 0.998871 |
| ozone-level-8hr | 0.919484 | 0.927900 | **0.930968** | 0.923252 | 0.925416 | 0.919315 | 0.930601 | 0.864067 |
| pc1 | **0.917591** | 0.832532 | 0.901736 | 0.844517 | 0.889458 | 0.870543 | 0.889312 | 0.804412 |
| pc3 | 0.828743 | 0.842511 | **0.845284** | 0.814590 | 0.829637 | 0.827322 | 0.843468 | 0.788151 |
| pc4 | 0.934944 | 0.945813 | 0.945019 | 0.939257 | 0.944447 | 0.934528 | **0.952956** | 0.920943 |
| pendigits | 0.999703 | 0.999705 | **0.999788** | 0.999659 | 0.999638 | 0.999782 | 0.999739 | 0.999753 |
| PhishingWebsites | 0.996760 | 0.996991 | **0.998275** | 0.997208 | 0.996975 | 0.996746 | 0.997636 | 0.996196 |
| phoneme | 0.965071 | 0.967617 | **0.977289** | 0.966456 | 0.963591 | 0.960382 | 0.971200 | 0.956279 |
| qsar-biodeg | 0.919584 | 0.924951 | 0.926952 | 0.929226 | 0.932220 | 0.930632 | **0.937730** | 0.902748 |
| satimage | 0.993516 | 0.992308 | 0.993801 | 0.993034 | 0.991995 | 0.992630 | **0.994291** | 0.987482 |
| segment | 0.994124 | 0.995046 | **0.996419** | 0.994075 | 0.993581 | 0.994831 | 0.994943 | 0.992317 |
| semeion | 0.995548 | 0.997350 | 0.998727 | **0.998976** | 0.997689 | 0.997630 | 0.998425 | 0.994019 |
| sick | 0.997937 | 0.997048 | 0.992457 | **0.998661** | 0.968841 | 0.998281 | 0.997317 | 0.981838 |
| spambase | 0.985969 | 0.988185 | 0.988651 | 0.987799 | 0.987683 | 0.986263 | **0.989244** | 0.980804 |
| splice | 0.992276 | 0.994053 | 0.994526 | 0.994420 | 0.993514 | **0.995073** | 0.995054 | 0.990441 |
| steel-plates-fault | 0.959182 | 0.964693 | **0.971934** | 0.959639 | 0.949067 | 0.955379 | 0.971043 | 0.947456 |
| texture | 0.999983 | 0.999991 | 0.999983 | **0.999999** | 0.999999 | 0.999976 | 0.999997 | 0.999763 |
| tic-tac-toe | 0.996152 | **1.000000** | 1.000000 | 0.999711 | 0.999462 | 0.999725 | 1.000000 | 0.993030 |
| vehicle | 0.963362 | 0.961813 | 0.964812 | 0.965844 | **0.967212** | 0.955127 | 0.965156 | 0.943787 |
| vowel | 0.999713 | 0.999638 | 0.999955 | 0.999955 | 0.999813 | 0.999875 | **0.999966** | 0.999686 |
| wall-robot-navigation | 0.999900 | 0.999689 | 0.999850 | 0.998720 | 0.999042 | 0.999844 | **0.999912** | 0.997585 |
| wdbc | 0.993967 | 0.996065 | 0.994995 | 0.996038 | 0.995409 | 0.995546 | **0.996573** | 0.986656 |
| wilt | 0.993047 | **0.997690** | 0.994959 | 0.993197 | 0.990726 | 0.993139 | 0.994857 | 0.991289 |

Table 20 shows the raw results after HPO for the meta-learned neural networks.

Table 20: Average test ROC-AUC per dataset for meta-learned neural networks after hyperparameter optimization across CV folds. Missing datasets are represented by "-".

| Dataset | CARTE | LimiX | Mitra | TPBerta | TabICL | TabPFN | TabPFNv2 | XTab |
|---|---|---|---|---|---|---|---|---|
| adult | 0.902677 | **0.930281** | - | - | 0.914430 | - | - | - |
| analcatdata_authorship | 0.999181 | **1.000000** | **1.000000** | - | **1.000000** | **1.000000** | **1.000000** | 0.999991 |
| analcatdata_dmft | 0.586376 | 0.589864 | **0.630016** | - | 0.591336 | 0.586630 | 0.588236 | 0.556971 |
| balance-scale | **0.999413** | 0.994483 | 0.992957 | - | 0.997980 | 0.997656 | 0.995312 | 0.997420 |
| bank-marketing | 0.924664 | **0.943913** | - | - | 0.940210 | - | - | - |
| banknote-authentication | **1.000000** | 1.000000 | 1.000000 | 0.994512 | 1.000000 | - | 1.000000 | 1.000000 |
| Bioresponse | - | **0.886859** | - | - | 0.885075 | - | - | - |
| blood-transfusion-service-center | 0.739571 | 0.743212 | **0.763112** | 0.633041 | 0.743063 | 0.752586 | 0.754893 | - |
| breast-w | 0.987912 | 0.993777 | 0.993595 | 0.986514 | 0.993152 | 0.994131 | **0.994132** | 0.989666 |
| car | 0.997126 | 0.999466 | 0.999812 | - | 0.999232 | - | 0.999963 | - |
| churn | 0.923626 | 0.929681 | **0.937453** | - | 0.923984 | - | 0.923208 | - |
| climate-model-simulation-crashes | 0.938531 | 0.963561 | **0.984184** | - | 0.932612 | 0.968010 | 0.958663 | 0.944367 |
| cmc | 0.738379 | 0.742419 | **0.749328** | - | 0.740035 | - | 0.746447 | - |
| cnae-9 | 0.990151 | 0.997695 | - | - | **0.997840** | - | - | - |
| connect-4 | - | - | - | - | **0.897904** | - | - | - |
| credit-approval | 0.909279 | 0.935976 | 0.936278 | 0.901989 | **0.941488** | 0.932397 | 0.940813 | 0.939620 |
| credit-g | 0.769619 | **0.801143** | 0.796262 | - | 0.799048 | 0.768476 | 0.793429 | - |
| cylinder-bands | 0.848539 | 0.952527 | **0.954056** | 0.820399 | 0.926679 | 0.886616 | 0.904451 | 0.881396 |
| diabetes | 0.823615 | 0.833345 | 0.835083 | 0.778356 | 0.835442 | 0.836120 | **0.844356** | 0.815847 |
| dna | 0.986120 | 0.994527 | 0.981938 | - | 0.994123 | - | **0.995658** | 0.992479 |
| dresses-sales | 0.589655 | 0.599343 | **0.657800** | 0.534893 | 0.605090 | 0.538916 | 0.608456 | 0.613136 |
| electricity | 0.909407 | **0.995629** | - | - | 0.970809 | - | - | 0.966899 |
| eucalyptus | 0.905245 | 0.936208 | **0.963758** | - | 0.934423 | 0.928493 | 0.933540 | 0.918317 |
| first-order-theorem-proving | 0.764092 | **0.835198** | 0.789747 | - | 0.834629 | - | 0.825502 | 0.798803 |
| GesturePhaseSegmentationProcessed | 0.798024 | **0.957290** | 0.809182 | - | 0.951408 | - | 0.936548 | 0.886960 |
| har | - | **0.999993** | - | - | 0.999913 | - | - | - |
| ilpd | 0.704712 | 0.773976 | **0.839529** | 0.586083 | 0.780714 | 0.757892 | 0.745193 | 0.726413 |
| Internet-Advertisements | - | 0.984687 | - | - | **0.989308** | - | - | - |
| isolet | - | **0.999570** | - | - | - | - | - | - |
| jm1 | 0.728512 | 0.772956 | - | - | **0.784425** | - | - | 0.727984 |
| jungle_chess_2pcs_raw_endgame_complete | 0.973383 | 0.993452 | - | - | 0.975471 | - | - | **0.999950** |
| kc1 | 0.797680 | 0.843266 | 0.830522 | - | **0.849627** | - | 0.836795 | 0.803082 |
| kc2 | 0.842828 | 0.837530 | **0.863051** | - | 0.834741 | 0.850065 | 0.837427 | 0.835476 |
| kr-vs-kp | 0.999685 | 0.999643 | 0.991192 | 0.855273 | **0.999792** | - | 0.999408 | 0.999616 |
| letter | 0.999440 | - | - | - | **0.999957** | - | - | 0.999859 |
| madelon | 0.836760 | **0.965107** | - | - | 0.711538 | - | - | 0.845746 |
| mfeat-factors | 0.996064 | 0.999733 | 0.999225 | - | **0.999808** | - | 0.999650 | 0.998443 |
| mfeat-fourier | 0.976986 | 0.991194 | 0.988586 | - | 0.989372 | - | **0.991319** | 0.982539 |
| mfeat-karhunen | 0.994814 | 0.999744 | 0.999250 | - | **0.999850** | - | 0.999622 | 0.998582 |
| mfeat-morphological | 0.967325 | 0.968194 | 0.967950 | - | 0.968919 | - | **0.969308** | 0.967136 |
| mfeat-pixel | 0.996175 | 0.999642 | 0.998814 | - | **0.999664** | - | 0.999503 | 0.998642 |
| mfeat-zernike | 0.978119 | 0.991686 | 0.985958 | - | **0.992247** | - | 0.991483 | 0.980183 |
| MiceProtein | 0.999582 | **1.000000** | 0.999991 | - | **1.000000** | - | **1.000000** | **1.000000** |
| nomao | - | **0.997403** | - | - | 0.996055 | - | - | 0.992727 |
| numerai28.6 | 0.514361 | - | - | - | 0.526838 | - | - | **0.528062** |
| optdigits | 0.999112 | 0.999961 | 0.999456 | - | **0.999989** | - | 0.999897 | 0.999712 |
| ozone-level-8hr | 0.890063 | 0.936072 | **0.970446** | - | 0.936234 | - | 0.933398 | 0.915744 |
| pc1 | 0.835444 | **0.914786** | 0.909491 | - | 0.912419 | - | 0.906419 | 0.855741 |
| pc3 | 0.831574 | 0.866891 | 0.853288 | 0.625642 | **0.867955** | - | 0.854460 | 0.823532 |
| pc4 | 0.937337 | **0.960869** | 0.910570 | 0.744304 | 0.959970 | - | 0.958887 | 0.938455 |
| pendigits | 0.999468 | 0.999820 | - | - | **0.999852** | - | - | 0.999751 |
| PhishingWebsites | 0.994582 | 0.997750 | - | - | **0.998342** | - | - | 0.996896 |
| phoneme | 0.948702 | **0.980179** | 0.955580 | 0.796404 | 0.977493 | - | 0.973546 | 0.961749 |
| qsar-biodeg | 0.921153 | 0.944323 | **0.970952** | 0.833852 | 0.943963 | - | 0.942895 | 0.926795 |
| satimage | 0.988038 | **0.995246** | 0.984750 | - | 0.993373 | - | 0.995122 | 0.992918 |
| segment | 0.993491 | 0.997477 | **0.997745** | - | 0.997462 | - | 0.997547 | 0.994697 |
| semeion | 0.993378 | **0.999240** | 0.995907 | - | 0.999033 | - | 0.998288 | 0.997064 |
| sick | 0.995762 | **0.998835** | 0.996697 | - | 0.997012 | - | 0.998008 | 0.998232 |
| spambase | 0.983228 | **0.992506** | 0.986240 | - | 0.992153 | - | 0.991218 | 0.986044 |
| splice | 0.987950 | 0.995214 | **0.995549** | - | 0.993905 | - | 0.995288 | 0.992444 |
| steel-plates-fault | 0.943636 | 0.982973 | 0.979979 | - | 0.978541 | - | **0.984454** | 0.957088 |
| texture | 0.999541 | - | - | - | **1.000000** | - | - | 0.999962 |
| tic-tac-toe | 0.984361 | **1.000000** | **1.000000** | 0.993803 | 0.999182 | 0.996086 | 0.999663 | **1.000000** |
| vehicle | 0.941691 | 0.975236 | **0.979394** | - | 0.978088 | 0.970556 | 0.975896 | 0.955838 |
| vowel | 0.998092 | - | - | - | **1.000000** | - | - | 0.999630 |
| wall-robot-navigation | 0.999505 | **0.999984** | 0.999976 | - | 0.999610 | - | 0.999936 | 0.999846 |
| wdbc | 0.990612 | 0.996713 | 0.997501 | - | 0.996697 | 0.996298 | **0.997761** | 0.994317 |
| wilt | 0.994858 | 0.993835 | **0.996723** | 0.880733 | 0.995557 | - | 0.996605 | 0.994261 |

Lastly, Table 21 shows the raw results of AutoGluon using HPO and AutoGluon with its recommended settings.

Table 21: Average test ROC-AUC per dataset for AutoGluon with HPO and AutoGluon with its recommended settings across CV folds.

| Dataset | AutoGluon | AutoGluon (HPO) |
|---|---|---|
| adult | **0.931792** | 0.931658 |
| analcatdata_authorship | **1.000000** | 0.999887 |
| analcatdata_dmft | **0.577809** | 0.553672 |
| balance-scale | **0.997339** | 0.995057 |
| bank-marketing | **0.941273** | 0.940659 |
| banknote-authentication | **1.000000** | 0.999957 |
| Bioresponse | **0.888693** | 0.881238 |
| blood-transfusion-service-center | **0.741733** | 0.733305 |
| breast-w | **0.994394** | 0.993510 |
| car | 0.999861 | **0.999998** |
| churn | **0.927520** | 0.920213 |
| climate-model-simulation-crashes | **0.970051** | 0.926306 |
| cmc | **0.737077** | 0.536500 |
| cnae-9 | **0.998524** | 0.997965 |
| connect-4 | 0.934636 | **0.941976** |
| credit-approval | **0.940476** | 0.933497 |
| credit-g | **0.802381** | 0.773238 |
| cylinder-bands | **0.933320** | 0.903658 |
| diabetes | **0.833641** | 0.827171 |
| dna | **0.995385** | 0.994906 |
| dresses-sales | **0.615107** | 0.597537 |
| electricity | **0.987260** | 0.986609 |
| eucalyptus | **0.933782** | 0.925856 |
| first-order-theorem-proving | **0.835425** | 0.825561 |
| GesturePhaseSegmentationProcessed | **0.936667** | 0.917835 |
| har | **0.999958** | 0.999942 |
| ilpd | **0.765098** | 0.745564 |
| Internet-Advertisements | **0.985963** | 0.984740 |
| isolet | **0.999744** | 0.999696 |
| jm1 | **0.770272** | 0.761065 |
| jungle_chess_2pcs_raw_endgame_complete | 0.999278 | **0.999444** |
| kc1 | **0.835974** | 0.815660 |
| kc2 | **0.834913** | 0.813625 |
| kr-vs-kp | 0.999405 | **0.999412** |
| letter | **0.999934** | 0.999933 |
| madelon | **0.932817** | 0.929882 |
| mfeat-factors | **0.999350** | 0.999111 |
| mfeat-fourier | 0.986058 | **0.986717** |
| mfeat-karhunen | **0.999575** | 0.998740 |
| mfeat-morphological | **0.977508** | 0.968908 |
| mfeat-pixel | **0.999403** | 0.999139 |
| mfeat-zernike | **0.995249** | 0.985279 |
| MiceProtein | 0.999929 | **0.999981** |
| nomao | **0.996892** | 0.996441 |
| numerai28.6 | **0.530150** | 0.527692 |
| optdigits | **0.999925** | 0.999893 |
| ozone-level-8hr | **0.936029** | 0.930880 |
| pc1 | **0.888177** | 0.860825 |
| pc3 | **0.865766** | 0.845648 |
| pc4 | **0.955384** | 0.950117 |
| pendigits | **0.999725** | 0.999642 |
| PhishingWebsites | **0.997572** | 0.997102 |
| phoneme | **0.973342** | 0.964555 |
| qsar-biodeg | **0.942988** | 0.932276 |
| satimage | **0.993557** | 0.993220 |
| segment | **0.996895** | 0.996421 |
| semeion | **0.998506** | 0.998210 |
| sick | **0.998367** | 0.997357 |
| spambase | **0.991092** | 0.989781 |
| splice | **0.995941** | 0.995249 |
| steel-plates-fault | **0.973843** | 0.972323 |
| texture | **0.999998** | 0.999995 |
| tic-tac-toe | **1.000000** | 0.996585 |
| vehicle | **0.969797** | 0.965886 |
| vowel | **0.999910** | 0.999618 |
| wall-robot-navigation | **0.999993** | 0.999984 |
| wdbc | **0.995799** | 0.992456 |
| wilt | **0.995652** | 0.994495 |

## D.2 RESULTS USING DEFAULT HYPERPARAMETER CONFIGURATIONS

Table 22 shows the raw results for CatBoost and XGBoost using the default hyperparameter configurations.

Table 22: Average test ROC-AUC per dataset for CatBoost, LightGBM and XGBoost using the default hyperparamater configurations across CV folds.

| Dataset | CatBoost | LightGBM | XGBoost |
|---|---|---|---|
| adult | **0.930571** | 0.929995 | 0.929316 |
| analcatdata_authorship | 0.999710 | **0.999970** | 0.999518 |
| analcatdata_dmft | **0.549171** | 0.538902 | 0.531850 |
| balance-scale | **0.952530** | 0.920593 | 0.926923 |
| bank-marketing | **0.938725** | 0.937425 | 0.934864 |
| banknote-authentication | **0.999957** | 0.999613 | 0.999914 |
| Bioresponse | 0.879217 | **0.880857** | 0.880176 |
| blood-transfusion-service-center | **0.729842** | 0.706352 | 0.712258 |
| breast-w | **0.991254** | 0.990699 | 0.990430 |
| car | 0.999509 | **0.999672** | 0.998790 |
| churn | **0.924606** | 0.917109 | 0.913882 |
| climate-model-simulation-crashes | **0.962296** | 0.949276 | 0.955828 |
| cmc | **0.709590** | 0.695848 | 0.684939 |
| cnae-9 | **0.996007** | 0.983430 | 0.994232 |
| connect-4 | 0.893587 | 0.886247 | **0.899588** |
| credit-approval | **0.937424** | 0.925672 | 0.930615 |
| credit-g | **0.800667** | 0.787000 | 0.788381 |
| cylinder-bands | 0.885160 | 0.907731 | **0.912564** |
| diabetes | **0.835137** | 0.798912 | 0.797009 |
| dna | 0.994641 | **0.994798** | 0.994699 |
| dresses-sales | **0.598768** | 0.565517 | 0.570699 |
| electricity | 0.958153 | 0.954700 | **0.971787** |
| eucalyptus | **0.921691** | 0.903510 | 0.902805 |
| first-order-theorem-proving | 0.826532 | **0.828733** | 0.826895 |
| GesturePhaseSegmentationProcessed | **0.898407** | 0.889753 | 0.892459 |
| har | 0.999899 | **0.999938** | 0.999905 |
| ilpd | 0.741153 | **0.745050** | 0.722052 |
| Internet-Advertisements | **0.979992** | 0.978933 | 0.976972 |
| isolet | **0.999407** | 0.999095 | 0.998854 |
| jm1 | 0.748060 | **0.749102** | 0.729353 |
| jungle_chess_2pcs_raw_endgame_complete | 0.972286 | 0.971617 | **0.974856** |
| kc1 | **0.823661** | 0.791316 | 0.791182 |
| kc2 | **0.821163** | 0.787678 | 0.771390 |
| kr-vs-kp | 0.999521 | **0.999727** | 0.999720 |
| letter | **0.999740** | 0.999724 | 0.999648 |
| madelon | **0.928172** | 0.902450 | 0.890107 |
| mfeat-factors | **0.999031** | 0.998867 | 0.998356 |
| mfeat-fourier | **0.984181** | 0.981478 | 0.982669 |
| mfeat-karhunen | **0.999128** | 0.998500 | 0.997700 |
| mfeat-morphological | **0.962489** | 0.955839 | 0.958908 |
| mfeat-pixel | **0.999289** | 0.998861 | 0.998703 |
| mfeat-zernike | **0.972961** | 0.965408 | 0.966633 |
| MiceProtein | **0.999983** | 0.999944 | 0.999680 |
| nomao | 0.995620 | 0.995122 | **0.995690** |
| numerai28.6 | 0.518341 | **0.521861** | 0.511976 |
| optdigits | **0.999808** | 0.999690 | 0.999586 |
| ozone-level-8hr | **0.925485** | 0.916990 | 0.911594 |
| pc1 | **0.891257** | 0.874492 | 0.857895 |
| pc3 | **0.850219** | 0.819425 | 0.816916 |
| pc4 | **0.953689** | 0.949308 | 0.942808 |
| pendigits | 0.999764 | **0.999772** | 0.999760 |
| PhishingWebsites | 0.995801 | 0.996155 | **0.996764** |
| phoneme | 0.955202 | 0.956695 | **0.957311** |
| qsar-biodeg | **0.934769** | 0.933991 | 0.926970 |
| satimage | **0.991815** | 0.990983 | 0.990907 |
| segment | 0.996012 | **0.996016** | 0.995267 |
| semeion | **0.998163** | 0.996984 | 0.996029 |
| sick | 0.998355 | **0.998355** | 0.996943 |
| spambase | 0.989066 | **0.990138** | 0.988888 |
| splice | **0.995198** | 0.994463 | 0.994788 |
| steel-plates-fault | 0.972233 | **0.973473** | 0.970148 |
| texture | **0.999908** | 0.999856 | 0.999795 |
| tic-tac-toe | **1.000000** | 0.998990 | 0.999181 |
| vehicle | **0.942832** | 0.936072 | 0.935079 |
| vowel | **0.999237** | 0.998215 | 0.996947 |
| wall-robot-navigation | **0.999989** | 0.999955 | 0.999934 |
| wdbc | 0.994217 | 0.992350 | **0.994471** |
| wilt | **0.991488** | 0.987326 | 0.988659 |

Table 23 shows the raw results for dataset-specific neural networks using the default hyperparameter configurations.

Table 23: Average test ROC-AUC per dataset for dataset-specific neural networks using default hyperparameter configurations across CV folds. Missing datasets are represented by "-".

| Dataset | FT-Transformer | MLP | ModernNCA | RealMLP | ResNet | SAINT | TabM | TabNet |
|---|---|---|---|---|---|---|---|---|
| adult | 0.893029 | 0.897504 | **0.927541** | 0.909085 | 0.905838 | 0.870099 | 0.908670 | 0.912781 |
| analcatdata_authorship | 0.999392 | 0.999934 | 0.972957 | 0.999952 | **1.000000** | 0.999983 | **1.000000** | 0.993186 |
| analcatdata_dmft | 0.553755 | 0.554240 | 0.554324 | **0.575007** | 0.553675 | 0.526597 | 0.539602 | 0.534271 |
| balance-scale | 0.988863 | 0.995111 | 0.989139 | 0.980107 | 0.992229 | 0.991970 | **0.998859** | 0.972816 |
| bank-marketing | 0.907667 | 0.904699 | 0.911086 | 0.814657 | 0.926617 | 0.892316 | **0.931979** | 0.927765 |
| banknote-authentication | **1.000000** | **1.000000** | 0.999957 | **1.000000** | **1.000000** | **1.000000** | **1.000000** | **1.000000** |
| Bioresponse | 0.804580 | 0.560952 | 0.726775 | 0.824996 | 0.843462 | - | **0.872512** | 0.812061 |
| blood-transfusion-service-center | 0.713181 | **0.762080** | 0.722056 | 0.746119 | 0.742088 | 0.723673 | 0.727600 | 0.728919 |
| breast-w | 0.988615 | 0.994222 | 0.990267 | 0.993411 | 0.991140 | 0.992220 | **0.995038** | 0.984383 |
| car | 0.999758 | 0.999678 | 0.999993 | **1.000000** | 0.998600 | 0.999828 | **1.000000** | 0.931659 |
| churn | 0.915966 | 0.903166 | 0.909130 | 0.917916 | 0.914732 | 0.910996 | **0.927037** | 0.905642 |
| climate-model-simulation-crashes | 0.840724 | 0.935571 | 0.872765 | **0.947857** | 0.904025 | 0.937306 | 0.946051 | 0.825571 |
| cmc | 0.686016 | 0.710134 | **0.711891** | 0.700557 | 0.687757 | 0.642394 | 0.692772 | 0.689043 |
| cnae-9 | 0.994801 | 0.500463 | 0.944599 | 0.992911 | 0.996595 | - | **0.997415** | 0.912423 |
| connect-4 | 0.922969 | 0.915051 | 0.908236 | 0.909829 | 0.926041 | 0.756318 | **0.938629** | 0.856762 |
| credit-approval | 0.915482 | **0.931215** | 0.892881 | 0.914193 | 0.916769 | 0.908623 | 0.920201 | 0.875614 |
| credit-g | 0.731714 | 0.726875 | 0.725714 | 0.758571 | 0.735071 | 0.744000 | **0.782714** | 0.632571 |
| cylinder-bands | 0.908565 | 0.874205 | 0.845421 | 0.904906 | 0.891759 | 0.909314 | **0.924863** | 0.710240 |
| diabetes | 0.755846 | **0.829853** | 0.755741 | 0.822211 | 0.789923 | 0.737127 | 0.789142 | 0.785077 |
| dna | 0.988362 | 0.990128 | 0.986305 | 0.988320 | 0.992218 | 0.520670 | **0.993741** | 0.962713 |
| dresses-sales | 0.571921 | 0.536782 | **0.636946** | 0.525944 | 0.536617 | 0.568144 | 0.542365 | 0.560591 |
| electricity | 0.963347 | 0.950665 | **0.994729** | 0.950555 | 0.930924 | 0.960991 | 0.959880 | 0.911419 |
| eucalyptus | 0.917340 | 0.922173 | 0.887657 | 0.903412 | 0.897582 | 0.904708 | **0.925170** | 0.877684 |
| first-order-theorem-proving | 0.796282 | 0.782461 | 0.800474 | 0.781809 | 0.793079 | 0.772449 | **0.814144** | 0.743350 |
| GesturePhaseSegmentationProcessed | 0.827939 | 0.819054 | **0.940474** | 0.890444 | 0.853272 | 0.893255 | 0.874995 | 0.781506 |
| har | 0.999876 | 0.999848 | 0.867672 | 0.999630 | 0.999859 | - | **0.999937** | 0.999147 |
| ilpd | 0.724591 | 0.748217 | 0.720786 | 0.727899 | 0.758030 | 0.713191 | **0.759658** | 0.715948 |
| Internet-Advertisements | 0.973465 | **0.982883** | 0.939059 | 0.961953 | 0.967077 | - | 0.981634 | 0.892480 |
| isolet | 0.999463 | 0.847095 | 0.692288 | 0.999135 | 0.999307 | - | **0.999671** | 0.997706 |
| jm1 | 0.723314 | 0.726646 | **0.759254** | 0.721977 | 0.734238 | 0.652524 | 0.738839 | 0.722615 |
| jungle_chess_2pcs_raw_endgame_complete | 0.998738 | 0.998486 | 0.997724 | 0.996257 | 0.977410 | **0.999876** | 0.998544 | 0.974173 |
| kc1 | 0.804719 | 0.801565 | 0.793950 | 0.806604 | 0.795200 | 0.742990 | **0.820436** | 0.792858 |
| kc2 | 0.805644 | **0.840419** | 0.715309 | 0.829826 | 0.771497 | 0.742400 | 0.818842 | 0.806986 |
| kr-vs-kp | 0.999792 | 0.999765 | 0.999151 | 0.998737 | 0.999476 | 0.723052 | **0.999796** | 0.987183 |
| letter | 0.999825 | 0.999640 | 0.999889 | 0.999820 | 0.999864 | 0.999784 | **0.999918** | 0.997271 |
| madelon | 0.770769 | 0.500000 | 0.502568 | **0.915592** | 0.600713 | - | 0.758018 | 0.559015 |
| mfeat-factors | 0.998765 | 0.998668 | 0.905171 | 0.999075 | 0.998892 | 0.499849 | **0.999589** | 0.993717 |
| mfeat-fourier | 0.977475 | 0.978653 | 0.894525 | 0.974028 | 0.980419 | 0.971772 | **0.986450** | 0.961111 |
| mfeat-karhunen | 0.997503 | 0.998582 | 0.962010 | **0.999439** | 0.998097 | 0.998387 | 0.998958 | 0.982592 |
| mfeat-morphological | 0.967733 | 0.965494 | 0.962733 | 0.968706 | **0.969308** | 0.967478 | 0.967408 | 0.963611 |
| mfeat-pixel | 0.997658 | 0.946632 | 0.957118 | **0.999500** | 0.998676 | 0.553414 | 0.999317 | 0.992500 |
| mfeat-zernike | 0.978039 | 0.980681 | 0.967817 | 0.965872 | **0.980858** | 0.969257 | 0.978049 | 0.966992 |
| MiceProtein | **1.000000** | 0.999963 | 0.995471 | **1.000000** | 0.999963 | **1.000000** | 0.999991 | 0.987043 |
| nomao | 0.992049 | 0.991436 | 0.920087 | 0.983015 | 0.992530 | 0.499521 | **0.994784** | 0.991441 |
| numerai28.6 | 0.507813 | 0.513601 | 0.500280 | 0.522412 | 0.517071 | 0.507780 | **0.523854** | 0.522797 |
| optdigits | 0.999631 | 0.999454 | 0.966265 | 0.999927 | 0.999837 | 0.999057 | **0.999927** | 0.998476 |
| ozone-level-8hr | 0.893747 | 0.906572 | 0.570489 | 0.822254 | 0.826296 | 0.881560 | **0.921495** | 0.869228 |
| pc1 | 0.852119 | 0.853077 | 0.858608 | 0.828996 | 0.820008 | 0.866325 | **0.891588** | 0.863233 |
| pc3 | 0.810311 | 0.784672 | 0.783831 | 0.768438 | 0.771759 | 0.804479 | **0.832408** | 0.809443 |
| pc4 | 0.944764 | 0.940799 | 0.914900 | 0.906347 | 0.936765 | 0.931286 | **0.949946** | 0.900752 |
| pendigits | 0.999740 | 0.999687 | 0.995932 | **0.999850** | 0.999691 | 0.999785 | 0.999748 | 0.999088 |
| PhishingWebsites | 0.996882 | 0.996479 | **0.998190** | 0.994417 | 0.997134 | 0.996805 | 0.997615 | 0.993856 |
| phoneme | 0.956543 | 0.948168 | **0.975103** | 0.952913 | 0.938565 | 0.956949 | 0.963998 | 0.933545 |
| qsar-biodeg | 0.916158 | 0.924529 | 0.892345 | 0.911174 | 0.916804 | 0.918103 | **0.934609** | 0.893489 |
| satimage | 0.992141 | 0.990975 | 0.989906 | 0.986944 | 0.990613 | 0.985874 | **0.993552** | 0.986280 |
| segment | 0.994709 | 0.993795 | **0.996624** | 0.994189 | 0.993821 | 0.993989 | 0.994829 | 0.992101 |
| semeion | 0.995507 | 0.968306 | 0.881622 | **0.998254** | 0.996745 | 0.576269 | 0.998174 | 0.957550 |
| sick | **0.997877** | 0.989590 | 0.995089 | 0.976784 | 0.969015 | 0.991121 | 0.995715 | 0.929353 |
| spambase | 0.983325 | 0.983168 | 0.943909 | 0.978382 | 0.985056 | 0.981111 | **0.988298** | 0.978240 |
| splice | 0.989898 | 0.990919 | 0.990821 | 0.991318 | 0.990917 | 0.991932 | **0.995071** | 0.972882 |
| steel-plates-fault | 0.959626 | 0.963250 | 0.963949 | 0.955133 | 0.959356 | 0.948021 | **0.966012** | 0.916561 |
| texture | 0.999976 | 0.999956 | 0.988863 | 0.999992 | **0.999999** | 0.996944 | 0.999997 | 0.999441 |
| tic-tac-toe | 0.998605 | 0.999145 | 0.999761 | 0.997548 | 0.999375 | 0.996921 | **0.999904** | 0.899715 |
| vehicle | 0.956404 | 0.944588 | 0.942420 | 0.961117 | **0.963268** | 0.944376 | 0.962549 | 0.923325 |
| vowel | 0.999618 | 0.997520 | 0.999888 | 0.999641 | **0.999966** | 0.999888 | 0.999854 | 0.986644 |
| wall-robot-navigation | **0.999757** | 0.999245 | 0.999105 | 0.998582 | 0.998972 | 0.999104 | 0.999520 | 0.997972 |
| wdbc | 0.994847 | 0.994219 | 0.988992 | **0.998021** | 0.997080 | 0.997234 | 0.997765 | 0.985323 |
| wilt | 0.994235 | 0.994105 | 0.992307 | 0.993080 | 0.994057 | 0.988766 | **0.994455** | 0.991840 |

Table 24 shows the raw results for the meta-learned neural networks using the default hyperparameter configurations.

Table 24: Average test ROC-AUC per dataset for meta-learned neural networks using default hyperparameter configurations across CV folds. Missing datasets are represented by "-".

| Dataset | CARTE | TPBerta | TabPFN | XTab |
|---|---|---|---|---|
| adult | **0.897259** | - | - | - |
| analcatdata_authorship | 0.998103 | - | **1.000000** | 0.997620 |
| analcatdata_dmft | 0.572113 | - | **0.586630** | 0.550627 |
| balance-scale | **0.998116** | - | 0.997656 | 0.895083 |
| bank-marketing | **0.907972** | - | - | - |
| banknote-authentication | **1.000000** | 0.997535 | - | 0.996615 |
| blood-transfusion-service-center | 0.705189 | 0.659754 | **0.752586** | - |
| breast-w | 0.984775 | 0.967673 | **0.994131** | 0.988527 |
| car | **0.992862** | - | - | - |
| churn | **0.920360** | - | - | - |
| climate-model-simulation-crashes | 0.938031 | - | **0.968010** | 0.568735 |
| cmc | **0.730370** | - | - | - |
| cnae-9 | **0.986921** | - | - | - |
| connect-4 | **0.500681** | - | - | - |
| credit-approval | 0.906552 | 0.891294 | **0.932397** | 0.922447 |
| credit-g | 0.700952 | - | **0.768476** | - |
| cylinder-bands | 0.810318 | 0.814857 | **0.886616** | 0.778646 |
| diabetes | 0.755348 | 0.768974 | **0.836120** | 0.822370 |
| dna | 0.981979 | - | - | **0.992857** |
| dresses-sales | **0.591297** | 0.565189 | 0.538916 | 0.585057 |
| electricity | 0.874950 | - | - | **0.900765** |
| eucalyptus | 0.907418 | - | **0.928493** | 0.814121 |
| first-order-theorem-proving | **0.735870** | - | - | 0.721997 |
| GesturePhaseSegmentationProcessed | **0.771707** | - | - | 0.737155 |
| har | - | - | - | **0.999241** |
| ilpd | 0.729851 | 0.672431 | **0.757892** | 0.724427 |
| isolet | 0.995113 | - | - | **0.998455** |
| jm1 | 0.704730 | - | - | **0.721445** |
| jungle_chess_2pcs_raw_endgame_complete | 0.918894 | - | - | **0.965961** |
| kc1 | **0.805108** | - | - | 0.793122 |
| kc2 | 0.826925 | - | **0.850065** | 0.835398 |
| kr-vs-kp | 0.958715 | **0.999107** | - | 0.995940 |
| letter | **0.998939** | - | - | 0.989493 |
| madelon | **0.789929** | - | - | 0.689657 |
| mfeat-factors | 0.794171 | - | - | **0.997867** |
| mfeat-fourier | **0.969911** | - | - | 0.956494 |
| mfeat-karhunen | 0.978967 | - | - | **0.990728** |
| mfeat-morphological | **0.961442** | - | - | 0.948069 |
| mfeat-pixel | 0.759099 | - | - | **0.997478** |
| mfeat-zernike | 0.964453 | - | - | **0.965907** |
| MiceProtein | **0.986177** | - | - | 0.972404 |
| nomao | 0.981817 | - | - | **0.991110** |
| numerai28.6 | 0.521094 | - | - | **0.527797** |
| optdigits | 0.998452 | - | - | **0.999031** |
| ozone-level-8hr | 0.861468 | - | - | **0.915294** |
| pc1 | **0.791339** | - | - | 0.729942 |
| pc3 | 0.784448 | 0.683751 | - | **0.816464** |
| pc4 | **0.907759** | 0.699487 | - | 0.888728 |
| pendigits | **0.999522** | - | - | 0.999222 |
| PhishingWebsites | **0.991886** | - | - | 0.987949 |
| phoneme | **0.932082** | 0.798855 | - | 0.911417 |
| qsar-biodeg | 0.914703 | 0.817997 | - | **0.919134** |
| satimage | 0.982299 | - | - | **0.982955** |
| segment | **0.992163** | - | - | 0.974072 |
| semeion | 0.983218 | - | - | **0.989977** |
| sick | **0.991907** | - | - | 0.950283 |
| spambase | 0.748573 | - | - | **0.982966** |
| splice | 0.701980 | - | - | **0.991116** |
| steel-plates-fault | **0.925718** | - | - | 0.848468 |
| texture | 0.993709 | - | - | **0.999521** |
| tic-tac-toe | 0.861176 | 0.958328 | **0.996086** | 0.744202 |
| vehicle | 0.929483 | - | **0.970556** | 0.893891 |
| vowel | **0.995589** | - | - | 0.812581 |
| wall-robot-navigation | **0.998981** | - | - | 0.986489 |
| wdbc | 0.993948 | - | **0.996298** | 0.984744 |
| wilt | **0.994112** | 0.960758 | - | 0.979966 |

Lastly, Table 25 shows the raw results of AutoGluon using the default settings.

Table 25: Average test ROC-AUC per dataset for AutoGluon using default configurations across CV folds.

| Dataset | AutoGluon |
|---|---|
| adult | **0.931179** |
| analcatdata_authorship | **0.999782** |
| analcatdata_dmft | **0.584732** |
| balance-scale | **0.594936** |
| bank-marketing | **0.939889** |
| banknote-authentication | **0.999957** |
| Bioresponse | **0.884276** |
| blood-transfusion-service-center | **0.741962** |
| breast-w | **0.992231** |
| car | **0.999593** |
| churn | **0.922201** |
| climate-model-simulation-crashes | **0.957745** |
| cmc | **0.691344** |
| cnae-9 | **0.997878** |
| connect-4 | **0.936000** |
| credit-approval | **0.935450** |
| credit-g | **0.783286** |
| cylinder-bands | **0.900459** |
| diabetes | **0.821997** |
| dna | **0.994904** |
| dresses-sales | **0.586043** |
| electricity | **0.987262** |
| eucalyptus | **0.754274** |
| first-order-theorem-proving | **0.830805** |
| GesturePhaseSegmentationProcessed | **0.920355** |
| har | **0.999938** |
| ilpd | **0.737184** |
| Internet-Advertisements | **0.984077** |
| isolet | **0.999636** |
| jm1 | **0.764863** |
| jungle_chess_2pcs_raw_endgame_complete | **0.992186** |
| kc1 | **0.821507** |
| kc2 | **0.812567** |
| kr-vs-kp | **0.999619** |
| letter | **0.999901** |
| madelon | **0.925627** |
| mfeat-factors | **0.999142** |
| mfeat-fourier | **0.984642** |
| mfeat-karhunen | **0.998693** |
| mfeat-morphological | **0.969200** |
| mfeat-pixel | **0.998731** |
| mfeat-zernike | **0.982779** |
| MiceProtein | **0.899990** |
| nomao | **0.996397** |
| numerai28.6 | **0.527789** |
| optdigits | **0.999670** |
| ozone-level-8hr | **0.927357** |
| pc1 | **0.876676** |
| pc3 | **0.849770** |
| pc4 | **0.952137** |
| pendigits | **0.999684** |
| PhishingWebsites | **0.997256** |
| phoneme | **0.966521** |
| qsar-biodeg | **0.931279** |
| satimage | **0.992096** |
| segment | **0.996333** |
| semeion | **0.998341** |
| sick | **0.997864** |
| spambase | **0.989571** |
| splice | **0.995584** |
| steel-plates-fault | **0.971070** |
| texture | **0.999996** |
| tic-tac-toe | **0.999951** |
| vehicle | **0.958256** |
| vowel | **0.999641** |
| wall-robot-navigation | **0.898793** |
| wdbc | **0.992978** |
| wilt | **0.994524** |

# E DATASETS

In Table 26 we show a summary of all the OpenMLCC18 datasets used in this study.

Table 26: Summary of OpenML-CC18 Datasets with Feature and Class Frequency Statistics.

| Dataset ID | Dataset Name | Number of Instances | Number of Features | Numerical Features | Categorical Features | Binary Features | Number of Classes | Min-Max Class Freq |
|---|---|---|---|---|---|---|---|---|
| 3 | kr-vs-kp | 3196 | 37 | 0 | 37 | 35 | 2 | 0.91 |
| 6 | letter | 20000 | 17 | 16 | 1 | 0 | 26 | 0.90 |
| 11 | balance-scale | 625 | 5 | 4 | 1 | 0 | 3 | 0.17 |
| 12 | mfeat-factors | 2000 | 217 | 216 | 1 | 0 | 10 | 1.00 |
| 14 | mfeat-fourier | 2000 | 77 | 76 | 1 | 0 | 10 | 1.00 |
| 15 | breast-w | 699 | 10 | 9 | 1 | 1 | 2 | 0.53 |
| 16 | mfeat-karhunen | 2000 | 65 | 64 | 1 | 0 | 10 | 1.00 |
| 18 | mfeat-morphological | 2000 | 7 | 6 | 1 | 0 | 10 | 1.00 |
| 22 | mfeat-zernike | 2000 | 48 | 47 | 1 | 0 | 10 | 1.00 |
| 23 | cmc | 1473 | 10 | 2 | 8 | 3 | 3 | 0.53 |
| 28 | optdigits | 5620 | 65 | 64 | 1 | 0 | 10 | 0.97 |
| 29 | credit-approval | 690 | 16 | 6 | 10 | 5 | 2 | 0.80 |
| 31 | credit-g | 1000 | 21 | 7 | 14 | 3 | 2 | 0.43 |
| 32 | pendigits | 10992 | 17 | 16 | 1 | 0 | 10 | 0.92 |
| 37 | diabetes | 768 | 9 | 8 | 1 | 1 | 2 | 0.54 |
| 38 | sick | 3772 | 30 | 7 | 23 | 21 | 2 | 0.07 |
| 44 | spambase | 4601 | 58 | 57 | 1 | 1 | 2 | 0.65 |
| 46 | splice | 3190 | 61 | 0 | 61 | 0 | 3 | 0.46 |
| 50 | tic-tac-toe | 958 | 10 | 0 | 10 | 1 | 2 | 0.53 |
| 54 | vehicle | 846 | 19 | 18 | 1 | 0 | 4 | 0.91 |
| 151 | electricity | 45312 | 9 | 7 | 2 | 1 | 2 | 0.74 |
| 182 | satimage | 6430 | 37 | 36 | 1 | 0 | 6 | 0.41 |
| 188 | eucalyptus | 736 | 20 | 14 | 6 | 0 | 5 | 0.49 |
| 300 | isolet | 7797 | 618 | 617 | 1 | 0 | 26 | 0.99 |
| 307 | vowel | 990 | 13 | 10 | 3 | 1 | 11 | 1.00 |
| 458 | analcatdata_authorship | 841 | 71 | 70 | 1 | 0 | 4 | 0.17 |
| 469 | analcatdata_dmft | 797 | 5 | 0 | 5 | 1 | 6 | 0.79 |
| 1049 | pc4 | 1458 | 38 | 37 | 1 | 1 | 2 | 0.14 |
| 1050 | pc3 | 1563 | 38 | 37 | 1 | 1 | 2 | 0.11 |
| 1053 | jm1 | 10885 | 22 | 21 | 1 | 1 | 2 | 0.24 |
| 1063 | kc2 | 522 | 22 | 21 | 1 | 1 | 2 | 0.26 |
| 1067 | kc1 | 2109 | 22 | 21 | 1 | 1 | 2 | 0.18 |
| 1068 | pc1 | 1109 | 22 | 21 | 1 | 1 | 2 | 0.07 |
| 1461 | bank-marketing | 45211 | 17 | 7 | 10 | 4 | 2 | 0.13 |
| 1462 | banknote-authentication | 1372 | 5 | 4 | 1 | 1 | 2 | 0.80 |
| 1464 | blood-transfusion-service-center | 748 | 5 | 4 | 1 | 1 | 2 | 0.31 |
| 1468 | cnae-9 | 1080 | 857 | 856 | 1 | 0 | 9 | 1.00 |
| 1475 | first-order-theorem-proving | 6118 | 52 | 51 | 1 | 0 | 6 | 0.19 |
| 1478 | har | 10299 | 562 | 561 | 1 | 0 | 6 | 0.72 |
| 1480 | ilpd | 583 | 11 | 9 | 2 | 2 | 2 | 0.40 |
| 1485 | madelon | 2600 | 501 | 500 | 1 | 1 | 2 | 1.00 |
| 1486 | nomao | 34465 | 119 | 89 | 30 | 3 | 2 | 0.40 |
| 1487 | ozone-level-8hr | 2534 | 73 | 72 | 1 | 1 | 2 | 0.07 |
| 1489 | phoneme | 5404 | 6 | 5 | 1 | 1 | 2 | 0.42 |
| 1494 | qsar-biodeg | 1055 | 42 | 41 | 1 | 1 | 2 | 0.51 |
| 1497 | wall-robot-navigation | 5456 | 25 | 24 | 1 | 0 | 4 | 0.15 |
| 1501 | semeion | 1593 | 257 | 256 | 1 | 0 | 10 | 0.96 |
| 1510 | wdbc | 569 | 31 | 30 | 1 | 1 | 2 | 0.59 |
| 1590 | adult | 48842 | 15 | 6 | 9 | 2 | 2 | 0.31 |
| 4134 | Bioresponse | 3751 | 1777 | 1776 | 1 | 1 | 2 | 0.84 |
| 4534 | PhishingWebsites | 11055 | 31 | 0 | 31 | 23 | 2 | 0.80 |
| 4538 | GesturePhaseSegmentationProcessed | 9873 | 33 | 32 | 1 | 0 | 5 | 0.34 |
| 6332 | cylinder-bands | 540 | 40 | 18 | 22 | 4 | 2 | 0.73 |
| 23381 | dresses-sales | 500 | 13 | 1 | 12 | 1 | 2 | 0.72 |
| 23517 | numerai28.6 | 96320 | 22 | 21 | 1 | 1 | 2 | 0.98 |
| 40499 | texture | 5500 | 41 | 40 | 1 | 0 | 11 | 1.00 |
| 40668 | connect-4 | 67557 | 43 | 0 | 43 | 0 | 3 | 0.15 |
| 40670 | dna | 3186 | 181 | 0 | 181 | 180 | 3 | 0.46 |
| 40701 | churn | 5000 | 21 | 16 | 5 | 3 | 2 | 0.16 |
| 40966 | MiceProtein | 1080 | 82 | 77 | 5 | 3 | 8 | 0.70 |
| 40975 | car | 1728 | 7 | 0 | 7 | 0 | 4 | 0.05 |
| 40978 | Internet-Advertisements | 3279 | 1559 | 3 | 1556 | 1556 | 2 | 0.16 |
| 40979 | mfeat-pixel | 2000 | 241 | 240 | 1 | 0 | 10 | 1.00 |
| 40982 | steel-plates-fault | 1941 | 28 | 27 | 1 | 0 | 7 | 0.08 |
| 40983 | wilt | 4839 | 6 | 5 | 1 | 1 | 2 | 0.06 |
| 40984 | segment | 2310 | 20 | 19 | 1 | 0 | 7 | 1.00 |
| 40994 | climate-model-simulation-crashes | 540 | 21 | 20 | 1 | 1 | 2 | 0.09 |
| 41027 | jungle_chess_2pcs_raw_endgame_complete | 44819 | 7 | 6 | 1 | 0 | 3 | 0.19 |

Table 27: Dataset coverage on OpenML CC-18 after excluding four image datasets (IDs: 40923, 554, 40996, 40927). Default pool = 68 datasets.

| Method | Used | Missing | Primary reason for missing datasets |
|---|---|---|---|
| AutoGluon | 68 | 0 | – |
| CatBoost | 68 | 0 | – |
| LightGBM | 68 | 0 | – |
| XGBoost | 68 | 0 | – |
| RealMLP | 68 | 0 | – |
| TabM | 68 | 0 | – |
| ResNet | 68 | 0 | – |
| FT-Transformer | 68 | 0 | – |
| MLP | 67 | 1 | Memory constraint on one dataset |
| ModernNCA | 65 | 3 | Memory constraints |
| SAINT | 61 | 7 | Memory constraints |
| TabNet | 62 | 6 | Memory constraints |
| CARTE | 62 | 6 | Memory constraints |
| Mitra | 49 | 19 | Method limits: $\leq$10,000 samples, $\leq$500 features, $\leq$10 classes |
| LimiX | 62 | 6 | Memory constraints |
| TabICL | 68 | 0 | – |
| TP-BERTa | 15 | 53 | Memory constraints |
| TabPFN | 17 | 51 | Method limits: $\leq$1000 samples, $\leq$100 features, $\leq$10 classes |
| TabPFNv2 | 49 | 19 | Method limits: $\leq$10,000 samples, $\leq$500 features, $\leq$10 classes |
| XTab | 55 | 13 | Excluded due to pretraining overlap |

We evaluate all methods on the OpenML CC-18 benchmark. Four very large image datasets are excluded a priori due to memory issues affecting most baselines: Devnagari-Script (OpenML ID 40923), MNIST (554), Fashion-MNIST (40996), and CIFAR-10 (40927). This leaves 68 datasets as the default pool. Some methods are run on fewer datasets due to memory limits, method-specific constraints, or pretraining overlap. Table 27 reports the coverage per method and the reason for any missing datasets.

# F ANALYSIS OF DATASET CHARACTERISTICS: INSTANCES AND FEATURES

To analyze the relationship between dataset size and the performance of different methods, we categorize datasets based on two key attributes: the number of instances and the number of features.

- **Instance-based Categorization:**

  - Datasets with 1000 or fewer instances.

  - Datasets with 1001 to 5000 instances.

  - Datasets with 5001 to 10000 instances.

  - Datasets with 10001 to 50000 instances.

  - Datasets with more than 50000 instances.

- **Feature-based Categorization:** Within each instance-based group, datasets are further divided based on the number of features:

  - Datasets with 100 or fewer features.

  - Datasets with 101 to 500 features.

  - Datasets with 501 to 1000 features.

  - Datasets with more than 1000 features.

- **Unavailable Results:** Having split the datasets into these groups, we note the ones in which no dataset belongs:

  - Datasets with instances between 5001 and 10000, and features between 100 and 500.

  - Datasets with instances between 5001 and 10000, and features greater than 1000.

  - Datasets with instances between 10001 and 50000, and features greater than 1000.

  - For datasets with more than 50000 instances, we only have results for datasets with 100 or fewer features.

  - For datasets with fewer than 1000 instances, we only have results for datasets with 100 or fewer features.

For the analysis, we present boxplots and critical difference diagrams, if the number of datasets is at least 10 for meaningful analysis. If the number of datasets in a group is fewer than 10, we use tabular results instead of boxplots or critical difference diagrams.

For a complementary, more fine-grained view in terms of absolute performance differences, Subsection F.6 reports mean and median ΔROC-AUC values (with confidence intervals) both versus the best GBDT per dataset family and versus the best method per dataset across all families.

## F.1 DATASETS WITH FEWER THAN 1000 INSTANCES

In this section, we focus on datasets with fewer than 100 features and fewer than 1000 instances, resulting in a total of 18 datasets used in our study. Consequently, most methods in Figure 29 are evaluated on 18 datasets. However, there are a few exceptions: TabPFN, TabPFNv2 and Mitra are incompatible with one dataset, "vowel," due to it containing more than 10 classes; XTab excludes 2 datasets that were part of its pretraining phase; and TP-Berta encounters memory limitations on 10 out of the 18 datasets, reducing its coverage.

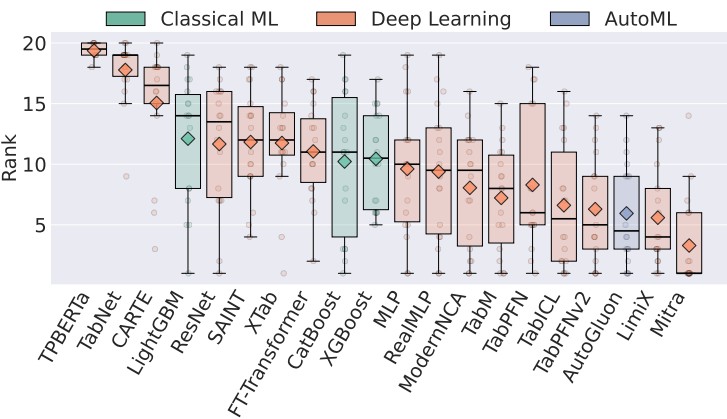

Figure 29: Distribution of ranks for all the methods in the small data domain. The boxplot illustrates the rank spread, with medians represented by black lines, means represented by diamonds and whiskers showing the range.

Figure 29 reveals that Mitra achieves the strongest overall performance, closely followed by LimiX and AutoGluon. Among feedforward networks, TabM, ModernNCA and RealMLP rank well, though TabM reaches a lower rank, while ModernNCA has a better mean rank compared to RealMLP. Among the other dataset-specific neural networks, FT-Transformer and SAINT perform comparably. Interestingly, MLP-like methods also show a lower median rank than the classical CatBoost, LightGBM and XGBoost. By contrast, TabNet and the fine-tuning–based models generally exhibit the weakest performance.

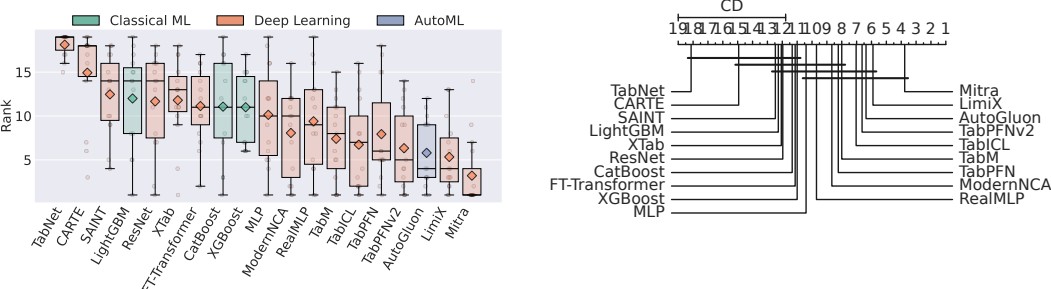

Figure 30: **Left:** Distribution of ranks for all the methods in the small data domain. The boxplot illustrates the rank spread, with medians represented by black lines, means represented by diamonds and whiskers showing the range. **Right:** Comparative analysis of all the methods.

Similarly, Figure 30 shows a boxplot on the left, evaluated on the same datasets but excluding TP-BERTa—and a critical difference diagram on the right. A clear pattern emerges: in the small-data domain, in-context learning methods, are highly competitive, followed by dataset-specific neural architectures (e.g., TabM, RealMLP, ModernNCA and MLP with PLR embeddings), surpassing CatBoost, LightGBM and XGBoost.

## F.2   DATASETS WITH 1,000 TO 5,000 INSTANCES

Following the previous analysis, we now examine datasets with 1,000–5,000 instances and fewer than 100 features. The results are shown in Figure 31. Similar to the small-data setting, ICL models dominate, outperforming even AutoGluon. A notable shift, however, is the increase in performance by CatBoost, which rises to seventh place overall, just behind TabM in third. Furthermore, dataset-specific neural networks continue to outperform fine-tuned networks, with TabM, ModernNCA and MLP with PLR embeddings standing out for their strong performance. They achieve better median ranks and narrower interquartile ranges than XGBoost and LightGBM. Another interesting obser-

vation, is the shift in rank from Mitra. While in the small data domain it performed best, now it performs worst relative to other ICL models.

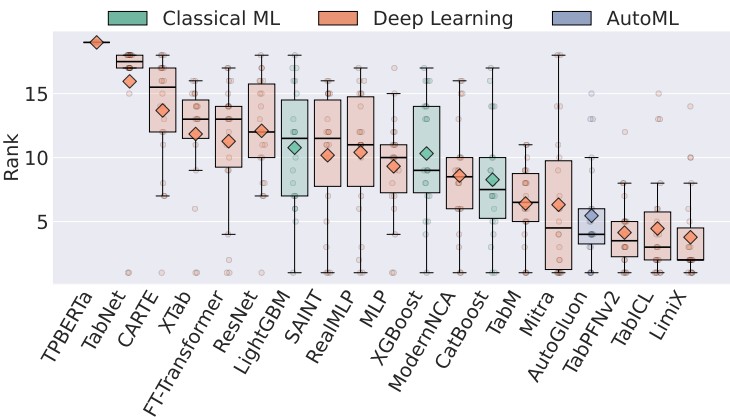

Figure 31: Distribution of ranks for all the methods in the datasets with 1000 to 5000 instances, and less than 100 features. The boxplot illustrates the rank spread, with medians represented by black lines, means represented by diamonds and whiskers showing the range.

In Figure 32, we exclude TP-BERTa again to ensure a reasonable number of common datasets, resulting in a total of 19 datasets. The left plot tells a similar story, with XGBoost now achieving slightly better median rank as the MLP with PLR embeddings. The right plot presents a critical difference diagram, showing LimiX, TabICL, TabPFNv2, and AutoGluon as the top-performing methods. Overall, GBDT methods increase in rank with the increase of dataset size, however, ICL models as well as TabM continue to dominate them.

For the remaining dataset categorization groups, we present only tabular results due to the limited number of datasets in these categories. Table 28 provides the results for datasets with 1000 to 5000 instances and 100 to 500 features. Similarly, Table 29 summarizes the performance for datasets in the 500 to 1000 features range, while Table 30 presents results for datasets with more than 1000 features. Detailed results for all other dataset categorization groups can be found below.

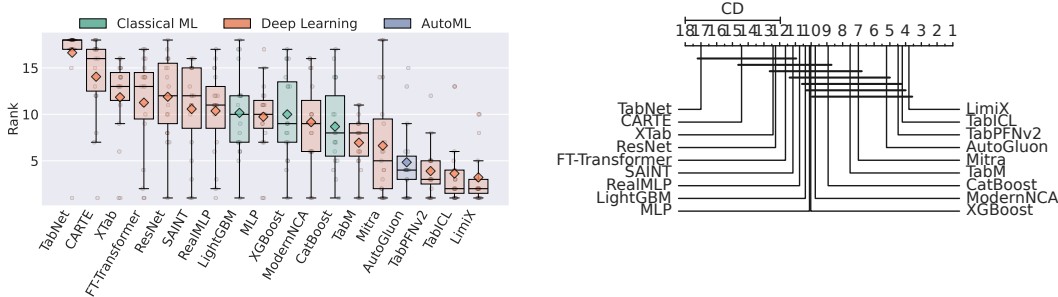

Figure 32: **Left:** Distribution of ranks for all the methods in the common datasets with instances between 1000 and 5000, and features fewer than 100. The boxplot illustrates the rank spread, with medians represented by red lines and whiskers showing the range. **Right:** Comparative analysis of all the methods.

Table 28: Classifier Performance for Instance Range: 1000-5000 and Feature Range: 100-500

| Dataset | dna | mfeat-factors | mfeat-pixel | semeion |
|---|---|---|---|---|
| AutoGluon | 0.995385 | 0.999350 | 0.999403 | 0.998506 |
| CARTE | 0.986120 | 0.996064 | 0.996175 | 0.993378 |
| CatBoost | 0.995028 | 0.998910 | 0.999422 | 0.998687 |
| FT-Transformer | 0.990937 | 0.999015 | 0.997451 | 0.995548 |
| LimiX | 0.994527 | 0.999733 | 0.999642 | **0.999240** |
| LightGBM | 0.994942 | 0.999125 | 0.999131 | 0.997646 |
| MLP | 0.992220 | 0.998875 | 0.998674 | 0.997350 |
| Mitra | 0.981938 | 0.999225 | 0.998814 | 0.995907 |
| ModernNCA | 0.993932 | 0.999478 | 0.999256 | 0.998727 |
| RealMLP | 0.994111 | 0.999625 | 0.999492 | 0.998976 |
| ResNet | 0.992543 | 0.999472 | 0.998690 | 0.997689 |
| SAINT | 0.992473 | 0.999385 | 0.999217 | 0.997630 |
| TabICL | 0.994123 | **0.999808** | **0.999664** | 0.999033 |
| TabM | 0.994505 | 0.999700 | 0.999478 | 0.998425 |
| TabNet | 0.991448 | 0.998125 | 0.998200 | 0.994019 |
| TabPFNv2 | **0.995658** | 0.999650 | 0.999503 | 0.998288 |
| XGBoost | 0.995278 | 0.999004 | 0.999378 | 0.998272 |
| XTab | 0.992479 | 0.998443 | 0.998642 | 0.997064 |

Table 29: Classifier Performance for Instance Range: 1000-5000 and Feature Range: 500-1000

| Dataset | cnae-9 | madelon |
|---|---|---|
| AutoGluon | 0.998524 | 0.932817 |
| CARTE | 0.990151 | 0.836760 |
| CatBoost | 0.996316 | 0.937562 |
| FT-Transformer | 0.994497 | 0.747391 |
| LimiX | 0.997695 | **0.965107** |
| LightGBM | 0.984404 | 0.924095 |
| MLP | 0.996716 | 0.883991 |
| ModernNCA | **0.998881** | 0.851408 |
| RealMLP | 0.997569 | 0.930302 |
| ResNet | 0.997106 | 0.605018 |
| TabICL | 0.997840 | 0.711538 |
| TabM | 0.998100 | 0.809941 |
| TabNet | - | 0.630669 |
| XGBoost | 0.997454 | 0.932249 |
| XTab | - | 0.845746 |

Table 30: Classifier Performance for Instance Range: 1000-5000 and Feature Range: > 1000

| Dataset | Bioresponse | Internet-Advertisements |
|---|---|---|
| AutoGluon | **0.888693** | 0.985963 |
| CatBoost | 0.885502 | 0.979120 |
| FT-Transformer | 0.820159 | 0.974513 |
| LimiX | 0.886859 | 0.984687 |
| LightGBM | 0.886734 | 0.980094 |
| MLP | 0.825631 | - |
| ModernNCA | - | 0.983818 |
| RealMLP | 0.859065 | 0.973810 |
| ResNet | 0.850801 | 0.974187 |
| TabICL | 0.885075 | **0.989308** |
| TabM | 0.876671 | 0.985640 |
| XGBoost | 0.888615 | 0.982276 |

## F.3 DATASETS WITH 5,000 TO 10,000 INSTANCES

Table 31: Classifier Performance for Instance Range: 5000-10000 and Feature Range: ≤ 100

| Dataset | GPhaseSeg | first-ord-TP | optdigits | phoneme | satimage | texture | wall-rob-nav |
|---|---|---|---|---|---|---|---|
| AutoGluon | 0.936667 | **0.835425** | 0.999925 | 0.973342 | 0.993557 | 0.999998 | **0.999993** |
| CARTE | 0.798024 | 0.764092 | 0.999112 | 0.948702 | 0.988038 | 0.999541 | 0.999505 |
| CatBoost | 0.916674 | 0.831775 | 0.999844 | 0.968024 | 0.991978 | 0.999948 | 0.999990 |
| FT-Transformer | 0.895166 | 0.796707 | 0.999616 | 0.965071 | 0.993516 | 0.999983 | 0.999900 |
| LimiX | **0.957290** | 0.835198 | 0.999961 | **0.980179** | **0.995246** | - | 0.999984 |
| LightGBM | 0.920034 | 0.833949 | 0.999818 | 0.965147 | 0.991309 | 0.999890 | 0.999968 |
| MLP | 0.911434 | 0.798812 | 0.999794 | 0.967617 | 0.992308 | 0.999991 | 0.999689 |
| Mitra | 0.809182 | 0.789747 | 0.999456 | 0.955580 | 0.984750 | - | 0.999976 |
| ModernNCA | 0.956087 | 0.825830 | 0.999886 | 0.977289 | 0.993801 | 0.999983 | 0.999850 |
| RealMLP | 0.901441 | 0.795637 | 0.999968 | 0.966456 | 0.993034 | 0.999999 | 0.998720 |
| ResNet | 0.914196 | 0.784636 | 0.999927 | 0.963591 | 0.991995 | 0.999999 | 0.999042 |
| SAINT | 0.919006 | 0.802392 | 0.999841 | 0.960382 | 0.992630 | 0.999976 | 0.999844 |
| TabPFNv2 | 0.936548 | 0.825502 | 0.999897 | 0.973546 | 0.995122 | 0.999963 | 0.999936 |
| TabM | 0.933828 | 0.818255 | 0.999939 | 0.971200 | 0.994291 | 0.999997 | 0.999912 |
| TabNet | 0.850596 | 0.774094 | 0.998871 | 0.956279 | 0.987482 | 0.999763 | 0.997585 |
| TabICL | 0.951408 | 0.834629 | **0.999989** | 0.977493 | 0.993373 | **1.000000** | 0.999610 |
| XGBoost | 0.916761 | 0.834883 | 0.999855 | 0.967421 | 0.992114 | 0.999940 | 0.999981 |
| XTab | 0.886960 | 0.798803 | 0.999712 | 0.961749 | 0.992918 | 0.999962 | 0.999846 |

Table 32: Classifier Performance for Instance Range: 5000-10000 and Feature Range: 500-1000

| Dataset | isolet |
|---|---|
| AutoGluon | 0.999744 |
| CatBoost | 0.999389 |
| FT-Transformer | 0.998817 |
| LightGBM | 0.999401 |
| MLP | 0.998295 |
| ModernNCA | 0.998432 |
| RealMLP | 0.999635 |
| ResNet | 0.999401 |
| SAINT | - |
| TabICL | 0.999570 |
| TabM | **0.999750** |
| TabNet | 0.998813 |
| XGBoost | 0.999488 |

Table 33: Classifier Performance for Instance Range: 10000-50000 and Feature Range: $\leq 100$

| Dataset | Phishing | Adult | BankMkt | Elec | JM1 | JngChess | Letter | PenDigits |
|---|---|---|---|---|---|---|---|---|
| AutoGluon | 0.997572 | **0.931792** | 0.941273 | 0.987260 | 0.770272 | 0.999278 | 0.999934 | 0.999725 |
| CARTE | 0.994582 | 0.902677 | 0.924664 | 0.909407 | 0.728512 | 0.973383 | 0.999440 | 0.999468 |
| CatBoost | 0.996482 | 0.930747 | 0.938831 | 0.980993 | 0.756611 | 0.976349 | 0.999854 | 0.999752 |
| FT-Transformer | 0.996760 | 0.914869 | 0.938198 | 0.963076 | 0.709321 | 0.999975 | 0.999919 | 0.999703 |
| LimiX | 0.997750 | 0.930281 | **0.943913** | **0.995629** | 0.772956 | 0.993452 | 0.999820 | 0.999820 |
| LightGBM | 0.997542 | 0.931261 | 0.938470 | 0.989807 | 0.753206 | 0.977605 | 0.999825 | 0.999735 |
| MLP | 0.996991 | 0.928689 | 0.937054 | 0.969201 | 0.715620 | 0.999965 | 0.999894 | 0.999705 |
| ModernNCA | 0.998275 | 0.930138 | 0.936903 | 0.995199 | 0.759996 | **0.999996** | **0.999964** | 0.999788 |
| RealMLP | 0.997208 | 0.923327 | 0.937031 | 0.961467 | 0.713988 | 0.999774 | 0.999914 | 0.999659 |
| ResNet | 0.996975 | 0.913790 | 0.935740 | 0.960658 | 0.720444 | 0.999956 | 0.999926 | 0.999638 |
| SAINT | 0.996746 | 0.920246 | 0.936560 | 0.967012 | 0.719464 | 0.999926 | 0.999853 | 0.999782 |
| TabICL | **0.998342** | 0.914430 | 0.940210 | 0.970809 | **0.784425** | 0.975471 | 0.999957 | **0.999852** |
| TabM | 0.997636 | 0.919662 | 0.941872 | 0.968760 | 0.751557 | 0.999985 | 0.999943 | 0.999739 |
| TabNet | 0.996196 | 0.882450 | 0.887319 | 0.938656 | 0.674043 | 0.991981 | 0.999606 | 0.999753 |
| XGBoost | 0.997425 | 0.930482 | 0.938384 | 0.987790 | 0.759652 | 0.974087 | 0.999819 | 0.999703 |
| XTab | 0.996896 | - | - | 0.966899 | 0.727984 | 0.999950 | 0.999859 | 0.999751 |

Table 34: Classifier Performance for Instance Range: 10000-50000 and Feature Range: 100-500

| Dataset | nomao |
|---|---|
| AutoGluon | 0.996892 |
| CatBoost | 0.996439 |
| FT-Transformer | 0.990908 |
| LightGBM | 0.996835 |
| LimiX | **0.997403** |
| MLP | 0.986577 |
| RealMLP | 0.989803 |
| ResNet | 0.993048 |
| TabICL | 0.996055 |
| TabM | 0.994828 |
| XGBoost | 0.996676 |
| XTab | 0.992727 |

Table 35: Classifier Performance for Instance Range: 10000-50000 and Feature Range: 500-1000

| Dataset | har |
|---|---|
| AutoGluon | 0.999958 |
| CatBoost | 0.999941 |
| FT-Transformer | 0.999685 |
| LightGBM | 0.999959 |
| LimiX | **0.999993** |
| MLP | 0.999783 |
| RealMLP | 0.999959 |
| ResNet | 0.999921 |
| TabICL | 0.999913 |
| TabM | 0.999966 |
| TabNet | 0.999515 |
| XGBoost | 0.999960 |

F.5 DATASETS WITH MORE THAN 50,000 INSTANCES

Table 36: Classifier Performance for Instance Range: $> 50000$ and Feature Range: $\leq 100$

| Dataset | connect-4 | numerai28.6 |
|---|---|---|
| AutoGluon | 0.934636 | 0.530150 |
| CARTE | - | 0.514361 |
| CatBoost | 0.921050 | 0.529404 |
| FT-Transformer | 0.901170 | **0.530315** |
| LightGBM | 0.932440 | 0.529077 |
| MLP | 0.927373 | 0.525920 |
| ModernNCA | 0.936124 | 0.508863 |
| RealMLP | 0.928258 | 0.529534 |
| ResNet | 0.933333 | 0.528012 |
| SAINT | - | 0.525822 |
| TabICL | 0.897904 | 0.526838 |
| TabM | **0.941654** | 0.529336 |
| XGBoost | 0.931952 | 0.529457 |
| XTab | - | 0.528062 |

F.6 ABSOLUTE DELTA ROC-AUC VS GBDT AND BEST METHODS

We complement the rank-based analysis with absolute performance differences in Tables 37–40. We group datasets into three families based on the number of instances: *small* ($n \leq 1000$), *medium* ($1000 < n \leq 10000$), and *large* ($n > 10000$). For each dataset family (Tables 37–39), we report the median and mean $\Delta$ROC-AUC of each method relative to the best GBDT (CatBoost, XGBoost, or LightGBM) on that dataset, where positive values indicate better performance than the top GBDT. The tables also include 95% confidence intervals for the mean $\Delta$ROC-AUC across datasets and the number of datasets per method. In addition, Table 40 summarizes, over all datasets, the median and mean *absolute* ROC-AUC difference to the best method per dataset (lower is better), again with 95% confidence intervals for the mean and the number of datasets.

Taken together, these tables show that several methods improve over strong GBDT baselines while remaining very close in absolute performance. On small datasets, Mitra achieves the largest average gain over the best GBDT (median $\Delta$AUC $\approx 0.008$), with TabPFNv2, LimiX, TabPFN, Auto-Gluon, TabICL, and ModernNCA also exhibiting small positive or near-zero deltas. On medium datasets, LimiX shows the highest median AUC gain with an approximate value of 0.002, followed by TabICL, TabPFNv2, AutoGluon, and TabM exhibiting positive median $\Delta$AUC. On large datasets, LimiX manages to again have the largest average gain over the best GBDT (median $\Delta$AUC $\approx 0.0005$), followed by AutoGluon, and ModernNCA.

Table 37: Median and mean $\Delta$AUC vs best GBDT for 18 small ($n \leq 1000$) datasets (positive is better). 95% CI is for the mean over datasets.

| Method | Median $\Delta$AUC | Mean $\Delta$AUC | 95% CI (mean) |
|---|---|---|---|
| Mitra | 0.008147 | 0.019731 | [0.007497, 0.031965] |
| TabPFNv2 | 0.002020 | 0.001240 | [-0.004512, 0.006992] |
| LimiX | 0.001665 | 0.004488 | [-0.002136, 0.011113] |
| TabPFN | 0.001569 | -0.005307 | [-0.017590, 0.006976] |
| AutoGluon | 0.000844 | 0.003054 | [-0.001641, 0.007750] |
| TabICL | 0.000293 | 0.002176 | [-0.004466, 0.008818] |
| ModernNCA | 0.000140 | 0.000148 | [-0.005568, 0.005864] |
| CatBoost | -0.000084 | -0.003930 | [-0.007635, -0.000225] |
| MLP | -0.000108 | -0.005469 | [-0.015744, 0.004806] |
| TabM | -0.000117 | 0.000251 | [-0.004191, 0.004692] |
| XGBoost | -0.000646 | -0.001837 | [-0.002870, -0.000804] |
| RealMLP | -0.000698 | -0.007755 | [-0.018028, 0.002518] |
| ResNet | -0.001893 | -0.007842 | [-0.015140, -0.000544] |
| FT-Transformer | -0.002285 | -0.003202 | [-0.007532, 0.001127] |
| SAINT | -0.002420 | -0.007840 | [-0.014933, -0.000747] |
| AutoGluon (HPO) | -0.002935 | -0.009675 | [-0.016683, -0.002666] |
| XTab | -0.003731 | -0.007840 | [-0.015051, -0.000629] |
| LightGBM | -0.004467 | -0.007792 | [-0.012630, -0.002955] |
| CARTE | -0.013601 | -0.016185 | [-0.026055, -0.006316] |
| TabNet | -0.053581 | -0.050038 | [-0.069705, -0.030372] |
| TPBERTa | -0.073517 | -0.073014 | [-0.112389, -0.033638] |

Table 38: Median and mean $\Delta$AUC vs best GBDT for 38 medium ($1000 < n \leq 10000$) datasets (positive is better). 95% CI is for the mean over datasets.

| Method | Median $\Delta$AUC | Mean $\Delta$AUC | 95% CI (mean) |
|---|---|---|---|
| LimiX | 0.001711 | 0.005951 | [0.002717, 0.009185] |
| TabICL | 0.000635 | -0.001551 | [-0.013747, 0.010646] |
| TabPFNv2 | 0.000509 | 0.003628 | [0.001235, 0.006022] |
| AutoGluon | 0.000418 | 0.002789 | [0.001169, 0.004408] |
| TabM | 0.000053 | -0.003267 | [-0.010150, 0.003615] |
| CatBoost | 0.000000 | -0.000823 | [-0.001322, -0.000324] |
| Mitra | -0.000014 | -0.002874 | [-0.011889, 0.006141] |
| ModernNCA | -0.000056 | -0.001599 | [-0.006965, 0.003767] |
| XGBoost | -0.000065 | -0.001425 | [-0.002300, -0.000550] |
| AutoGluon (HPO) | -0.000328 | -0.007025 | [-0.017543, 0.003493] |
| MLP | -0.000749 | -0.007210 | [-0.012405, -0.002016] |
| LightGBM | -0.000851 | -0.003453 | [-0.005274, -0.001633] |
| SAINT | -0.001028 | -0.005339 | [-0.008711, -0.001968] |
| RealMLP | -0.001109 | -0.006865 | [-0.010775, -0.002955] |
| XTab | -0.001622 | -0.010070 | [-0.016729, -0.003411] |
| FT-Transformer | -0.002040 | -0.011699 | [-0.022624, -0.000774] |
| ResNet | -0.002486 | -0.014910 | [-0.032201, 0.002381] |
| CARTE | -0.004450 | -0.016450 | [-0.025891, -0.007009] |
| TabNet | -0.009357 | -0.029409 | [-0.047764, -0.011055] |
| TPBERTa | -0.144523 | -0.138857 | [-0.194713, -0.083001] |

Table 39: Median and mean $\Delta$AUC vs best GBDT for 12 large ($n > 10000$) datasets (positive is better). 95% CI is for the mean over datasets.

| Method | Median $\Delta$AUC | Mean $\Delta$AUC | 95% CI (mean) |
|---|---|---|---|
| LimiX | 0.000568 | 0.004439 | [0.000360, 0.008518] |
| AutoGluon | 0.000305 | 0.002979 | [-0.000813, 0.006771] |
| ModernNCA | 0.000227 | 0.000905 | [-0.005515, 0.007324] |
| AutoGluon (HPO) | 0.000030 | 0.002431 | [-0.001445, 0.006306] |
| LightGBM | -0.000001 | -0.000603 | [-0.001647, 0.000441] |
| TabM | -0.000003 | -0.000674 | [-0.006676, 0.005328] |
| XGBoost | -0.000138 | -0.000634 | [-0.001242, -0.000026] |
| TabICL | -0.000413 | -0.004066 | [-0.012188, 0.004057] |
| CatBoost | -0.000455 | -0.002212 | [-0.004377, -0.000047] |
| FT-Transformer | -0.000707 | -0.009091 | [-0.019934, 0.001751] |
| ResNet | -0.001006 | -0.005963 | [-0.014977, 0.003051] |
| XTab | -0.001021 | -0.004797 | [-0.016099, 0.006505] |
| RealMLP | -0.001067 | -0.006090 | [-0.015539, 0.003360] |
| TabNet | -0.001346 | -0.024972 | [-0.047577, -0.002366] |
| MLP | -0.002174 | -0.005519 | [-0.014303, 0.003266] |
| SAINT | -0.002271 | -0.006483 | [-0.017787, 0.004821] |
| CARTE | -0.014167 | -0.019696 | [-0.036377, -0.003016] |

Table 40: Median and mean absolute $\Delta$AUC to the best method per dataset across all datasets (lower is better). 95% CI is for the mean over datasets.

| Method | Median $\Delta$AUC | Mean $\Delta$AUC | 95% CI (mean) |
|---|---|---|---|
| Mitra | 0.000850 | 0.008632 | [0.002236, 0.015029] |
| LimiX | 0.001097 | 0.006970 | [0.003557, 0.010383] |
| AutoGluon | 0.001212 | 0.008490 | [0.005070, 0.011910] |
| TabPFNv2 | 0.001714 | 0.010801 | [0.005702, 0.015900] |
| TabICL | 0.001848 | 0.012391 | [0.004523, 0.020258] |
| TabM | 0.002827 | 0.013261 | [0.007534, 0.018988] |
| ModernNCA | 0.003855 | 0.012627 | [0.007363, 0.017892] |
| AutoGluon (HPO) | 0.003992 | 0.017440 | [0.009862, 0.025018] |
| MLP | 0.004321 | 0.017887 | [0.011010, 0.024764] |
| XGBoost | 0.004654 | 0.012777 | [0.008508, 0.017046] |
| XTab | 0.004676 | 0.021565 | [0.013914, 0.029215] |
| CatBoost | 0.005785 | 0.013273 | [0.008934, 0.017611] |
| FT-Transformer | 0.006419 | 0.020372 | [0.012587, 0.028157] |
| LightGBM | 0.006645 | 0.015481 | [0.010390, 0.020573] |
| RealMLP | 0.006865 | 0.018346 | [0.012218, 0.024474] |
| SAINT | 0.007685 | 0.018387 | [0.011960, 0.024813] |
| ResNet | 0.008750 | 0.022842 | [0.011738, 0.033947] |
| TabPFN | 0.012985 | 0.027254 | [0.011308, 0.043200] |
| CARTE | 0.014986 | 0.029046 | [0.020101, 0.037991] |
| TabNet | 0.021235 | 0.046924 | [0.032363, 0.061485] |
| TPBERTa | 0.133657 | 0.121023 | [0.079278, 0.162768] |

# G  COMPARISON WITH AUTOML METHODS

In our study, we include AutoGluon Erickson et al. (2020), a prominent AutoML library, to compare against the Deep Learning methods. We consider two versions of AutoGluon: one where we perform hyperparameter optimization (HPO) and the officially recommended version configured with `presets=best_quality`. We compare all methods within the Deep Learning family to these versions of AutoGluon.

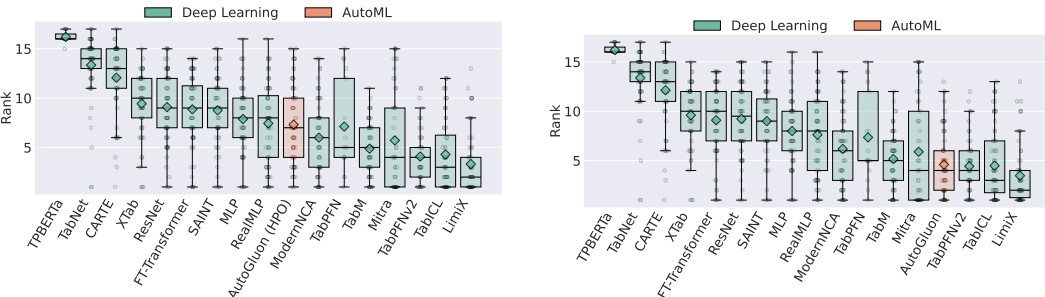

Figure 33: Distribution of ranks for the Deep Learning Models (13 methods) and AutoML (1 method) classifier families. **Left:** AutoGluon with hyperparameter optimization (HPO). **Right:** AutoGluon in its recommended configuration. The boxplot illustrates the rank spread, with medians represented by red lines and whiskers showing the range.

Figure 33 presents boxplots of the rank distributions for all Deep Learning methods compared to AutoGluon. The left-hand side shows results against the HPO-tuned version of AutoGluon. LimiX, TabICL and TabPFNv2 achieve the best overall performance. TabM and ModernNCA also perform competitively with a median rank of 5 and 6 respectively, just behind Mitra, while most other Deep Learning methods rank lower.

The right-hand side compares against the recommended version of AutoGluon and reveals a similar picture. LimiX, TabICL and TabPFNv2 perform best, outperformindAutoGluon too in both mean and median rank, though AutoGluon's interquartile range extends slightly lower compared to TabPFNv2, indicating marginally better performance. LimiX attains the best mean and median rank.

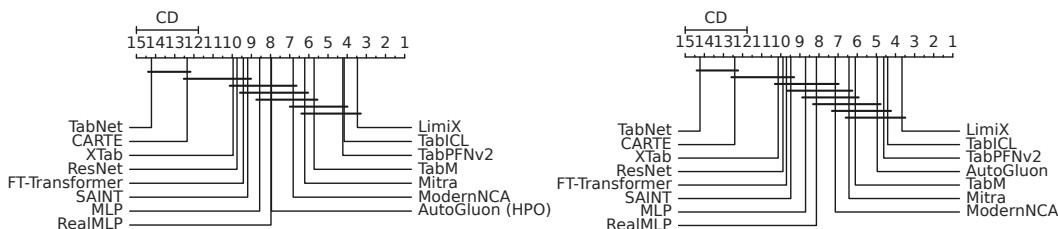

Figure 34: Comparative analysis of Deep learning models against AutoGluon. **Left:** AutoGluon with hyperparameter optimization (HPO). **Right:** AutoGluon in its recommended configuration.

Consistent with our previous analyses, we also present critical difference (CD) diagrams to summarize the ranking comparisons. Figure 34 shows, on the left, the CD diagram comparing AutoGluon with HPO against the Deep Learning methods across all datasets, and on the right, the diagram for AutoGluon with its recommended configuration.

On the left, LimiX, TabICL and TabPFNv2 achieve the best average rank of 3.5, and 4.25 respectively followed by TabM and then Mitra. The diagram indicates that differences among LimiX, TabICL, TabPFNv2, and Mitra are not statistically significant; however, LimiX significantly outperforms all remaining methods.

On the right, again LimiX is the top-ranked method, followed closely by TabICL, TabPFNv2 and AutoGluon. Similarly, with the exception of TabICL, TabPFNv2, AutoGluon, TabM and Mitra, LimiX outperforms all the other baselines significantly.

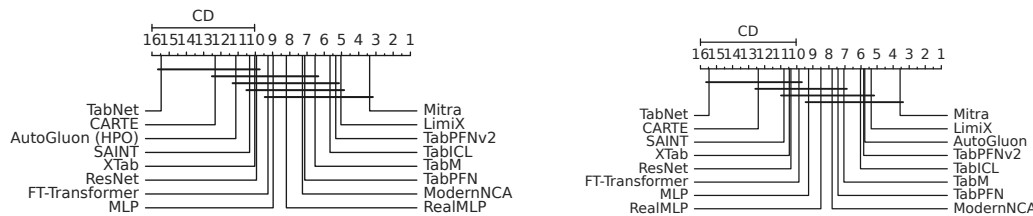

Figure 35: Comparative analysis of Deep learning models against AutoGluon in the small data domain (number of instances ≤ 1000). **Left:** AutoGluon with hyperparameter optimization (HPO). **Right:** AutoGluon in its recommended configuration.

We further perform the same analysis in the small-data setting, defined as datasets with ≤ 1000 instances, which allows us to include TabPFN in the comparison. The results are shown in Figure 35.

In the left diagram, Mitra leads the rankings, followed by LimiX, TabPFNv2 and TabICL. With the exception of TabNet and Carte all the other methods outperform the HPO-tuned version of Auto-Gluon. In contrast, the right diagram shows that AutoGluon with its recommended configuration ranks third, just behind Mitra and LimiX. Overall, Mitra performs best in the small data domain.

## H  COST VS. EFFICIENCY RELATION OF VARIOUS MODEL FAMILIES

To observe what is the cost vs. efficiency relation of various model families, we plot the intra-search space normalized Average Distance to the Maximum (ADTM) Wistuba et al. (2016) in Figure 36, illustrating how quickly each method converges to its best solution during the HPO process.

The plot shows that LightGBM and XGBoost are the fastest, reaching nearly optimal performance within just 2 hours. The ResNet and MLP architecture also demonstrate notable speed, followed closely by CatBoost.

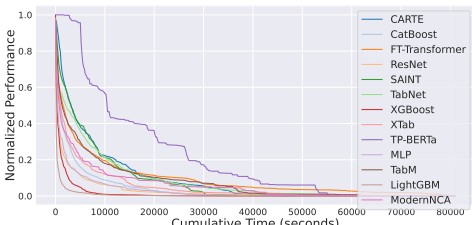

Figure 36: Intra-search space normalized average distance to the maximum over cumulative training time (seconds).

Overall, the gradient boosting methods LightGBM and XGBoost converge faster than the deep learning models. XTab, which shares the same transformer architecture as FT-Transformer, exhibits

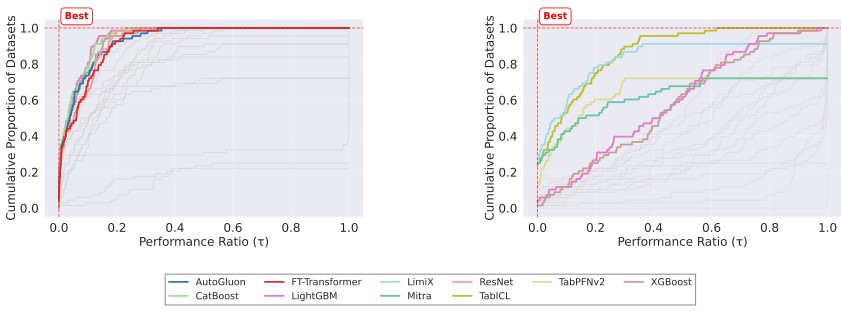

Figure 37: **Left:** Performance profiles based on inference time. **Right:** Performance profiles based on total time. Steeper curves indicate better overall performance and efficiency across datasets.

quicker convergence, likely due to its static architecture, while the FT-Transformer's architectural components were also tuned. ModernNCA also shows that it reaches good performance fast, however, needs around 11 hours to get to the optimum. On the other hand, TP-BERTa is the slowest to converge, likely due to the high computational demands of its BERT-like architecture.

In Figure 37, we show the performance profiles of the models considered. We first normalize the performance values and the logarithmic time values.

Formally, let $P_i^{(j)}$ denote the performance of an algorithm $i$ on dataset $j$, and $T_i^{(j)}$ the corresponding executing time. We define $m_*^{(j)} = \max_i \, P_i^{(j)}$ to be the best performance achieved across all algorithms, and $t_\dagger^{(j)} = \max_i \, T_i^{(j)}$ as the longest runtime observed. Next, we compute for each algorithm $i$ and dataset $j$ the performance gap $\mathrm{gap}_i^{(j)} = (m_*^{(j)} - P_i(j))/m_*^{(j)}$ and the temporal gain $\mathrm{tgain}_i^{(j)} = (t_\dagger^{(j)} - T_i^{(j)})/t_\dagger^{(j)}$. Using these, we define the *Performance-Time Ratio* $\mathrm{ptr}_i^{(j)} = \mathrm{gap}_i^{(j)} \,/\, \mathrm{tgain}_i^{(j)}$ quantifying the trade-off between performance loss and time savings. To further enable comparison across datasets, we normalize the PTR values to the range $[0, 1]$, such that values closer to 1 (0) indicate a better (poorer) performance-time trade-off. When computing the cumulative distribution of these normalized values, we count how many PTR values are less than or equal to a threshold $\tau \in [0, 1]$ for each algorithm $i$. Hence, the resulting plot indicates how often an algorithm achieves favorable performance-time trade-offs. Curves that rise more steeply and reach higher proportions for lower $\tau$ values correspond to better overall performance-time characteristics. A red dashed corner frame in the top-left highlights the optimal trade-off.

We highlight the best performing models, while keep the others grayed out to reduce the clutter. On the left of Figure 37, the performance profiles are shown w.r.t. the measured inference time. The evaluation shows that GBDT models yield high performance-time ratios. LightGBM leads the performance-time ratios, followed by CatBoost, ResNet, and XGBoost. Although the AutoML framework AutoGluon shows strong performance values as discussed in more detail in Appendix G, this entails a higher computational burden resulting in increased temporal costs. The right plot shows the equivalent performance profiles w.r.t. the measured total time. Notably, TabICL and LimiX achieve strong performance-time ratios, but this also due to other methods undergoing HPO. LightGBM and XGBoost catch up later.

**Small-Data Domain.** In Figure 38, the performance profiles are shown w.r.t. the measured inference time (left) and the measured total time (right) in the small-data regime. Mitra is superior to the competitors for smaller thresholds for the performance ratios. The models CatBoost and LightGBM yield the best performance-time ratios over the whole range of performance ratios, followed by XGBoost and MLP. On the right side, the performance plots are given w.r.t. the total time. Likewise to the inference time, Mitra shows superior performance for smaller ratios $\tau$. The models following an in-context learning paradigm, TabICL, LimiX, and TabPFNv2, show second best results and obtain competitive results to Mitra for mid-range values for the performance ratio.

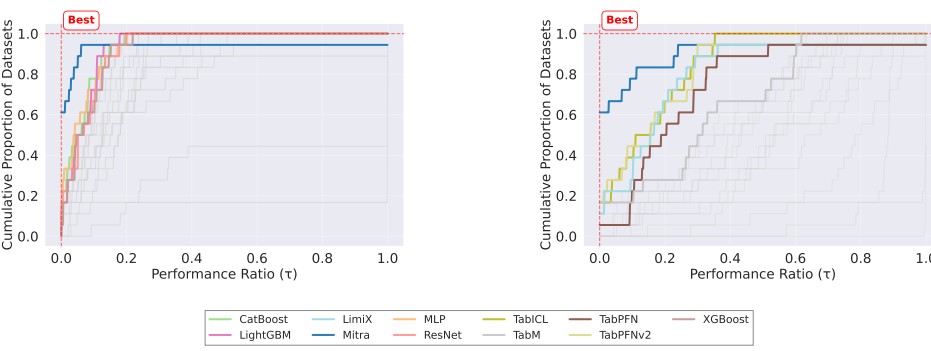

Figure 38: Performance profiles in the small data domain. **Left:** Performance profiles based on inference time. **Right:** Performance profiles based on total time. Steeper curves indicate better overall performance and efficiency across datasets.

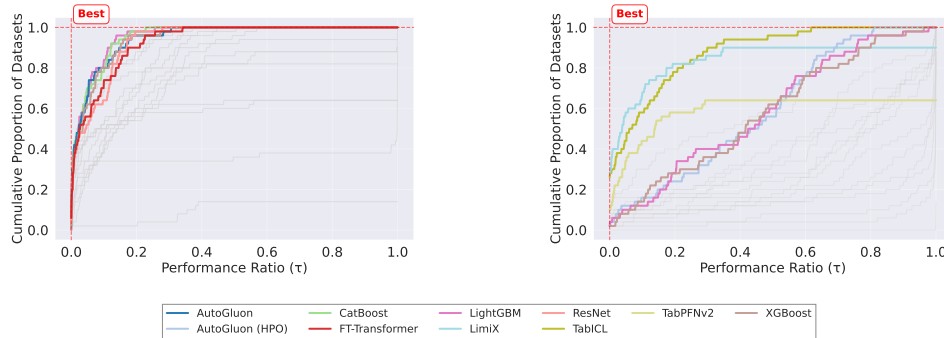

Figure 39: Performance profiles in the large data domain. **Left:** Performance profiles based on inference time. **Right:** Performance profiles based on total time. Steeper curves indicate better overall performance and efficiency across datasets.

**Large-Data Domain.** In Figure 39, the performance profiles are shown w.r.t. the measured inference time (left) and the measured total time (right) in the large-data regime. Regarding the inference time, the model family of gradient boosted decision trees, LightGBM, CatBoost, and XGBoost, show strong performance compared to all other competitors. It is followed by the models AutoGluon as an AutoML-driven approach and ResNet as a FFN approach. FT-Transformer shows up with best results for the transformer-based approaches. When considering the total amount of time, the picture is similar to the small-data regime where models following the in-context learning paradigm, LimiX, TabICL, and TabPFNv2 show the best trade-offs. It is followed by the models XGBoost and LightGBM as GBDTs, and AutoGluon with HPO.

# I INFLUENCE OF META-FEATURE CHARACTERISTICS ON THE PREDICTIVE PERFORMANCE

Following the methodology of McElfresh et al. (2023), we employed the PyMFE library (Alcobaça et al., 2020) to extract meta-features from the datasets used in our study. Specifically, we extracted General, Statistical, and Information-theoretical meta-features.

Figure 40 displays the mean correlation coefficients of the most significant meta-features concerning the performance of all methods, averaged across datasets. To produce this plot, we first calculate the correlation coefficients between each method's performance and each meta-feature for all datasets. For each method, we then selected the top $k$ meta-features with the highest absolute value of the correlation coefficients across all datasets, identifying them as the most important ones for that specific method. We compiled a list of significant meta-features by taking the union of these top meta-features across all methods.

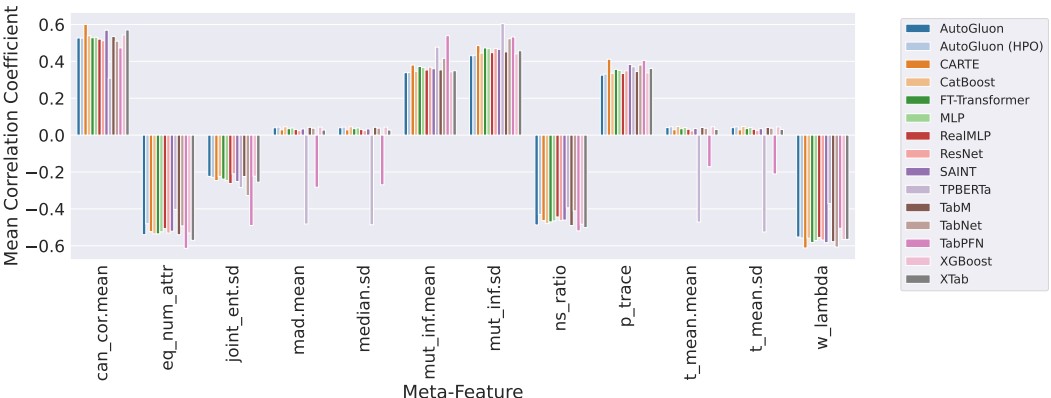

Figure 40: Mean correlation coefficient of most important meta-features with performance across all methods

For each meta-feature in this combined list, we computed the mean of its correlation coefficients across datasets for all methods. Figure 40 illustrates that TabPFN and TPBERTa significantly deviate from the overall pattern observed in the other methods, exhibiting negative correlations for the meta-features `mad.mean`, `median.sd`, `t_mean.mean`, and `t_mean.sd`. To determine whether this deviation is due to the inherent properties of these methods or is a consequence of the limited number of datasets they were evaluated on, we repeated the analysis for all methods using only the datasets on which TabPFN and TPBERTa were run.

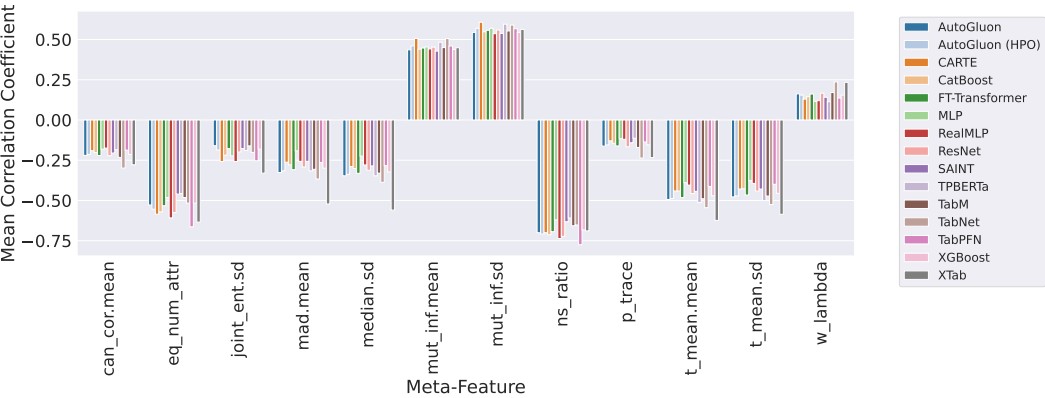

Figure 41: Mean correlation coefficient of most important meta-features with performance across all methods on datasets with results for TabPFN and TPBERTa

Figure 41 demonstrates that when the analysis is confined to only the intersection of datasets on which TabPFN and TPBERTa were evaluated, the previously observed deviation disappears. This suggests that the initial divergence was likely due to the limited number of datasets rather than the inherent properties of these methods. Therefore, it appears that all methods, regardless of their method families, are similarly influenced by the meta-features in terms of their predictive performance. In general, the strongest correlation coefficients are observed for three meta-features: `eq_num_attr`, `w_lambda`, and `can_cor.mean`.

The `eq_num_attr` meta-feature, which measures the number of attributes equivalent in information content for the predictive task, exhibits a strong negative correlation with performance across most methods. This suggests that methods generally perform worse on datasets with high feature redundancy, likely due to challenges in handling overlapping information or overfitting. Similarly, the `w_lambda` meta-feature, which computes Wilk's Lambda to quantify the separability of classes in the feature space, also shows a consistently negative correlation. This indicates that methods struggle on datasets with poor class separability, where the features do not adequately distinguish between the target classes. Conversely, the `can_cor.mean` meta-feature, representing the mean canonical correlation between features and the target, shows a positive correlation with performance. This implies that methods perform better on datasets where the features are strongly predictive of the target variable, highlighting their reliance on well-aligned feature-target relationships.

Generally, the findings align with the common intuition of the performance of ML methods under sub-optimal class separation and further validate the empirical protocol of our study. For detailed explanations of the meta-feature abbreviations used in the plots, please refer to the official PyMFE documentation[16].

---

[16]`https://pymfe.readthedocs.io/en/latest/auto_pages/meta_features_description.html`

