# OpenReview forum: "Tabular Data: Is Deep Learning All You Need?"
_ICLR.cc/2026/Conference — Submitted to ICLR 2026_

### Official Review · Reviewer_oVYN · 2025-10-17

**Soundness:** 3
**Presentation:** 3
**Contribution:** 3
**Rating:** 8
**Confidence:** 4

**Summary:**

This is a rigorous, large-scale empirical study asking whether modern deep learning now consistently outperforms classical methods on tabular classification. The authors benchmark 17 methods (foundation models, dataset-specific NNs, GBDTs, and AutoML) across 68 OpenMLCC18 datasets with nested 10-fold CV, model-based HPO (Optuna/TPE; up to 100 trials or 23h), and a refit-after-HPO protocol. Headline results: deep learning—especially in-context foundation models (TabICL, TabPFNv2) and the MLP ensemble TabM—dominates GBDTs overall; refitting after HPO improves ranks and can reshuffle winners; and DL wins across data-regimes, including “small data,” contrary to long-standing folklore. Code is released for reproducibility.

**Strengths:**

The nested CV + model-based HPO + refitting design is stronger than prior surveys relying on random search and no refit. The authors explicitly motivate these choices and quantify their impact.

Inclusion of both in-context tabular FMs and strong MLP ensembles (TabM), alongside AutoML, makes conclusions relevant to practice in 2025.

Rank distributions, one-vs-one win-rates, CD diagrams, and regime plots tell a consistent story (TabICL/TabPFNv2 ≳ TabM ≳ CatBoost/XGBoost).

Actionable insights: (i) Refitting after HPO improves results and may change leaderboards; (ii) HPO helps several methods materially (e.g., XGBoost, XTab), with detailed hyperparameter importance analyses.

Contrary to the common belief that trees rule small tables, TabICL/TabPFNv2 (and TabM) are highly competitive, often surpassing CatBoost/LightGBM.

**Weaknesses:**

Many plots focus on ranks/win-rates. Please complement with absolute ROC-AUC deltas (mean/median ± CIs) and per-dataset paired tests to quantify practical margins.

You note ~8M evaluations; please add wall-clock/energy summaries and cost-normalized leaderboards (e.g., AUC per hour) to contextualize results for practitioners.

Some models use bespoke preprocessing; batch size is heuristic due to memory. A short ablation on preprocessing/batch sensitivity for a few methods would strengthen fairness claims.

**Questions:**

Add tables with median ΔAUC vs top GBDT per dataset family (small/medium/large), with 95% CIs.
Provide cost-performance plots (AUC vs GPU-hours) and a practitioner-oriented “best under X hours” guide.
Include a compact regression panel or, at minimum, temper the title/claims to “classification.”

---

> ### Author Response · Authors · 2025-11-23
>
> We thank the reviewer for the valuable feedback and for the positive evaluation of our work. Below we address the main concerns from the reviewer:
>
> **Regarding: “Add tables with median ΔAUC vs top GBDT per dataset family (small/medium/large), with 95% CIs.”**
>
> Following the reviewer’s suggestion we have added the requested analysis in Appendix F, Subsection F.6 (Tables 37–40). For each dataset family: small (datasets with at most 1000 instances), medium (1000–10000 instances), and large (more than 10000 instances), we now report tables (Tables 37–39) containing the median and mean $\Delta$ROC-AUC of each method relative to the best GBDT (CatBoost, XGBoost, or LightGBM) on that dataset, along with 95% confidence intervals for the mean. In addition, we provide a complementary table (Table 40) that reports, across all datasets, the median and mean absolute $\Delta$ROC-AUC to the best method per dataset, again with 95% confidence intervals for the mean.
>
> **Regarding: Provide cost-performance plots (AUC vs GPU-hours) and a practitioner-oriented “best under X hours” guide.**
>
> We thank the reviewer for the insightful suggestion. We would like to point the reviewer to Appendix H, where we have provided a cost efficiency analysis of the different methods and their performance over the total time.
>
> **Regarding: Include a compact regression panel or, at minimum, temper the title/claims to “classification.”**
>
> We thank the reviewer for the suggestion. When accepted for the camera-ready version we plan to introduce an additional limitation section where we explicitly point-out that our study focuses on classification tasks and not regression tasks. Moreover, we modified the abstract and contribution bullet list according to the reviewer's suggestion to clarify any confusion it might arise. We are also willing to modify the title of the paper if the reviewer has a better suggestion that clarifies the confusion.
>
> We believe to have addressed the concerns listed by the reviewer and we would welcome any additional questions the reviewer might have.

---

### Official Review · Reviewer_qEWD · 2025-10-23

**Soundness:** 3
**Presentation:** 3
**Contribution:** 2
**Rating:** 2
**Confidence:** 5

**Summary:**

The paper presents a survey and benchmarking study of ML and deep learning methods on a diverse collection of tabular datasets. It also offers useful insights into the role of hyperparameter optimization and the cost-performance trade-offs between different model families. However, the main weakness is its novelty: the recent TabArena benchmark [1] already provides an extensive, reproducible, and **continuously maintained** benchmarking effort. This raises questions about how the present work advances the state of knowledge beyond TabArena.



References:
[1] Erickson, Nick, Lennart Purucker, Andrej Tschalzev, David Holzmüller, Prateek Mutalik Desai, David Salinas, and Frank Hutter. "Tabarena: A living benchmark for machine learning on tabular data." arXiv preprint arXiv:2506.16791 (2025).

**Strengths:**

- The manuscript addresses a central challenge in tabular learning e.g. fairly benchmarking ML and DL methods on tabular datasets
- It provides valuable insights for practitioners, particularly regarding training strategies and hyperparameter sensitivity.
- The paper is clearly written and easy to follow.

**Weaknesses:**

The primary weakness is limited novelty. The TabArena work [1] already provides: (1) a large-scale, reproducible benchmarking ecosystem, (2) a live leaderboard that can be continuously updated, (3) a carefully curated dataset collection, and (4) strong baselines with advanced evaluation protocols. Moreover, TabArena reports similar empirical conclusions—for example, that DL methods can outperform classical methods in certain regimes. As a result, it is unclear what unique contribution this paper makes beyond prior work.

**Questions:**

I have one main question: What are advantages of the presented work over the TabArena?

Also, If the intention is to study a simpler or more controlled benchmarking setting than TabArena, this should be explicitly justified, along with the scientific insight that such a restriction reveals.

I'm ready to update my rating depending on the authors response.

---

> ### Author Response · Authors · 2025-11-22
>
> We thank the reviewer for the valuable comments and for evaluating our insights as useful for the practitioners.
>
> **Regarding the limited novelty and advantages to TabArena.**
>
> We hope that after the discussion with the AC the reviewer agrees that our work is not limited in novelty. While we agree that there exists an overlap with TabArena, we would like to note the following differences:
>
> * Refitting: We would like to point out to the reviewer that this is an aspect that is not followed by the related works, as we show in Table 1. As described in Research Question 4 (Section 5), we analyze the impact of refitting and note that refitting significantly impacts predictive performance, with refitted models obtaining better performance. Moreover, we observed that the overall model ranking is affected when refitting is incorporated.
> * Model-based HPO optimization: We would like to point out to the reviewer that TabArena employs random search for hyperparameter optimization. In our study, we incorporate model-based hyperparameter optimization, which in general, outperforms random search as shown in prior work [1][2][3].
> * We include meta-learned non-ICL methods.
>
> Furthermore, we see our work as complementary to TabArena, providing the following additional valuable insights to the community:
>
> * An investigation of how hyperparameter optimization affects performance, and which hyperparameters have the highest influence on the predictive performance of individual methods.
> * An investigation of the impact of refitting in the predictive performance
> * An investigation into the cost-efficiency of the considered baselines
> * An investigation into how the different methods behave in different dataset regimes
>
> Lastly, we added 3 new baselines to our experimental protocol, namely ModernNCA, Mitra, and LimiX, where the latter achieves state-of-the-art results. We believe the additions further enhance our experimental protocol and provide a view into the current state of the tabular data domain.
>
> We believe to have cleared out the main concern from the reviewer on the novelty of our work and on the differences and contributions related to TabArena. Based on the clarifications we would kindly invite the reviewer to update the score and recommend acceptance.
>
>
> [1] Lindauer, M., Eggensperger, K., Feurer, M., Biedenkapp, A., Deng, D., Benjamins, C., ... & Hutter, F. (2022). SMAC3: A versatile Bayesian optimization package for hyperparameter optimization. Journal of Machine Learning Research, 23(54), 1-9.
>
> [2] Falkner, S., Klein, A., & Hutter, F. (2018, July). BOHB: Robust and efficient hyperparameter optimization at scale. In International conference on machine learning (pp. 1437-1446). PMLR.
>
> [3] Bergstra, J., Bardenet, R., Bengio, Y., & Kégl, B. (2011). Algorithms for hyper-parameter optimization. Advances in neural information processing systems, 24.

---

> ### Comment · Reviewer_qEWD · 2025-11-24
> **response**
>
> Dear Authors,
>
> Thank you for addressing my question about clarifying the differences between your work and TabArena, as well as integrating this discussion into the paper.
>
> Regarding the novelty,  the submitted survey (as you named it) still has substantial overlap with TabArena this is apparent even from the abstract:
>
> > However, recent deep learning models have not been subjected to a comprehensive evaluation under conditions that allow for a fair comparison with existing classical approaches. This situation motivates an investigation into whether recent deep-learning paradigms outperform classical ML methods on tabular data. Our survey fills this gap by benchmarking seventeen state-of-the-art methods, spanning neural networks, classical ML and AutoML techniques.
>
> I am not an author of the TabArena paper and I am not trying to defend it, TabArena has limitations as well. I also do not think continuing this discussion is productive, since you already addressed it within the paper.
>
> What I consider important, however, is that in its current form the claim that your paper “fills the gap” is at the very least not fully accurate. Next, the refitting step, I do not see particular novelty here, it is intuitive that providing additional data and retraining may change the ranking of models.
>
> However, as I mentioned in my original review, the paper does provide new insights into (deep) tabular learning and it is well written and well organized, I have therefore updated my score.
>
> Lastly, I kindly ask you to be allowed to carry out my review independently, without recommendations on how I should update the score.

---

> ### Author Response · Authors · 2025-11-24
>
> We would like to thank the reviewer for the reply and we apologize if our personal recommendation was perceived as putting any pressure on the review process.
>
> Again, we would like to emphasize that if there exists any overlap with TabArena this is a testament of the quality of our work. At the same time, the part that the reviewer quotes, we still believe to be true. Because we advocate for a fair comparison and we believe that: i) empirically we have demonstrated that refitting positively affects performance, and ii) model-based HPO has already been shown in previous works that it outperforms non-model based HPO methods. [1][2][3]
>
> Concerning the point that refitting is intuitive, we respectfully note that, despite this “intuition”, prior large-scale benchmarking efforts, including TabArena, do not make use of refitting, as we highlight in Table 1. We therefore believe that making this design choice explicit, and evaluating its impact, is a meaningful and novel contribution.
> Additionally, we believe that our contributions are not limited to refitting and model based HPO. As previously mentioned, we provide:
> - Detailed insights into head-to-head comparison of different method families (DL vs. GBDT, meta-learned vs. dataset specific, ICL vs. non-ICL)
> - An investigation of how hyperparameter optimization affects performance, and which hyperparameters have the highest influence on the predictive performance of different methods.
> - An investigation into the cost-efficiency of the considered baselines
> - An investigation into how the different methods behave in different dataset regimes
>
> Lastly, compared to TabArena we additionally include non-ICL meta-learned methods. If the reviewer would be interested in a specific analysis we would be happy to provide it, or if the reviewer has any additional concerns we are happy to answer them.
>
> [1] Lindauer, M., Eggensperger, K., Feurer, M., Biedenkapp, A., Deng, D., Benjamins, C., ... & Hutter, F. (2022). SMAC3: A versatile Bayesian optimization package for hyperparameter optimization. Journal of Machine Learning Research, 23(54), 1-9.
>
> [2] Falkner, S., Klein, A., & Hutter, F. (2018, July). BOHB: Robust and efficient hyperparameter optimization at scale. In International conference on machine learning (pp. 1437-1446). PMLR.
>
> [3] Bergstra, J., Bardenet, R., Bengio, Y., & Kégl, B. (2011). Algorithms for hyper-parameter optimization. Advances in neural information processing systems, 24.

---

### Official Review · Reviewer_81kb · 2025-10-29

**Soundness:** 3
**Presentation:** 3
**Contribution:** 2
**Rating:** 2
**Confidence:** 5

**Summary:**

This paper performs an extensive and thourough comparison of the recent tabural ML approaches. The authors demonstrate that for classification tasks DNN-based methods outperform GBDTs on the benchmark of classification problems.

**Strengths:**

The fact that refitting (after performing hyperparameter optimization) is more beneficial for GBDTs than for DNNs is interesting and new to me, at least I have never met it in the literature.

**Weaknesses:**

(1) The paper investigates only classification problems and it is not clear from the title/abstract/contributions bullet list. The final claim in the abstract can be misleading, for instance, "deep learning methods outperform classical approaches" can be false for regression, see the recent TabArena leaderboard for regression.

(2) The chosen wording can also be misleading. For instance, the claim "nonfinetuned foundation models outperform fine-tuned ones" can be unclear, since rigorously speaking, meta-learned foundational models can be finetuned (e.g. see "On finetuning tabular foundation models", Rubachev et al.) and then the authors' claim is false.

(3) The authors do not include the recent non-parametric models (TabR, ModernNCA) in their evaluation, despite these models being strong players in the field. For instance, TabICL paper demonstrated that TabR and ModernNCA can outperform meta-learned models. I believe that in such kind of papers all the recent models should be used.

(4) In my opinion, most of the news from this submission are already known by the people in the field. For instance, TabM paper already reported the advantage of DNNs over GBDT, as well as TabICL and TabArena papers.

**Questions:**

I do not have any specific questions for the rebuttal, if the authors will address by concerns from the weaknesses section, I will appreciate that.

---

> ### Author Response · Authors · 2025-11-22
>
> We thank the reviewer for the insightful comments. Below we address the concerns listed by the reviewer:
>
> **In my opinion, most of the news from this submission are already known by the people in the field. For instance, TabM paper already reported the advantage of DNNs over GBDT, as well as TabICL and TabArena papers.**
>
> We agree with the reviewer that there exists an overlap with TabArena related to the claim of deep learning outperforming classical approaches, however, we hope that after the discussion with the AC, the reviewer agrees that this is a testament to the quality of our conducted investigation.
> Secondly, we would like to point out that the message of deep learning being better than classical approaches for tabular data is not the only contribution of our work. We additionally provide:
>
> * An investigation of how hyperparameter optimization affects performance, and to which hyperparameters have the highest influence on the predictive performance of individual methods.
> * An investigation of refitting in the predictive performance
> * An investigation into the cost-efficiency of the considered baselines
> * An investigation into how the different methods behave in different dataset regimes
>
> Thirdly, we would like to point out, that our experimental protocol brings the following additions compared to prior works, as summarized in Table 1 (which now additionally includes TabArena):
>
> * Refitting: We would like to point out to the reviewer that this is an aspect that is not followed by the related works, as we show in Table 1. As described in Research Question 4 (Section 5), we analyze the impact of refitting and note that refitting significantly impacts predictive performance, with refitted models obtaining better performance. Moreover, we observed that the overall model ranking is affected when refitting is incorporated.
> * Model-based HPO optimization: We would like to point out to the reviewer that TabArena employs random search for hyperparameter optimization. In our study, we incorporate model-based hyperparameter optimization, which in general, outperforms random search as shown in prior work [1][2][3].
> * We include meta-learned non-ICL methods.
>
> [1] Lindauer, M., Eggensperger, K., Feurer, M., Biedenkapp, A., Deng, D., Benjamins, C., ... & Hutter, F. (2022). SMAC3: A versatile Bayesian optimization package for hyperparameter optimization. Journal of Machine Learning Research, 23(54), 1-9.
>
> [2] Falkner, S., Klein, A., & Hutter, F. (2018, July). BOHB: Robust and efficient hyperparameter optimization at scale. In International conference on machine learning (pp. 1437-1446). PMLR.
>
> [3] Bergstra, J., Bardenet, R., Bengio, Y., & Kégl, B. (2011). Algorithms for hyper-parameter optimization. Advances in neural information processing systems, 24.

---

> > ### Author Response · Authors · 2025-11-22
> >
> > **The paper investigates only classification problems and it is not clear from the title/abstract/contributions bullet list. The final claim in the abstract can be misleading, for instance, "deep learning methods outperform classical approaches" can be false for regression, see the recent TabArena leaderboard for regression.**
> >
> > We believe we have made it clear in our Experimental protocol (Line 173, Lines 183-184) that we focus on classification tasks,  Additionally, when accepted for the camera-ready version we plan to introduce an additional limitation section where we explicitly point-out that our study focuses on classification tasks and not regression tasks. Moreover, we modified the abstract and contribution bullet list according to the reviewer's suggestion to clarify any confusion it might arise. We are also willing to modify the title of the paper if the reviewer has a better suggestion that clarifies the confusion.
> > Looking at Figure A.4 in the TabArena paper, we observe that our claim “deep learning approaches outperform classical approaches”  still stands when ensembling is incorporated. This is further verified by recent work [1] (Table 4), where deep learning methods do outperform classical approaches in a regression setting.
> >
> > [1] Zhang, X., Maddix, D. C., Yin, J., Erickson, N., Ansari, A. F., Han, B., ... & Wang, B. Mitra: Mixed Synthetic Priors for Enhancing Tabular Foundation Models. In The Thirty-ninth Annual Conference on Neural Information Processing Systems.
> >
> > **The chosen wording can also be misleading. For instance, the claim "nonfinetuned foundation models outperform fine-tuned ones" can be unclear, since rigorously speaking, meta-learned foundational models can be finetuned (e.g. see "On finetuning tabular foundation models", Rubachev et al.) and then the authors' claim is false.**
> >
> > We thank the reviewer for pointing out the flaw in our wording. We agree with the reviewer that the chosen wording is misleading.
> >
> > With our initial wording we wanted to convey that meta-learned non-ICL methods do not outperform meta learned ICL methods. To that aim, we have modified the submission so that “fine-tuned foundation models” is replaced with “meta-learned non-ICL models”, following the reviewer’s suggestion.
> >
> > **The authors do not include the recent non-parametric models (TabR, ModernNCA) in their evaluation, despite these models being strong players in the field. For instance, TabICL paper demonstrated that TabR and ModernNCA can outperform meta-learned models. I believe that in such kind of papers all the recent models should be used.**
> >
> > We would like to thank the reviewer for raising an interesting point. We agree that non-parametric models are important baselines in modern tabular deep learning. To that aim, we followed the suggestion from the reviewer and incorporated ModernNCA into our experimental protocol.
> > Moreover, we additionally added two recent tabular foundation models, Mitra and LimiX, the latter which achieves state-of-the-art results.
> > We believe the additions further strengthen our experimental results and provide a view into the current state of the domain of tabular data.
> >
> > We believe we have addressed all the concerns raised by the reviewer. We would kindly ask the reviewer to increase the score and recommend acceptance based on the provided clarifications.

---

### Official Review · Reviewer_15cs · 2025-10-29

**Soundness:** 3
**Presentation:** 3
**Contribution:** 1
**Rating:** 2
**Confidence:** 4

**Summary:**

This paper re-examines the question of deep learning vs decision trees on tabular data. It uses the OpenML-CC18 benchmark suite and proposes it's own experimental protocol (with main difference from prior work being refitting). The paper compares AutoML system (in autogluon), in-context learning (ICL) based tabular foundation models (in TabPFN(v1, v2) and TabICL), parametirc transformer-based foundation models that require finetuning (XTab, TPBERTa, CARTE), GBDT and neural network baselines (in TabM, RealMLP, FT-Transformer, SAINT, ResNet and MLP). The paper core message seems to be the highlight of paradigm shift, where present-day tabular DL models outperform GBDTs. Mostly through TabM and ICL-based foundation models. The paper also makes multiple finer observations like ICL-based foundation models being superior to finetuning-based parametric ones or looking at the dataset size win-rate profiles and model hyperparameter sensitivity.

**Strengths:**

The first (broader) strength and argument for the paper is in it's ovearall message and survey-ish nature. The field of Tabular Deep Learning did progress in recent years and the paper manages to convey this message. I think it may be important for the broader DL community to know about the subfield advances.

Another strong aspect (much less important in my view) is in more subtle findings that are novel:
- RealMLP performance seems to be much less good compared to it's resulst on TabArena - this is interesting and warrants at least some investigation? Does the setup difference matter this much?
- XTab and TPBERTa lacking behind ICL-based foundation models - this result is important to set the record straight (but may also warrant some explanation of why this is the case, compared to the success in the original papers)
- An observation that small datasets is where the ICL-based foundation models have more wins
- Demonstration that refitting provides some improvements

**Weaknesses:**

The core weakness of this work is in lacking the field context. By lacking context I mean that the paper for it's main goal (that seems to be conveying the message of progress in DL for tabular data), misses a bit on where the field is actually at.

First, I think that recent focus on dataset quality in benchmarks brought up in recent work ([Erickson et al.](https://arxiv.org/abs/2506.16791), [Rubachev et al.](https://arxiv.org/abs/2406.19380), [Tschalzev et al.](https://arxiv.org/abs/2503.09159) should be discussed and taken into account in a new "benchmark"/"revisiting the state of X" paper.

Second, It is very hard to discount the existence of TabArena (([Erickson et al.](https://arxiv.org/abs/2506.16791)), which is not discussed in the current paper. In my view TabArena is a step in the right direction for the field in general. There is a certain blueprint to revisiting DL for tabular data publications (e.g. [Tabzilla](https://arxiv.org/abs/2305.02997), [TALENT](https://arxiv.org/abs/2407.00956), [MultiTab](https://arxiv.org/abs/2505.14312), and I'm probably missing some more, but you see the point) which is roughly take some dataset suite, take some recent models, compare. I believe that it is wastefull to reinvent the evaluation over and over again each year. From my point of view the field is slowly growing past that blueprint: more downstream task relevant benchmarks being introduced (like TabReD [Rubachev et al.](https://arxiv.org/abs/2406.19380) - covering Industry ML use-cases or [Barkov et al.](https://arxiv.org/abs/2508.09888) covering digital soil mapping applications, and I think we should have much more of that kind of work). With all the more specific benchmarks, TabArena stands for the general MMLU/GLUE-like proxy for researcher fast iteration.

Furthermore, the present paper study misses some important baselines in ModernNCA, TabR and the more recent LimiX tabular foundation model. In contrast, TabArena has the first two (covering a paradigm important in modern tabular deep learning), and the LimiX (which is recent and may be hard to ask to put in an ICLR submission) is on the way there https://github.com/autogluon/tabarena/pull/208

To circle back and summarize my argument: I think that the paper may be important to share with the broader DL community at ICLR, but as I outlined above, it feels a bit disconnected from many recent developments in the field of tabular deep learning.

There are other strenghts that are related to more specific findings, but these lack deeper analysis and provide limited insight or little to no-explanation.

**Questions:**

I am very skeptical that it is possible to address the core weakness outlined above, but I believe that I can be swayed by deeper analysis, specifically in two of the following areas:

- Why does RealMLP performance differ in your protocol and in tabarena?
- Why do non-ICL foundation models fail?
- A  more  in-depth study of dataset-size and ICL models dominance.

---

> ### Author Response · Authors · 2025-11-21
>
> We thank the reviewer for the valuable comments and appreciate the assessment that our work is “important to share with the broader community.”
>
> **Regarding TabArena and related work:**
>
> Following the reviewer’s suggestion, we incorporated TabArena into the related work section and Table 1 to highlight how our evaluation protocol delineates from theirs. We hope the reviewer agrees with us, after the discussion with the AC, that the fact that some of our claims overlap with TabArena’s is a testament of the high quality of our work and that we did not reinvent the evaluation. Additionally, we provide the following differences:
>
> - **Refitting:** We would like to point out to the reviewer that this is an aspect that is not followed by the related works, as we show in Table 1. Then, as described in Research Question 4 (Section 5), we analyze the impact of refitting and note that refitting significantly impacts predictive performance, with refitted models obtaining better performance. Moreover, we observed that the overall model ranking is affected when refitting is incorporated.
>
> - **Model-based HPO optimization:** We would like to point out to the reviewer that TabArena employs random search for hyperparameter optimization. While in our study, we incorporate model-based hyperparameter optimization, which in general, outperforms random search as shown in prior work [1][2][3].
> - We include meta-learned non-ICL methods.
>
> Additionally, we would like to point out to the reviewer that we provide an in-depth investigation, Research Question 4 is not the only outcome of the refitting analysis, we would like to kindly direct the reviewer to Appendix C for more details.
>
> Lastly, we would like to point out to the reviewer that these are not the only aspects we investigate, we additionally provide insights on:
>
> - The impact of hyperparameter optimization and the most important hyperparameters per-method (cf. Appendix D).
> - A cost-performance analysis which describes the most efficient methods (cf. Appendix H).
> - An investigation of the method performances in different data regimes (cf. Appendix F).
>
> Because of the limitation in paper length, a part of our contributions are in the Appendix. However, we believe we have done our best to reference them from the main paper in the related research questions.
>
> Based on the aforementioned points, we hope the reviewer agrees that, through a proper evaluation protocol and the accompanying empirical results, our work provides valuable insights for the community.
>
>
> References:
>
> [1] Lindauer, M., Eggensperger, K., Feurer, M., Biedenkapp, A., Deng, D., Benjamins, C., ... & Hutter, F. (2022). SMAC3: A versatile Bayesian optimization package for hyperparameter optimization. Journal of Machine Learning Research, 23(54), 1-9.
>
> [2] Falkner, S., Klein, A., & Hutter, F. (2018, July). BOHB: Robust and efficient hyperparameter optimization at scale. In International conference on machine learning (pp. 1437-1446). PMLR.
>
> [3] Bergstra, J., Bardenet, R., Bengio, Y., & Kégl, B. (2011). Algorithms for hyper-parameter optimization. Advances in neural information processing systems, 24.

---

> ### Author Response · Authors · 2025-11-21
>
> **Missing retrieval baseline and recent in-context learning methods**
>
> We thank the reviewer for raising an interesting point. Based on the reviewer’s comments we added ModernNCA to our work, to cover retrieval methods, an important paradigm in tabular data that we added also to Figure 1. Additionally, we added Mitra and LimiX, two recent state-of-the-art in-context learning methods. As pointed out by the reviewer, LimiX manages to outperform all methods and achieve state-of-the-art results as shown by the updated evaluation section.
>
> We believe the additions strengthen our experimental protocol and provide a much needed view into the current state of the tabular domain.
>
>
> **Regarding RealMLP performance difference compared to tabarena**
>
> We would like to thank the reviewer for raising an interesting point. Firstly, we would like to point out to the reviewer that our results and TabArena’s are similar, if one looks at the non-ensembled performance. Secondly, our work employs refitting and model-based hyperparameter optimization, which is additionally a factor that contributes in the final performances. As we show in our work, both refitting and model-based hyperparameter optimization yield in general better results.
>
>
> **Why do non-ICL foundation models fail?**
>
> We thank the reviewer for the interesting question. What the non-ICL methods have in common is that they require dataset-specific fine-tuning and they do not provide generalization without applying fine-tuning. This is the major differentiating point to ICL methods.
>
> Firstly, we would like to point out that CARTE was designed to interact with column names that provide information, which in reality is lacking in common domain benchmarks. Secondly, XTab instantiates a relatively small FT-Transformer backbone which might be limiting. Lastly, the way that TP-BERTa encodes numerical values might be a limiting factor on how well it can represent the information. We believe these are the factors that contribute to the subpar performance of these methods.
> We have updated Research Question 3 with the insights that we provide to the reviewer.
>
> **A more in-depth study of dataset-size and ICL models dominance.**
>
>
> We thank the reviewer for proposing this interesting investigation. We would like to note that we have already conducted the analysis the reviewer suggests in Appendix F, and that this section was already referenced in Line 348.
>
> We believe we have addressed all the concerns raised by the reviewer. We would kindly ask the reviewer to increase the score and recommend acceptance based on the provided clarifications.

---

### Meta-Review · Area_Chair_JKJz · 2025-12-10

**Summary:**

The main concern of the reviewers was the limited novelty and significance of the findings compared to existing papers and surveys. One core aspect was the comparison with the new TabArena benchmark, which was partially addressed by the authors and is furthermore difficult to assess due to the timelines of the two papers. However, even considering these clarifications, the novelty of the findings remains limited and the impact is not sufficiently demonstrated. The criticism from the reviewers, which not only covered the comparison with TabArena but also addressed the findings in general, was not sufficiently resolved during the rebuttal. Even though the authors made some effort with new experiments, the core issues remain.

**Reviewer Concerns:**

While there were some minor concerns that were addressed by the rebuttal, one aspect was at the center of the discussion and also the most relevant concern raised by the reviewers, namely the limited novelty over existing benchmarks, specifically TabArena. The authors addressed this during the rebuttal in two ways:

1. They highlighted that their paper was indexed before TabArena and even cited by TabArena (noting that disclosing this was discussed with the previous AC).
2. They clarified the novelty of several insights, specifically concerning refitting, model-based HPO, and considering meta-learned non-ICL methods.

While this does resolve some concerns, I do not think that they are sufficiently resolved. First, it is important to clarify that while TabArena indeed cites this paper, it is only a small note about an existing benchmark and does not play any relevant role in the paper. Second, the paper still has to be evaluated at the time of submission and not based on the time it was written, which concerns not only the comparison with TabArena. For instance, the fact that neural and deep learning methods have begun to outperform classical, mostly tree-based approaches has already been demonstrated by TabR (ICLR 2024), TabM (ICLR 2025), RealMLP (NeurIPS 2024), ModernNCA (ICLR 2025), as well as foundation model approaches such as TabPFN (ICLR 2023; Nature 2025) and TabICL (ICML 2025).
While the comparison with TabArena took a large part in the reviews, the core point is that the reviewers do not assign the findings in the paper enough novelty and relevance.

Unfortunately, only one of the three critical reviewers was able to participate in the discussion before it was closed. While that reviewer raised the score slightly (from 2 to 4), they did not sound entirely convinced and still mentioned concerns about the overlap.

Additional note on TabArena, which was acknowledged by the AC: ICLR mentions that papers published after July 24, 2025 do not need to be considered; TabArena was published on June 20, 2025 on arXiv. Being only on arXiv, the lack of this comparison cannot be a basis for rejection.

**Reviewer Scores:**

This paper involved a lot of discussion, and it is hard to tell how the reviewers would have responded if the rebuttal had continued. Reviewer [qEWD] indicated a score increase, which most likely was from 2 to 4. Even though the authors hypothesize that the reviewer would increase the score further, I think they would have maintained the rating of 4, since the concerns were not fully resolved.

Reviewer [81kb] and [15cs] were very sceptical about the impact of this paper in light of the more recent benchmarks. I do not think they were convinced by the fact that this paper was cited by TabArena, since the citation was only a small sidenote and the new insights still remain limited. While I could imagine that they might have increased their scores, I am not certain. If they had increased their scores, I think they would have raised them to a maximum of 4.

Reviewer [oVYN] already gave a score of 8 and most likely retained this score. However, the other reviewers seemed to be much more involved with the topic and more up to date with the current state of research on tabular data benchmarking.

---

### Decision · Program_Chairs · 2026-01-26

Reject